



# What is the Surface Mass Balance of Antarctica? An Intercomparison of Regional Climate Model Estimates

Ruth Mottram[1], Nicolaj Hansen[1,2], Christoph Kittel[3], Melchior van Wessem[4], Cécile Agosta[5], Charles Amory[3], Fredrik Boberg[1], Willem Jan van de Berg[5], Xavier Fettweis[3], Alexandra Gossart[6], Nicole P.M. van Lipzig[6], Erik van Meijgaard[7], Andrew Orr[8], Tony Phillips[8], Stuart Webster[9], Sebastian B. Simonsen[2], and Niels Souverijns[6,10]

[1]DMI, Lyngbyvej 100, Copenhagen, 2100, Denmark
[2]DTU-Space, Kongens Lyngby, Denmark
[3]Laboratory of Climatology, Department of Geography, SPHERES, University of Liège, Liège, Belgium
[4]Institute for Marine and Atmospheric Research Utrecht, Utrecht University, Utrecht, the Netherlands
[5]Laboratoire des Sciences du Climat et de l'Environnement, LSCE-IPSL, CEA-CNRS-UVSQ, Université Paris-Saclay, Gif-sur-Yvette, France
[6]Department of Earth and Environmental Sciences, KU Leuven, Belgium
[7]Royal Netherlands Meteorological Institute, De Bilt, the Netherlands
[8]British Antarctic Survey, High Cross, Madingley Road, Cambridge, UK
[9]UK Met Office, FitzRoy Road, Exeter, Devon, EX1 3PB, UK
[10]Unit Remote Sensing and Earth Observation Processes, Flemish Institute for Technological Research (VITO), Mol, Belgium

**Correspondence:** Ruth Mottram (rum@dmi.dk)

**Abstract.** Antarctic ice sheet mass loss is currently equivalent to around 1 mm year$^{-1}$ of global mean sea level rise. Most mass is lost due to sub-ice shelf melting and calving of icebergs. Ice sheet models of the Antarctic ice sheet have thus largely concentrated on parameterising sub-shelf and calving processes. However, surface mass balance (SMB) is also of crucial importance in controlling the stability and evolution of the vast Antarctic ice sheet. In this paper we compare the performance

of five different regional climate models (COSMO-CLM$^2$, HIRHAM5, MAR3.10, MetUM and RACMO2.3p2) in simulating the near surface climate and SMB of Antarctica. Our results show that, when regional climate models (RCMs) are forced by the ERA-Interim reanalysis, the integrated Antarctic ice sheet ensemble mean annual SMB is $2329 \pm 94$ Gigatonnes (Gt) year$^{-1}$ over the common 1987 to 2015 period. However, individual model estimates vary from $1961 \pm 70$ to $2519 \pm 118$ Gt year$^{-1}$. The large differences are mostly explained by different SMB estimates in West Antarctica and the peninsula as well as around the

Transantarctic mountains. The calculated annual average SMB is very sensitive to the period chosen but over the climatological mean period of 1980 to 2010 the ensemble mean is 2486 Gt year$^{-1}$. The interannual variability in SMB is consistent between the models and dominated by variability in the driving ERA-Interim reanalysis. The declining trend in Antarctic SMB reported in other studies is also very sensitive to period chosen and models disagree on the sign and magnitude of the trend in Antarctic SMB over the ERA-Interim period.

Evaluation of models shows that they simulate Antarctic climate well when compared with daily observed temperature (Pearson correlation of 0.85 and higher) and pressure (bias ranges from -0.39 hPa in HIRHAM5 to -6.01 hPa in MAR with a mean of -3.49 hPa over all models) and nudged models, constrained within the domain as well as at lateral boundaries, perform



better than un-nudged models. We compar modelled surface mass balance with a large dataset of observations which, though biased by undersampling in some regions, indicates that many of the biases in modelled SMB are common between models. The inclusion of drifting snow schemes improves modelled SMB on ice sheet slopes between 1000 and 2000 m where strong katabatic winds form but other regions where precipitation rates are high lack observations needed for the evaluation of different

SMB estimates. Different ice masks have a substantial impact on the integrated total SMB and along with model resolution is therefore factored into our analysis. The majority of the different values for continental SMB are due to differences in modelled precipitation at relatively few grid points in coastal areas. Our analysis suggests that targeting coastal areas for observational campaigns will be key to improving and refining estimates of the total surface mass balance of Antarctica.

**1   Introduction**

The Antarctic Ice Sheet (AIS) is the largest body of freshwater on the planet and thus a potentially important potential contributor to global sea level rise as well as a significant part of the climate system. Studies by Rignot et al. (2011) and Shepherd et al. (2018a) showed the AIS to have had a net loss since at least 2002, current estimates suggest that around 10 % of observed sea level rise since 1993 is from Antarctica, however that rate is increasing (IPCC SROCC Chap. 4 Oppenheimer et al. (2019)).

Most ice loss in Antarctica occurs as a result of submarine melting and calving from ice shelves and recent ice dynamics studies (DeConto and Pollard, 2003; Edwards et al., 2019; Sutter et al., 2016; Shepherd et al., 2018a) have shown that there is potential for rapid ice sheet loss owing to ice sheet dynamics that are currently poorly understood, especially in West Antarctica. Changes in precipitation and increases in surface melt and runoff will change the mass budget and therefore both ice dynamics and the sea level rise contribution from Antarctica in the future. Moreover there has been disagreement between studies focused

on the SMB contribution to the total mass budget of Antarctica and therefore the contribution to sea level rise (SCAMBOS and SHUMAN, 2016; Zwally et al., 2015), that makes it essential to understand potential biases and uncertainties.

Surface mass budget (also known as surface mass balance or climate mass balance (Cogley et al., 2011)) is the difference between accumulation and ablation at the surface of a glacier. In Antarctica, accumulation is derived primarily from solid precipitation, but on local or regional scales wind-driven processes can have a significant effect on accumulation rates. Surface

ablation in Antarctica is primarily a result of evaporation and sublimation due to the high winds and generally dry atmosphere (Scambos et al., 2012; Das et al., 2013; Agosta et al., 2019), although increasing melt rates are documented in some areas (Stokes et al., 2019). In the future a "greenlandification" of the ice sheet climate with increasing melt and refreezing within the snowpack is projected due to anthropogenically induced climate change (Trusel et al., 2018). Currently runoff is a relatively minor contribution (Lenaerts et al., 2019) to mass loss in Antarctica and an increase in snowfall associated with higher

saturated vapour pressure is expected to dominate future changes in SMB, compensating for the projected increase in surface runoff (Krinner et al., 2008; Lenaerts et al., 2016) but the balance between these processes is still a matter of debate. This



makes it even more important to evaluate the effectiveness of modelled precipitation and sublimation across the continent to be able to estimate SMB at present. Accurate SMB estimates are required to both drive ice sheet dynamical models and to accurately partition sea level rise contributions determined from observations. SMB from regional climate models (RCMs) is also used to correct altimetry measurements by accounting for firn compaction processes for remote sensing applications.

The most common way to observe SMB is by geodetic mass balance stakes (Lenaerts et al., 2019) but this is challenging due to the size and environmental conditions in Antarctica and the most practical alternative is to use output from (high-resolution) RCMs to make continent-wide estimates. There are now an increasing number of RCMs downscaling Antarctic climate simulations available via the CORDEX (CoOrdinated Regional climate Downscaling EXperiments) database. CORDEX is a project of the World Climate Research Programme that aims to produce representative ensembles of regional climate models for dif-

ferent regions of the world. The purpose is to better understand regional climate change, assess regional impacts and improve adaptation to future climate conditions (http://www.cordex.org/). In the polar regions, CORDEX simulations can also used to assess the mass budget of the large polar ice sheets, but have not yet been evaluated together for Antarctica. Souverijns et al. (2019) made a 30 years hindcast with COSMO-CLM$^2$, and Agosta et al. (2019) estimated the SMB using MAR, while various versions of RACMO2 have been used to estimate the SMB of the AIS (Van Wessem et al., 2014; van Wessem et al.,

2018) but while both MetUM and HIRHAM5 have been run for the Antarctic domain, evaluation of the SMB results have not yet been published, in peer review literature (Hansen, 2019). Here, we use the framework of the Polar CORDEX project to assess climate model performance in Antarctica for the period 1979-2018 derived from an ensemble of six simulations from five different RCMs. The RCMs cover a range of resolutions, physical and dynamical schemes in the atmosphere and types of surface and snow/ice schemes. This allows us to determine the relative importance of individual model components needed to

accurately model the climate by comparing the modelled SMB against the sparse observational data-sets available in Antarctica. We also investigate some of the uncertainties within the individual models and between the ensemble members.

In this paper we seek to quantify present-day Antarctic SMB and understand the sources of variation as a baseline to assess mass budget changes and better understand drive sea level rise observations and projections both directly in terms of the amount of meltwater added to oceans and indirectly as surface forcing for ice sheet dynamical models (Robel et al., 2019; Nowicki

et al., 2016).

## 2 Methods

We compare six climate simulations made with five different RCMs in the newest available version of the given RCM. However, to provide backwards continuity we also briefly compare three older versions that have been widely used in earlier studies, to examine how results have varied (or not) as RCMs have been developed. We assess the climate of Antarctica in the models

and derive estimates for SMB. All models were forced on the lateral boundaries with the ERA-Interim climate reanalysis (Dee et al., 2011) but downscaling used different grids, over slightly different domains and at different resolutions with slightly different ice masks used in the different model versions (see A1 in the appendix). Some of the models (MAR, RACMO, COSMO-CLM$^2$) were nudged within the domain using upper air relaxation and one model (MetUM) was run as a 12 hour





reinitialised hindcast, while two versions (one high resolution, one lower resolution) of the HIRHAM5 model were allowed to run freely within the domain and forced only on the boundaries.

We first give a brief overview of each of the participating models, summarised in Table 1. The CORDEX protocol (Christensen et al. (2014)) prescribes a simulation domain for Antarctica with a minimum common analysis extent and a resolution of 0.44°.

Lucas-Picher et al. (2012); Lenaerts et al. (2012b); Franco et al. (2012); van Wessem et al. (2018) among others have found that a higher spatial model resolution gives more physically plausible results especially with respect to precipitation processes in areas with steep terrain. Hence, several participating groups have chosen to run their RCMs at higher spatial resolution. Outputs from the different models are compared with each other and the ensemble mean to quantify both the absolute and relative integrated and basin scale SMB for the continent. The models are also compared to SMB observations (including ice

cores and stakes) and near-surface climate observations (surface pressure, temperature and wind speed) measured across the continent.

## 2.1 Models

The model versions included in this paper were selected as fulfilling the two requirements of being the most up-to-date model version as well as being forced on the boundaries with ERA-Interim reanalysis. We include the earlier RACMO version 2.1

and MAR 3.6 as part of the initial SMB comparison as these models have been widely used and are still available for scientific use online; for example, results from RACMO2.1P were used in compiling the IPCC AR5 climate atlas. However, they are no longer considered up to date and have been replaced by RACMO2.3p2 and $MAR_{v3.10}$ respectively therefore we do not consider them in the detailed results analysis in this paper.

### 2.1.1 COSMO-CLM$^2$

**COSMO-CLM**$^2$ is a non-hydrostatic RCM developed at the German Weather service together with an extensive scientific community (Rockel et al., 2008). The model is applied over the Antarctic at a spatial resolution of ~25 km and 40 vertical levels in the atmosphere. The model is forced every 6 hours at the boundaries by ERA-Interim. Additionally this model is coupled to the Community Land Model (version 4.5; Oleson and Lawrence, 2013), with adjustments in the perennial snow proposed by van Kampenhout et al. (2017) to better represent the SMB of ice sheets (COSMO-CLM$^2$). Apart from this,

several model parameters were adjusted for polar regions, particularly those related to the turbulent kinetic energy scheme and the cloud scheme. A full description of the setup over Antarctica including an evaluation of its performance in simulating the Antarctic climate and SMB is available in Souverijns et al. (2019). In this paper, precipitation minus sublimation is taken as a proxy for the SMB.

### 2.1.2 HIRHAM5

**HIRHAM5** is an RCM developed at the Danish Meteorological Institute and run in this study at both low (∼50 km) and high (∼12 km) resolution, with all other model elements being kept identical. The model combines the atmospheric dynamics of



the HIRLAM7 numerical weather prediction model (Eerola, 2006), and the physics of the ECHAM5 global climate model (Roeckner et al., 2003). There are 31 vertical levels in the atmosphere and the model is forced at 6 hourly intervals on the lateral boundaries with temperature, pressure, relative humidity and the wind vectors. Sea surface temperatures (SST) and sea ice concentration (SIC) are forced on the lower boundary at daily intervals. The set-up for Antarctica is similar to that of

Lucas-Picher et al. (2012) in Greenland, that is with only a very simple surface physics scheme over glacier ice. A subsurface scheme developed in Greenland by Langen et al. (2017) is currently undergoing optimisation for Antarctic SMB processes but was not available for use in these simulations. The model outputs of precipitation, evaporation and sublimation are therefore used to make a simple SMB calculation.

### 2.1.3   MetUM

The UK Met Office Unified Model (MetUM) is a numerical modelling system based on non-hydrostatic dynamics (Walters et al., 2017), which can be run either as a global model or a regional mesoscale model e.g Orr et al. (2015). Here we run version 11.1 of the mesoscale model over the standard Antarctic CORDEX domain at a spatial resolution of 50 km and 70 vertical levels (reaching up to 80 km). The mesoscale model is nested within a global version of the MetUM with a horizontal resolution of N320 (i.e. 640 × 480 longitude-latitude grid implying a nominal 40 km horizontal mesh), which was initialised

by ERA-Interim. The model was used to run a series of consecutive twice-daily 24-hour forecasts at 00 and 12 UTC from the beginning of 1980 to the end of 2018. The first 12 hours of each forecast were discarded as spin-up, with the remaining output concatenated together to form a continuous time-series. Note that although the mesoscale model includes a multi-layer snow scheme (Walters et al., 2019), these simulations used a simplified single-layer scheme with for example no refreezing (Cox et al., 1999) and therefore the simplified SMB was calculated based on ouput precipitation and sublimation and evaporation.

### 2.1.4   MAR$_{v3.10}$

The «Modèle Atmosphérique Régional»(MAR) (Gallée and Schayes, 1994) is a hydrostatic RCM specifically designed for polar areas (e.g., Fettweis et al., 2017; Kittel et al., 2018; Agosta et al., 2019). The model has 24 atmospheric vertical levels and an horizontal resolution of 35 km. MAR is coupled to the 1-D multilayer surface scheme SISVAT (Soil Ice Snow Vegetation

Atmosphere Transfer; De Ridder and Gallée, 1998), which simulates mass and energy fluxes between the atmosphere and the surface. The snow-ice module, based on the CROCUS model (Brun et al., 1992), represents the evolution of the snowpack for 30 snow layers through subroutines of snow metamorphism, surface albedo, meltwater runoff, percolation, retention and refreezing. MAR is forced with ERA-Interim every 6 hours over 1979 – 2018 at its atmospheric lateral and upper boundaries (pressure, wind, specific humidity and temperature at each vertical level) and over the ocean surface (SST and SIC).

Furthermore, an upper-air relaxation is used to constrain the MAR general atmospheric circulation (van de Berg and Medley, 2016). Relative to previous studies over the Antarctic ice sheet (Kittel et al., 2018; Agosta et al., 2019), the version used in this study (MAR$_{v3.10}$) only improves the cloud lifetime, the model stability and its computational efficiency enhancing a larger independence of MAR to its timesteps. Furthermore, the definition of the Antarctic ice sheet mask has also been improved by





taking into account rock outcrops. An extensive description of the adaptation of MAR to the Antarctic ice sheet can be found in Agosta et al. (2019).

### 2.1.5 RACMO2.3p2

The Regional Atmospheric Climate Model RACMO2.3p2 combines the dynamical process of the High Resolution Limited
Area Model (HIRLAM) (Undén et al., 2002) with the physics package CY33r1 of the European Centre for Medium-range Weather Forecasts (ECMWF) Integrated Forecast System (IFS). RACMO2.3p1 was built by porting the polar physics components that were part of RACMO2.1P into the standard climate model RACMO2.3 developed at the Royal Netherlands Meteorology Institute (KNMI). RACMO2.3p2 is the follow-up of RACMO2.3p1 and has been applied to the polar ice sheets of Greenland and Antarctica by the Institute for Marine and Atmospheric research Utrecht (IMAU). RACMO2.3p2 includes a multilayer
snow model that calculates melt, percolation, refreezing and runoff of liquid water (Ettema et al., 2010). RACMO2.3p2 also uses a prognostic scheme for snow grain size used to calculate surface albedo (Kuipers Munneke et al., 2011); and a drifting snow routine that simulates the interaction of drifting snow with the surface and the lower atmosphere (Lenaerts et al., 2012a). For this study, the model operates at a horizontal resolution of ∼27 km, with 40 vertical atmospheric levels. Surface topography is based on Cook et al. (2012) and Bamber and Gomez-Dans (2009). At the lateral and the upper atmospheric boundaries the
model is forced by ERA-Interim reanalysis data every 6 hours, and at the ocean boundaries by prescribed ocean temperatures and sea ice cover. The model atmosphere is initialised Jan, 1st, 1979 with the ERA-Interim reanalysis data, and the snow/firn layer with data generated by the IMAU Firn Densification Model (IMAU-FDM) (Ligtenberg et al., 2011). The precursor version, RACMO2.3p1 includes an older ice mask and surface topography, no upper air nudging, a more severe drifting snow formulation eroding more snow and changes in the formulations of surface melting and precipitation. Further details can be
found in van Wessem et al. (2018) that intercompares versions p1 and p2 more fully.

### 2.1.6 RACMO2.1P

RACMO2.1P is an earlier version of RACMO2 using the ECMWF-IFS physics package CY23r4 which includes no ice cloud super-saturation and utilizes earlier parameterizations for short wave radiation and boundary-layer turbulence as described in Van Wessem et al. (2014). This version of RACMO2.1 includes the polar multi-layer snow routines, as well as the the schemes
for drifting snow and albedo as described for RACMO2.3p2 above. In essence, its polar physics components are identical to those in RACMO2.3p1. Simulations with RACMO2.1P have been performed on a modelling domain matching the CORDEX ANT-44 domain in the interior plus a 16-point extension on each domain side for boundary relaxation of ERA-Interim fields. There is also no nudging within the domain in this version.



| Model | Period | Resolution [km] (degrees) | Nudging | SMB scheme | Topography | Atmos. Levels |
|---|---|---|---|---|---|---|
| COSMO-CLM$^2$ | 1987-2016 | 25 (0.22) | Yes | Yes | GLOBE | 40 |
| HIRHAM5 | 1979-2017 | 50 (0.44); 12.5 (0.11) | No | No | GTOPO | 31 |
| MetUM | 1979-2018 | 50 (0.44) | Reinit. | No | GLOBE | 70 |
| MAR$_{v3.6}$ | 1979-2018 | 35 | Yes | Yes | Bedmap2 | 23 |
| MAR$_{v3.10}$ | 1979-2018 | 35 | Yes | Yes | Bedmap2 | 24 |
| RACMO2.1P$_{v1}$ | 1979-2012 | 50 (0.44) | No | Yes | RAMPv2 | 40 |
| RACMO2.3p2 | 1979-2018 | 27 (0.25) | Yes | Yes | Cook, Bamber | 40 |

**Table 1.** Summary of differences and similarities between the RCMs. Horizontal resolution is given in degrees and (kilometres), while the number of atmospheric levels refers to the vertical resolution. Nudging refers to the level of forcing within the domain, refer to the individual model descriptions for more details.

## 2.2 Model Set-up and Outputs

### 2.2.1 Surface Mass Balance Calculations in RCMs

Two of the models (RACMO and MAR) have subsurface schemes optimised over snow and ice for Antarctica (see references under the model descriptions). Parameterisations are included that account for retention and refreezing of meltwater and also in the case of RACMO2.3p2 wind-driven processes such as erosion at the surface and sublimation of blowing snow. Thus, the definition of the calculation of the SMB changes depending on the complexity of the model. Three models (HIRHAM5, METUM, COSMO-CLM$^2$,) have only simple surface snow physics over ice surfaces in these experiments. The basic SMB we calculate for them in this study is:

$$SMB = precipitation - evaporation - sublimation \qquad (1)$$

For MAR with an optimised subsurface schemes the SMB is calculated from Eq.2:

$$SMB = precipitation - evaporation - sublimation - runoff \qquad (2)$$

This differs slightly in RACMO2.3p2/RACMO2.1P as sublimation and erosion of drifting snow are also included as a mass loss term as in Equation 3:

$$SMB = precipitation - evaporation - sublimation - runoff - SU_{ds} - ER_{ds} \qquad (3)$$

Both models account for refreezing and retention and thus use runoff rather than melt. Due to the low temperatures in Antarctica, most meltwater refreezes and runoff is largely negligible in the current climate (van Wessem et al., 2018; Agosta et al., 2019) so for the remaining models without the multi-layer subsurface schemes SMB is calculated without the runoff component.



### 2.2.2 Nudging and upper atmosphere relaxation

As von Storch et al. (2000) pointed out, nudging whether spectral or with simpler techniques keeps a regional model closer to the driving large-scale fields (GCM or reanalysis) and is thus a valuable technique where a close match to observations or to a driving GCM is required. Within Polar CORDEX, upper-air relaxation and other forms of nudging have been included as a

standard where observational campaigns in large domains require close matches between modelled and observed weather. For example, Arctic cyclone systems and cloud physics in particular appear to be better resolved in models that include nudging (Akperov et al., 2018) and (Sedlar et al., 2011). Similarly, nudging of RCMs run over Antarctica ties their synoptic evolution to these of the driving reanalysis, improving the representation of the interannual variability in SMB to similar levels as in the reanalysis as shown in van de Berg and Medley (2016).

In the experiments presented here, COSMO-CLM$^2$, MAR$_{v3.10}$, and RACMO2.3p2 are nudged by adjusting temperature and wind fields to the global fields with a minimum relaxation time scale of 6 hours. Strongest relaxation is applied at the top of the atmosphere and relaxation decreases gradually for lower levels. Below typically 4 km (ocean) to 6.2 km (4 km land topography) no relaxation is applied. In the case of MAR$_{v3.10}$, the relaxation of the temperature is weaker than the relaxation of the wind between the highest cloud level and the lowest nudging level. This prevents inconsistency between the temperature

inherited from the reanalyses and the humidity and clouds conditioned by the MAR microphysics scheme. Moisture fields are not adjusted by nudging as this would introduce artificial uphill moisture transport. HIRHAM5 and MetUM are not nudged but MetUM is run in a 12-hourly reinitialisation hindcast that keeps the model evolution close to the driving reanalysis.

### 2.2.3 Grids and land-sea-ice masks

All models have been run for a domain covering the entire Antarctic continent but not all of the domains are the same.

HIRHAM5 0.44° and MetUM use the standard CORDEX domain and grid. However, COSMO-CLM$^2$ extends this slightly to cover more ocean around Dronning Maud Land while the HIRHAM5 0.11° simulations and MAR$_{v3.10}$ were run over slightly smaller domains than the CORDEX domain to reduce computational time, though only after running experiments to determine that e.g. precipitation was not affected. RACMO2.3p2 and RACMO2.1 are run for a domain slightly larger than CORDEX but are trimmed back to remove the relaxation zone such that final results are presented on the CORDEX domain. As the model

resolutions are different and each model had its own land-sea mask, the area of Antarctica is not the same in all models, which complicates the SMB results when integrated over the continent. To correct for this areal difference, all the data have been bilinearly regridded to the HIRHAM5 0.11° grid, with the unglaciated land of MAR$_{v3.10}$ included, with a threshold for the ice mask of 50%. This was used to generate a common ice mask for the models in order to calculate the integrated SMB over the ice sheet. In the appendix, figure A1 shows all masks compared to the common mask. Most models had very few grid points

different from the common mask but these were also areas with high precipitation rates and this therefore accounts for some fairly large differences in annual SMB on the native masks.

Modelled SMB is integrated over drainage basins defined as in Shepherd et al. (2018b). The horizontal resolution of the models is not altered and the drainage basin masks are defined by selecting all model grid points that fall within the drainage basin





outlines. In addition to the drainage basins, that are by definition grounded ice, outlines of the ice shelves that the basins drain into are also used. This allows us to partition SMB over grounded ice (GrIS) and ice shelves (IS) as well as over the ice sheet as a whole including ice shelves (TotIS).

## 2.3 Observations

### 2.3.1 AWS observations

Weather observations are used to assess how well RCMs reproduce the meteorological conditions over the Antarctic ice sheet. Although a detailed evaluation of the near-surface model climates is not the purpose of this study, this comparison helps to explain model biases in simulating SMB and especially the coherence between the modelled SMB and the near-surface climate. The original dataset is a compilation of surface pressure, near-surface temperature and wind speed from 307 AWS over the ice sheet used in the MET-READER database (Turner et al., 2004) , but also collected by the BAS, IMAU (van Wessem et al., 2014), and the IGE/IPEV (Amory, 2019). The original data were available at several sampling time steps (sub-hourly, hourly, 3-hourly) and were averaged to obtain daily values. Only daily averages computed from more than 75% of the original data are considered as representative of the entire measurement (UTC) day and are used for comparison. Several stations displayed suspicious measurements (sudden discontinuity in pressure and temperature, temperature values capped to the lower bound of the measurement range during the whole winter season, etc) and these were removed from the dataset. Stations occasionally exhibited wind speeds of 0 m/s for day-long periods, probably as a result of sensor riming. For these cases the daily averages were considered as no data (See Kittel et al. in preparation for details on the full list of AWS and data selection protocol). Although we use a homogenised and quality-controlled dataset for the comparison, observations may still be biased in ways that are hard to quantify due to e.g. burial of stations by snow, battery failures, tilt due to strong winds and other instrument failures that remained undetected, reflecting the difficulties involved in collecting data in the harsh and remote Antarctic environment.

In order to take into account the different ice masks and topographies used by the models, only the stations belonging to the common mask and having a difference in elevation lower than 500 m for each model are retained, leading to the use of 184 AWS (See the supplementary material for locations of AWS used in this study). The modelled surface pressure, near-surface temperature and wind speed, as well as the model elevation, are computed using a four-nearest inverse-distance-weighted method. Finally, the vertical level closest to the surface (10 m or 2 m) of the models is used for all comparisons with the observations, since the measurement height of the observations is not known for every stations.

### 2.3.2 Comparison with 10 m snow temperature observations

Deep snow temperatures in Antarctica are indicative of the annual long-term mean surface air temperature. Here, 64 observations of 10 m snow temperature are used that are collected from a broad range of climatic regions of Antarctica, representing a spatially complete picture of climatological surface temperature (Van Wessem et al., 2014).





### 2.3.3 Observed SMB

Observations of SMB are sparse over the wide Antarctic continent, and have been obtained from diverse measurement techniques such as stake measurements, ice core, or radar stratigraphy. For the purpose of our evaluation, we use the SAMBA dataset from Favier et al. (2013), that has been completed with Wang et al. (2016) and yearly values of the ice cores from Thomas et al. (2017) to obtain an original dataset of around 7136 observations for various time periods and for a wide range of locations scattered across the Antarctic ice sheet. The radar measurements published by Medley et al. (2014) are not used in this study as the spatial variability is very high and difficult to smooth appropriately for all model grids.

To evaluate the models, we selected observations of SMB belonging to the common ice mask and for which the measurement period began after 1950 to 2018. These conditions reduced the total number of observations used in the comparison to 3671. Observations between 1950 and 1987, or 2015 and 2018 not fully included in the common modelling period (ie., 1987 to 2015), were used for evaluation only if they covered more than 5 years. These 1849 SMB observations are compared to modelled values averaged over the common modelling period in order to compute a climatological mean while we averaged modelled SMB values over the exactly same period for the observations between 1987 to 2015 (1822 observations).

Since the models have different resolutions and grids, we do not directly compare the modelled SMB values to the observations. As in Kittel et al. (2018) and Agosta et al. (2019), we compute modelled and observed SMB values in 3 steps. Firstly, the original resolution modelled SMB values were interpolated, as for AWS observations, to the observation location using a four-nearest inverse-distance-weighted method. Secondly, all the interpolated SMB values contained in the same grid cell from the common ice mask were averaged as well as the observations for finally creating 923 comparison pairs. This leads to a fair comparison for each model that takes into account the benefit of using a higher resolution for a specific model and removing the very high spatial variability of the observations that cannot be reproduced by the models.

Like the meteorological data, SMB observations are subject to measurement biases notably due to post-depositional redistribution of snow and the related formation of sastrugi that can considerably complicate the interpretation of measurements at the very local scale (Andersen et al., 2006). SMB observations should therefore be considered as a best estimate of accumulation rather than an absolute value. As SMB observations are not evenly distributed over the ice sheet, the comparison statistics may be artificially influenced by over- and/or under-sampled regions.

## 3 Results

We first focus on how the RCMs characterise the surface climate over the ice sheet before turning to assessing the SMB and taking note of the differences in precipitation distribution.

### 3.1 Temperature, Surface Pressure and Wind Speed from Models and Observations

Weather observations in Antarctica extend further back in time and there is generally better coverage than for direct SMB measurements. In figure 1 we show Taylor diagrams for pressure, temperature and wind velocities. Taylor diagrams offer an




efficient way to assess model skill by comparing the Pearson correlation coefficient, the root mean square error (RMSE) and the standard deviation of the modelled output with the observed values. In this case modelled values closest to the dashed line have a more correct representation of the standard deviation and the closer to the black reference star the closer the model correlates to the observations values. We also list the bias below the diagrams.

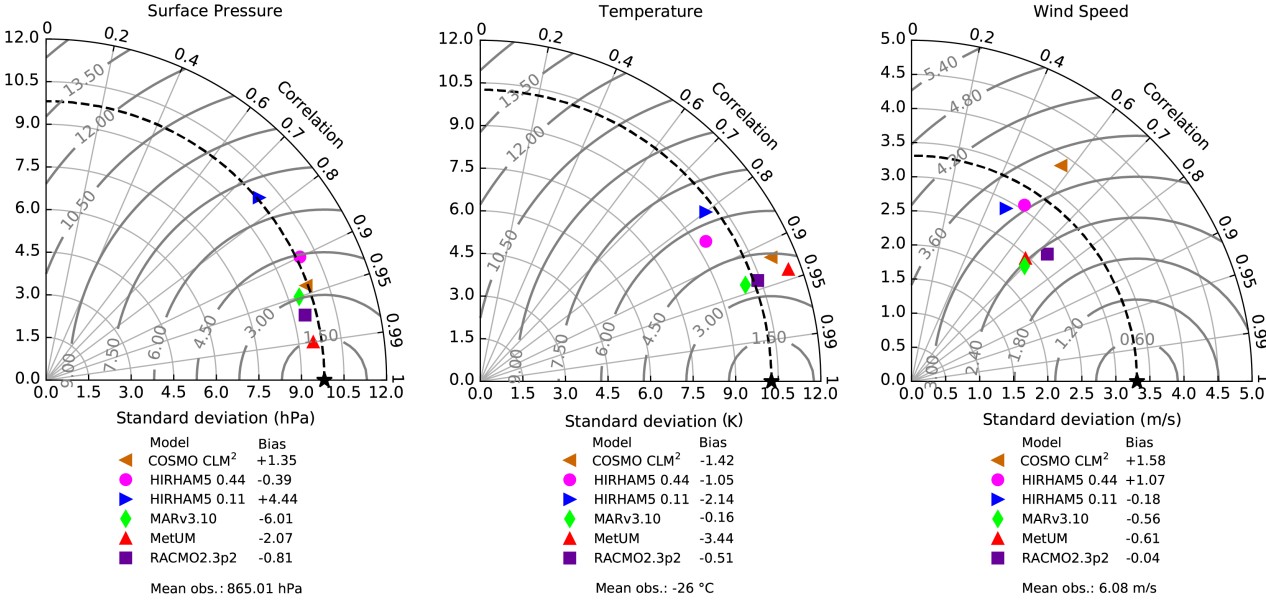

**Figure 1.** Taylor diagrams showing model performance compared to daily observations of surface pressure, near-surface temperature and observed wind speeds as well as the bias statistics for each model

Figure 1 analysis shows that depending on variable the models perform reasonably well with some variation. With respect to surface pressure, the majority of models are similarly skillful with the exception of HIRHAM5 0.11°, though the model is still close to the pattern of the standard deviation. The other models have quite a high degree of nudging including upper atmosphere pressure fields within the domain so it is not so surprising to see the good performance here. The high resolution (0.11°) version of HIRHAM5 has many more grid cells than the low resolution (0.44°) version, which in effect increases the

degrees of freedom associated with the high-resolution run and allows a higher divergence due to internal variability. MetUM is not nudged by surface relaxation but is run in daily reinitialisation mode and while this probably also helps to keep surface pressure close to observed it is also likely that the large number of atmospheric levels in MetUM also improves modelled surface pressures. The near-surface temperatures in figure 1 show that although overall the models perform well (Pearson correlation of 0.85 and higher) on average all the models are too cold and only MAR$_{v3.10}$ and RACMO2.3p2 have a bias of less than 1

degree (respectively -0.16 and -0.51 K), with MetUM having the highest bias (-3.44 K). As with the surface pressure analysis, the HIRHAM5 high resolution simulations have a lower correlation coefficient and this may well be again the consequence of the unnudged simulations. However, biases in cloud cover and long-wave radiation reaching the surface are probably the main





explanation for divergence from observations and should be investigated for all RCMs run for Antarctica. Furthermore, the lack of detailed subsurface snow pack schemes also likely has an impact on the temperature bias in HIRHAM5 and MetUM (see also figure 2).

It is clear in figure 1 that all of the models perform less well for wind speeds than for temperature or pressure. This is likely
in part due to large uncertainties in the observations and the effects of resolution but may also relate to differences in turbulent schemes between the models and in particular the extremely stable boundary layer over most of Antarctica that is hard to represent in models particularly at lower resolutions (Zentek and Heinemann). The models can be roughly divided into two groups: MAR$_{v3.10}$, MetUM and RACMO2.3p2 on the one hand, and the two HIRHAM5 runs and COSMO-CLM$^2$ on the other hand. In the case of COSMO-CLM$^2$ wind speeds are output at 20 m and then interpolated to 10 m using Monin-Obukhov theory
(Souverijns et al., 2019), which may not be sufficient to properly represent near-surface winds and associated interactions. The HIRHAM5 results may again be biased due to the lack of nudging within the domain. However it is worth pointing out that HIRHAM5 correctly represents the mean spatial variability (both runs are the closest to the dashed line indicating the standard deviation) and in the case of the high resolution run the mean observed wind speed, but not the daily evolution of the wind, resulting from the biases in the atmospheric circulation.

## 3.2   Comparison with 10 m snow temperature observations

Figure 2 shows the modelled surface temperature of the RCMs as a function of 64 measurements of temperature at 10 m depth as also used by Van Wessem et al. (2014). The majority of the AIS has negligible snow melt and in these regions the 10 m snow temperature is representative of the annual long-term average surface temperature. This comparison therefore is a robust assessment of the climatological surface signal calculated by the models, also because the observations are evenly
scattered across the continent and represent most climatic regions. All models capture the wide range of surface temperatures from $\approx$ 218 K to 260 K. HIRHAM5 0.44°consistently underestimates temperature for most locations, a bias that closely resembles RACMO2.1 in Van Wessem et al. (2014)) and which the authors concluded was predominantly related to biases in the downwelling long-wave radiation. The other models overestimate temperature in the colder, and therefore higher elevation locations, while underestimating temperature for the coastal regions. For the colder regions below $\approx$ 240 K, these biases are
mostly related to discrepancies in cloud cover and most likely snowfall, affecting downwelling longwave and surface albedo. Some of the Antarctic models have been tuned to improve the dry and cold biases in the interior that were persistent in earlier model versions (see RACMO2.1P; Van Wessem et al. (2014); van Wessem et al. (2018)), but now overestimate temperature slightly instead. While subsequent model updates have led to significant improvements in simulated SMB, this has come at the expense of surface temperature due to excessive increases in downwelling radiative fluxes that accompany increases in
snowfall.

For the warmer coastal regions most models have a cold bias. This bias is likely related to the effects of surface meltwater percolating into the firn and refreezing within, raising deeper snow temperature, implying modelled surface temperature is not a good metric for observed 10 m snow temperatures in the percolation zone. A more accurate comparison would therefore be to directly compare 10 m snow temperatures from the models with the observations. However, not all models calculate snow



temperatures, and given the scope of this manuscript, we only intercompare the surface temperature. Here, Figure 2 illustrates a consistent intermodel scatter, with mainly the models that do not include a sophisticated snow model outside of this range. This clearly points to a significant potential source of improvements for modelled SMB in the future.

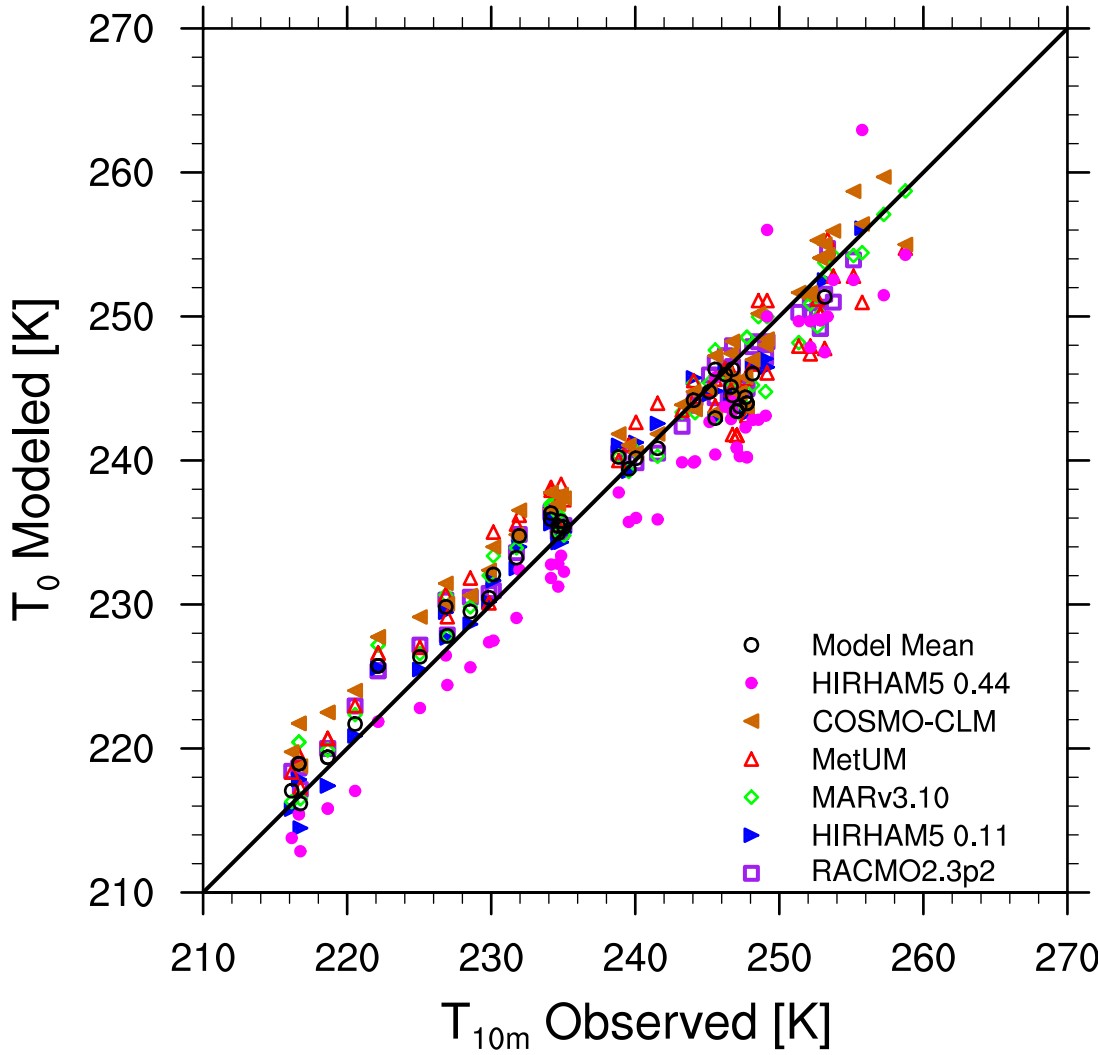

**Figure 2.** Modelled surface temperature as a function of observed 10 m-snow temperature (Van Wessem et al., 2014). Observations that are not fully located on the model ice-mask are excluded.

### 3.3 Comparison with Observed SMB

5    Evaluating SMB is hindered by poor observations across the cryosphere, particularly in Antarctica where remoteness and extreme weather conditions add to the challenge of observing SMB. In our analysis, in spite of using a large dataset of observations, shows that there are large areas that are significantly undersampled ( See for example, Figure 4). We therefore





separate the comparison of modelled and observed SMB into elevation bins in Figure 3 in order to make the results clearer. It is important to mention that for the scatter plots by elevation class, if an observation or one of the models had a negative value, the observation and modelled values were removed from the analysis using logarithmic values (hereafter, rlog is the correlation computed on the logarithm of SMB values), but are retained in the analysis using the original populations. Statistics for the

SMB comparison are given in Table 2 but to show the large scatter in the observations and the models clearly we show all models plotted against observations in Figure 4 and plotted against each model individually in the appendix (Figures A2 , A3, A4 A5, A6,A7).

Apart from COSMO-CLM$^2$ and HIRHAM5 0.11°the RCMs show similar Root Mean Square Error (RMSE) and r$^2$ values when compared over the full dataset but broken down by elevation class or locally by regions shows a more complex story.

In general all models underestimate SMB over the ice shelves and at the low elevation coastal regions of Antarctica (see also statistics in Table 2 a. and b. and Figure 3). This is particularly true for COSMO-CLM$^2$ over both locations and HIRHAM5 0.11°over the ice shelves leading to higher RMSE compared to the others RCMs, but while all the RCMs underestimate SMB over the Ross ice shelf, MAR$_{v3.10}$ overestimates it, probably related to a poorer representation of the surface climate by the model over this ice shelf. The blowing snow module included in RACMO2.3p2 may explain the good results between 0 and

1200 and 1200 and 2200 m (Table 2 b and c), where it outperforms the other models. A previous comparison shows higher sublimation in RACMO2.3p2 than in MAR$_{v3.10}$ (Agosta et al., 2019) notably at the elevations where katabatic winds are strong due to the slope of the ice sheet and where the atmosphere is not too cold enabling large amounts of sublimation from blowing snow particles. COSMO-CLM$^2$ and HIRHAM5 0.44°have the highest RMSE while HIRHAM5.011, MAR$_{v3.10}$ and MetUM have similar statistics at this elevation. For the highest elevations (above 2200 m), all the model RMSE scores are relatively

low and similar to each other except HIRHAM5 0.44°(and to a lesser extent MAR$_{v3.10}$) between 2800 m and 3400 m (Table 2 e). However, the less extensively evaluated models (HIRHAM5 at both resolutions and MetUM) are both too dry over the high plateau of the Antarctic ice sheet. If we look at all the elevation ranges, no model is systematically in the top 3 for every range but, RACMO2.3p2 has the best comparison with all the observations, closely followed by MetUM with MAR$_{v3.10}$ and HIRHAM5 0.44°performing almost equally. It is worth emphasising though that as Fig. 4 shows, the observations in this

elevation class are also very noisy and the poor relative performance of the models may result as much from unrepresentative and sparse repeat observations as it does from missing or poorly resolved processes in models. Analysis of these results not only indicates areas where models need to be improved but also areas where more observations to test models are desirable, notably between 1200 and 2200 where the mean biases of the models used in this study display large discrepancies (Table 2 c). It is also likely that there are compensating errors within each model that hide the true performance. For example, the mean

bias between the two different HIRHAM runs has opposite signs in the 1200 - 2800 m range, likely reflecting the difference in model resolution. Orographic precipitation is very sensitive to slope effects and the presence of steep topography is very different between the two resolutions, affecting where precipitation falls across the continent. The wide scatter in modelled SMB in the 2200-2800 m elevation range is therefore also likely to reflect in part the resolution of the different models and how well they capture orography and the consequent precipitation. Studies by for example Hermann et al. (2018) and Schmidt

et al. (2017) show that hydrostatic models like HIRHAM5 and RACMO2.3 typically overestimate precipitation on the upslope



and have a dry bias downwind of initial steep topography, this pattern seems to some extent to be repeated here in Figure 3 and 4. Comparing the observations used in this analysis with the RCM ensemble modelled SMB in Figure 6 also highlights that the largest differences between models and compared with the ensemble mean are mostly in regions with very few or no observations, also regions where precipitation is typically high, making it difficult to assess the ability of models to truly

5    simulate the SMB of Antarctica. Our analysis therefore also helps to identify areas where increased observations will be most useful to help assess and improve model processes.

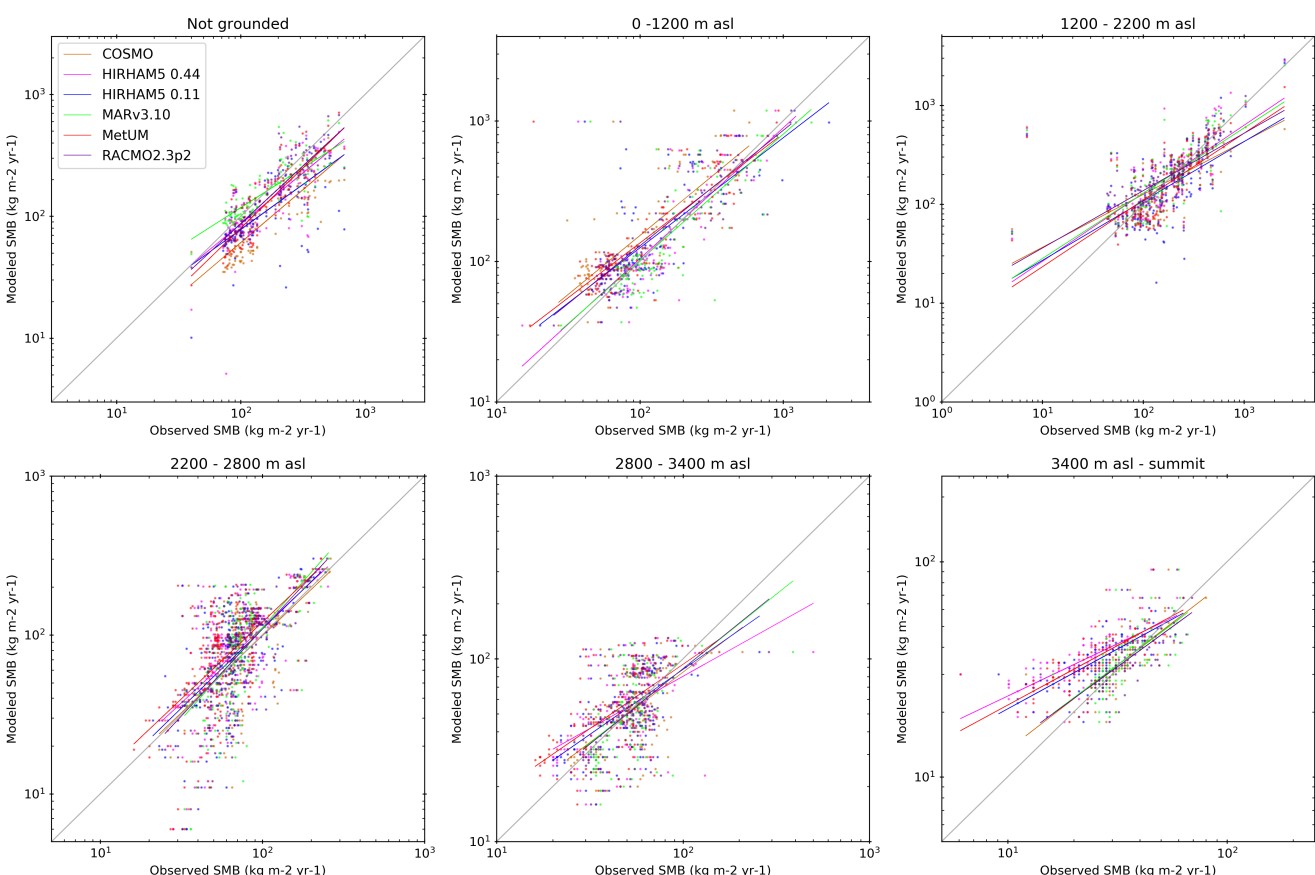

**Figure 3.** Comparison between modelled SMB and observed SMB in a gridded dataset. Trend lines and points are plotted for each model in a different colour, note different x and y axes for different elevation bins.





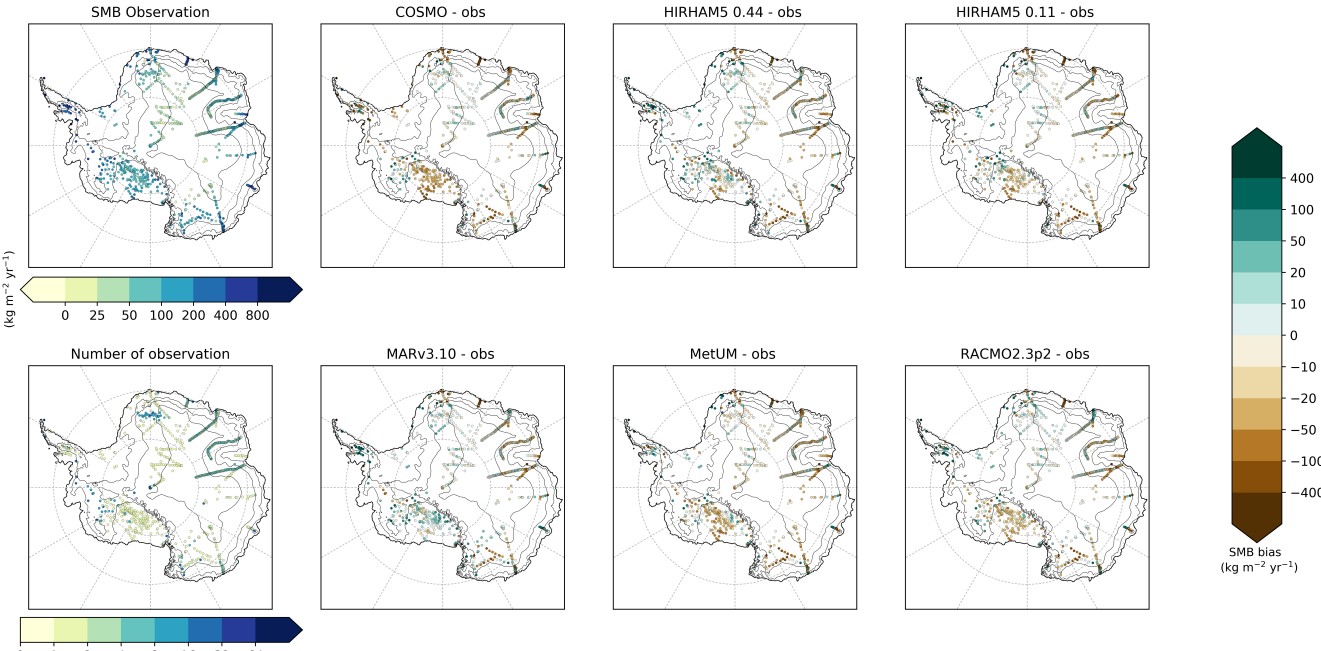

**Figure 4.** a) Observed SMB values. b) Number of observations per pixel. Difference between observed and modelled SMB from COSMO-CLM$^2$ (c), HIRHAM5 0.44°(d), HIRHAM5 0.11°(e), MAR$_{v3.10}$ (f), MetUM (g), RACMO2.3p2 (h).

### 3.4 Assessing the Surface Mass Balance of Antarctica

Bearing in mind the results presented in the preceding section evaluating the RCMs, we here show the range of best estimates for Antarctic SMB based on RCMs. Figure 5 shows the modelled Specific Surface Mass Balance (SSMB) and integrated modelled SMB for the AIS of the nine climate simulations for 19 drainage basins as defined in Shepherd et al. (2018b). The

SSMB is shown by the colour shading and is defined as the integrated SMB (in mm per year), divided by the area of the basin. The SMB is the total integrated SMB of the basin in units of Gigatonnes (shown by the numbers in a box in each basin). Figure 5 illustrates that all models simulate a comparable SSMB for EAIS, with values between 100 and 400 mm per year. Due to the moist coastal climates over the ice shelves, SSMB values here reach values as high as 1000 mm per year. The main intermodel differences are found over the WAIS and the Antarctic peninsula (AP) and are most likely related

to differences in horizontal resolution and therefore orographic precipitation. The higher resolution models (RACMO2.3p2, HIRHAM5 0.11°and MAR$_{v3.10}$), generate the highest SSMB values over the AP and WAIS basins, up to 2000 mm per year. The other models have considerably lower SSMB, especially over the adjacent ice shelves. The exception is COSMO-CLM$^2$ which is drier than all other models in all basins with the exception of the Queen Mary Land basin in the EAIS, where HIRHAM5 0.11°is slightly drier, and the interior of the EAIS where MAR$_{v3.10}$ is slightly drier. The two areas with the largest ensemble

mean deviation are the western peninsula basin but also the interior of the EAIS bordering the Transantarctic mountains and



|  | a. Shelves (N=112, L=112) Mean obs: 199±132 | | | | b. 0 - 1200 m (N=130, L=128) Mean obs: 223±224 | | | |
|---|---|---|---|---|---|---|---|---|
|  | MB | RMSE | r | rlog | MB | RMSE | r | rlog |
| COSMO-CLM[2] | -85 | 125 | 0.75 | 0.84 | -79 | 174 | 0.73 | 0.81 |
| HIRHAM5 0.44° | -37 | 89 | 0.79 | 0.75 | -22 | 143 | 0.77 | 0.82 |
| HIRHAM5 0.11° | -59 | 122 | 0.60 | 0.67 | -26 | 194 | 0.68 | 0.76 |
| MAR$_{v3.10}$ | -12 | 98 | 0.69 | 0.79 | -5 | 159 | 0.74 | 0.79 |
| MetUM | -32 | 83 | 0.82 | 0.82 | -41 | 142 | 0.79 | 0.84 |
| RACMO2.3p2 | -25 | 90 | 0.78 | 0.78 | -29 | 147 | 0.78 | 0.87 |

|  | c. 1200 - 2200 m (N=158, L=154) Mean obs: 225±240 | | | | d. 2200 - 2800 m (N=259, L=258) Mean obs: 89±55 | | | |
|---|---|---|---|---|---|---|---|---|
|  | MB | RMSE | r | rlog | MB | RMSE | r | rlog |
| COSMO-CLM[2] | -22 | 187 | 0.63 | 0.75 | -9 | 42 | 0.67 | 0.61 |
| HIRHAM5 0.44° | 33 | 143 | 0.89 | 0.78 | -18 | 45 | 0.65 | 0.59 |
| HIRHAM5 0.11° | -19 | 119 | 0.89 | 0.68 | -16 | 46 | 0.64 | 0.56 |
| MAR$_{v3.10}$ | 20 | 115 | 0.90 | 0.79 | -14 | 42 | 0.70 | 0.63 |
| MetUM | -16 | 119 | 0.87 | 0.80 | -22 | 46 | 0.68 | 0.63 |
| RACMO2.3p2 | 12 | 95 | 0.94 | 0.77 | -13 | 41 | 0.68 | 0.66 |

|  | e. 2800 - 3400 m (N=161, L=161) Mean obs: 58±27 | | | | f. 3400 m - top (N=103, L=103) Mean obs: 36±12 | | | |
|---|---|---|---|---|---|---|---|---|
|  | MB | RMSE | r | rlog | MB | RMSE | r | rlog |
| COSMO-CLM[2] | -1 | 23 | 0.59 | 0.61 | -1 | 9 | 0.70 | 0.72 |
| HIRHAM5 0.44° | -6 | 40 | 0.35 | 0.53 | -12 | 15 | 0.72 | 0.72 |
| HIRHAM5 0.11° | -5 | 26 | 0.55 | 0.62 | -9 | 12 | 0.72 | 0.72 |
| MAR$_{v3.10}$ | -2 | 32 | 0.41 | 0.54 | -1 | 9 | 0.67 | 0.69 |
| MetUM | -10 | 25 | 0.59 | 0.61 | -10 | 14 | 0.73 | 0.73 |
| RACMO2.3p2 | -2 | 27 | 0.46 | 0.56 | 0 | 9 | 0.70 | 0.72 |

|  | g. All (N=923, L=916) Mean obs: 133±160 | | | |
|---|---|---|---|---|
|  | MB | RMSE | r | rlog |
| COSMO-CLM[2] | -28 | 113 | 0.74 | 0.79 |
| HIRHAM5 0.44° | -9 | 91 | 0.85 | 0.82 |
| HIRHAM5 0.11° | -20 | 101 | 0.81 | 0.79 |
| MAR$_{v3.10}$ | -3 | 88 | 0.85 | 0.83 |
| MetUM | -22 | 82 | 0.87 | 0.84 |
| RACMO2.3p2 | -9 | 79 | 0.88 | 0.85 |

**Table 2.** Comparison of the modelled SMB to the SMB observations over the ice shelves (A), by elevation bins (B-F) and over the whole Antarctic ice sheet (G). Unit of Mean Biases (MB), Root Mean Square error (RMSE), and Mean of the observation is kg/m²yr. N denotes the number of comparison used for each bin while L represents the number of comparison used the log distribution (See the supplementary materials for more details)





including the South Pole. In this region the MAR$_{v3.10}$ model has the highest SMB (196 Gt) but MetUM has the lowest (77 Gt). Figure 5 also shows some of the striking features in the pattern of SMB present in all the models where the magnitude differs, for example, all models have a steep gradient in the SMB over the Antarctic peninsula, but this is much more pronounced in HIRHAM5 0.11°than in HIRHAM5 0.44°, demonstrating the importance of resolution in this region. MetUM and COSMO-

5 CLM$^2$ also show the same pattern but with considerably lower absolute values particularly on the western side, than the other models. These differences in modelled SMB on the basin scale may have considerable impact on dynamic ice sheet models used to determine the evolution of the Antarctic ice sheet and are consequently important to take into account when selecting SMB to force ice dynamics models. Looking at the total surface mass budget including ice shelves for the period 1980 to 2010 (numbers in the caption and summarised in Table 3) generated by the models, the HIRHAM5 0.44°simulation is the wettest

model (2752 Gt per year; 2328 Gt excluding ice shelves), while COSMO-CLM$^2$ is clearly the driest (2031 Gt per year; 1751 Gt excluding ice shelves). The other simulations are all closer to each other and are within an SMB range of $\pm$ 200 GT per year, while the two dedicated polar models (RACMO2.3p2 and MAR$_{v3.10}$) have only a small difference of 83 Gt per year on average, corresponding to around 3% of the total budget. These two models have been evaluated and optimised for Antarctica the most intensely of all the models (van Wessem et al., 2018; Agosta et al., 2019). We also include MAR3.6 and RACMO2.1

in this figure to give context to earlier studies. The two closest models overall are in fact HIRHAM5 0.11°and MAR$_{v3.10}$ which differ by only 26 Gt overall with much of the difference accounted for by the SMB of the ice shelves.




**Figure 5.** Integrated SMB and specific SMB (SSMB) for the 9 models included in this study: RACMO2.1, RACMO2.3p2, RACMO2.3p1, HIRHAM5 50 km, HIRHAM5 12.5 km, MAR$_{v3.10}$, MAR$_{v3.6}$, MetUM and COSMO-CLM$^2$, as well as the ensemble mean and standard deviation. Colours denote the SSMB in mm w.e. per year for all grounded ice sheet basins as well as the ice shelves these drain into, defined in Shepherd et al. (2019). The numbers included in the basins denote the basin integrated SMB in Gt year$^{-1}$ for the grounded ice sheet for the period 1980 to 2010 with the exception of COSMO-CLM$^2$ where the time series starts in 1987. Finally, the total integrated number for the grounded ice sheet including ice shelves is shown in the figure label.





| Model | GIS (Gt y$^{-1}$) | IS (Gt y$^{-1}$) | ToTIS (Gt y$^{-1}$) | Area (km$^{-2}$) |
|---|---|---|---|---|
| RACMO2.1p | 1933 | 471 | 2395 | 13.85 |
| RACMO2.3p2 | 2133 | 430 | 2556 | 13.85 |
| RACMO2.3p1 | 2035 | 438 | 2466 | 13.85 |
| MAR$_{v3.10}$ | 2227 | 413 | 2633 | 13.92 |
| MAR$_{v3.10}$ | 2158 | 396 | 2547 | 13.92 |
| HIRHAM-5 0.44° | 2328 | 438 | 2757 | 13.87 |
| HIRHAM-5 0.11° | 2235 | 434 | 2659 | 13.83 |
| MetUM | 1884 | 452 | 2328 | 13.82 |
| COSMO-CLM$^2$ | 1751 | 288 | 2031 | 13.84 |
| Model Mean | 2076 | 418 | 2486 | 13.86 |
| Model $/sigma$ | 306 | 77 | 266 | 0.085 |

**Table 3.** Integrated mean annual SMB for the six models used in this study, for the period 1980 to 2010 except for COSMO-CLM$^2$ where the period was 1987 to 2010. Three older model versions, ensemble mean and standard deviation as shown in Figure 5. All calculations done on the original grid of the individual models using a common set of drainage basins and ice mask defined by IMBIE2 Shepherd et al. (2018b). GIS denotes grounded ice sheet, IS denotes ice shelves and ToTIS denotes the full Antarctic ice sheet including ice shelves.

As the basin scale SMB values differ quite substantially between models, in Figure 6 we plot the mean annual SMB from the ensemble mean and the anomaly to that for each of the different models. The ensemble mean is calculated on a common grid but the model anomalies are calculated from it on their own grids which shows better the effects of the different resolutions on the SMB. The figure shows quite substantial agreement between models over large areas of Antarctica but also some considerable

local variability. Features such as the Transantarctic mountains and the rugged coastal topography in west Antarctica both of which substantially influence local weather patterns are for example much clearer in the higher resolution runs. However, the ensemble mean can also hide large disagreements between the models. For example there is an interesting asymmetry in the model results for the region of the Queen Maud Mountains and Queen Elizabeth ranges of the Transantarctic mountains.The MAR model and to a lesser extent the HIRHAM5 0.44°model show rather different patterns in SMB compared to the other

models, with higher SMB south of the Range and lower than ensemble mean values north of the range. The other models show the reverse with lower than the mean south of the range and higher to the north. A similar, but less clear pattern is also seen along the Ross and Amundsen Sea coastal sectors. The coastal margin of the whole continent in general shows a blotchy pattern in the SMB anomaly plots that reflects rugged topography. In these regions the resolution of the model determines the location of orographic precipitation. Analysis of similar SMB simulations in Greenland with the HIRHAM, MAR and RACMO models

(Hermann et al., 2018; Schmidt et al., 2017) suggests that in these types of locations HIRHAM and RACMO overestimate precipitation at lower elevations in steep terrain, whereas MAR tends to have a wet bias at a slightly higher elevation where the other two models are drier. Agosta et al. (2019) related this different pattern of biases in MAR to the advection of precipitation in the models prognostic precipitation scheme. Understanding these biases is crucial to understanding and interpreting modelled





SMB and comparing Figure 6 with Figure 4 it is clear that the locations where there is the highest disagreement between models are also the regions with the poorest systematic observational coverage of SMB, especially in coastal regions and in west Antarctica.

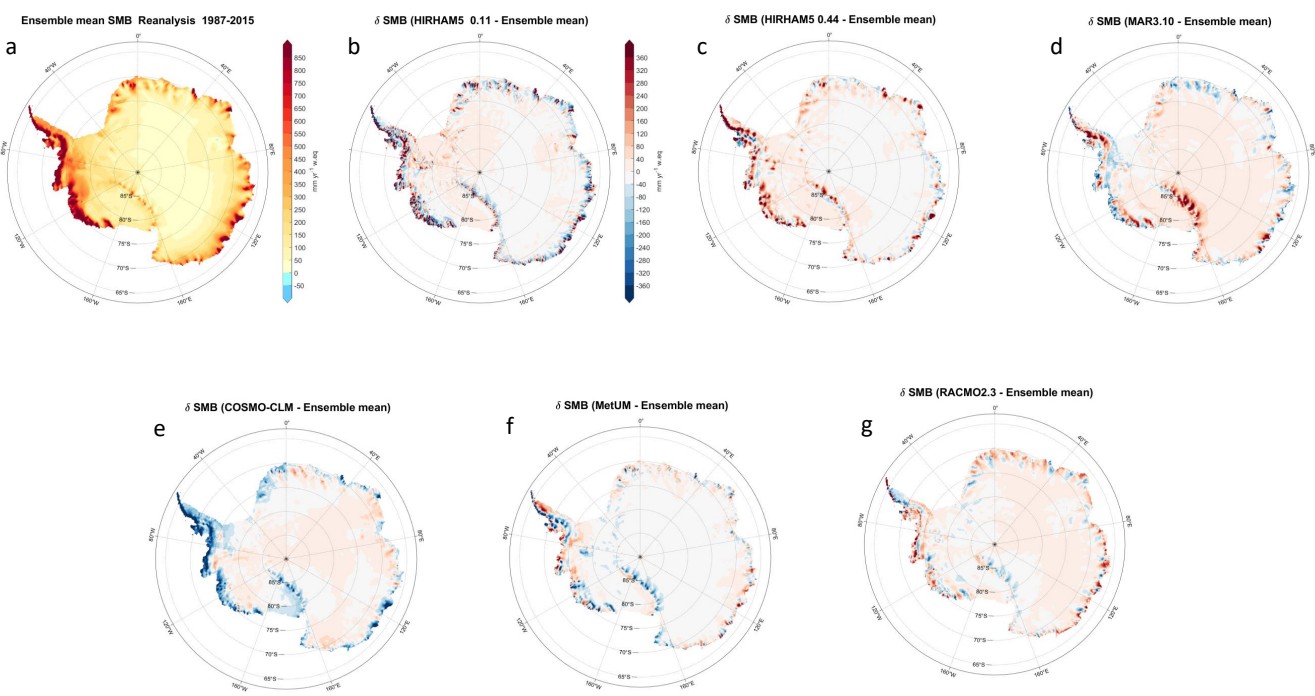

**Figure 6.** Sub-figure **a** shows the SMB ensemble mean for the common period, on the common mask. Sub-figure **b-g** show the difference between each model and the ensemble mean.

SMB varies not only spatially but also temporally and average annual SMB values hide very large interannual variability in SMB as depicted in Figure 7. The spread in the range of estimates of SMB is however, consistent from year to year. The integrated continental SMB calculated over the common mask has a spread of more that 550 GT between the highest and lowest estimate on average (see also Table 4) but all the models show similar annual and decadal scale variability.This implies that the driving model, in this case ERA-Interim, is the most important source of SMB variability but that the individual models are important when considering both the absolute number and the local spatial variability.



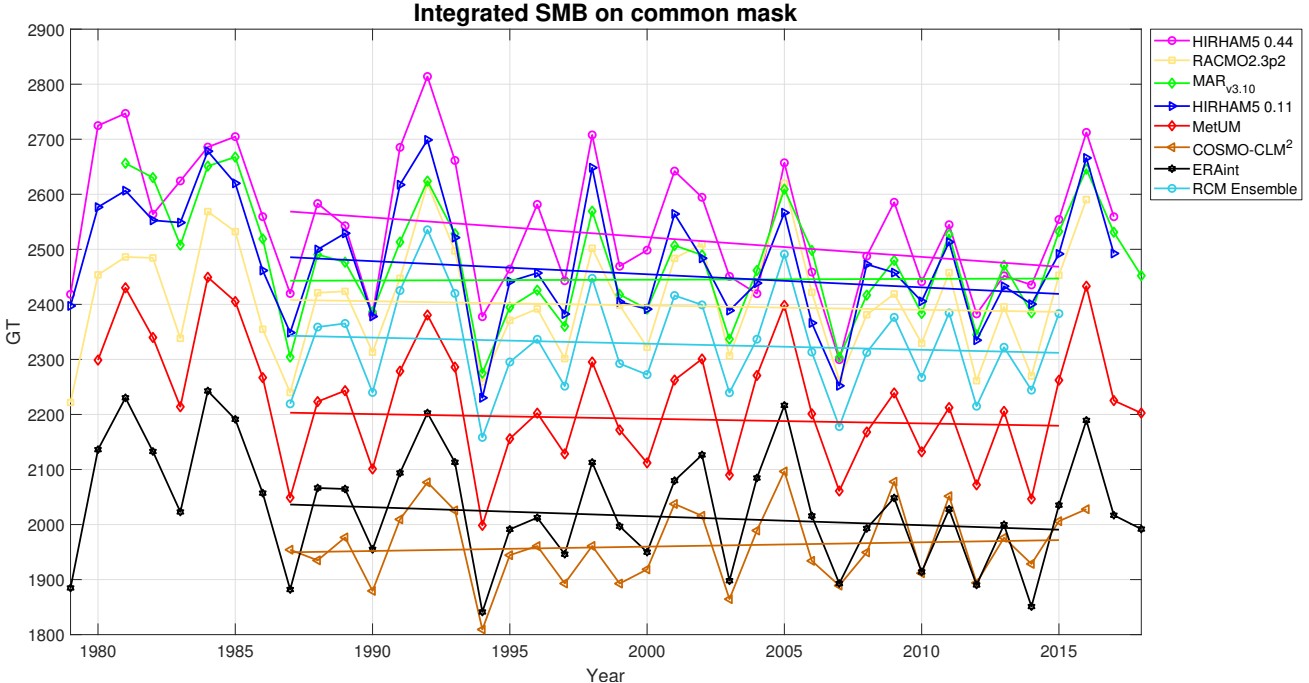

**Figure 7.** Annually resolved SMB integrated over the common ice mask for the different RCMs, in the period 1979-2018. All RCMs are driven by ERA-Interim. The ensemble is a mean calculated from all 6 RCM's in the period 1987-2015 where there is data from all the models. All the trend lines are calculated for the period 1987-2015.

Over the period 1987 to 2015 for which data are available for all the models, the mean annual SMB and components are calculated across the continent as given in Table 4 below, note that this is calculated over a common ice mask and a common simulation period and results are therefore slightly different to those already published for different models or shown in Figure 5 or Table 3. In this time series HIRHAM5 0.11° and MAR$_{v3.10}$ are again the closest two models to each other. RACMO2.3p2
5   is closest to the ensemble mean but COSMO-CLM$^2$ is closest to the driving ERA-Interim modelled values. The trend lines are very sensitive to starting and ending years and in some cases change sign if a longer period is chosen, but as we have only a short common period we have chosen to calculate the trend over the common period. For this chosen period COSMO-CLM$^2$, and MAR$_{v3.10}$ show a slightly increasing trend in SMB, whereas the rest show a slightly declining trend in SMB although the trend in RACMO2.3 and MetUM are almost flat. The ERA-Interim trend over the period declines slightly more than the
10   MetUM trend, which is otherwise extremely close. The different trends from the models and in particular the sensitivity to different start and end points does not give us confidence to ascribe a statistically significant trend to Antarctic SMB over the whole continent. We note though that all models show a declining trend in the 1990s and early 2000s but with a recent increase in SMB since 2014. The early part of the record appears to have higher variability but this may be related to changes in data assimilation in the driving reanalysis (Dee et al., 2011).





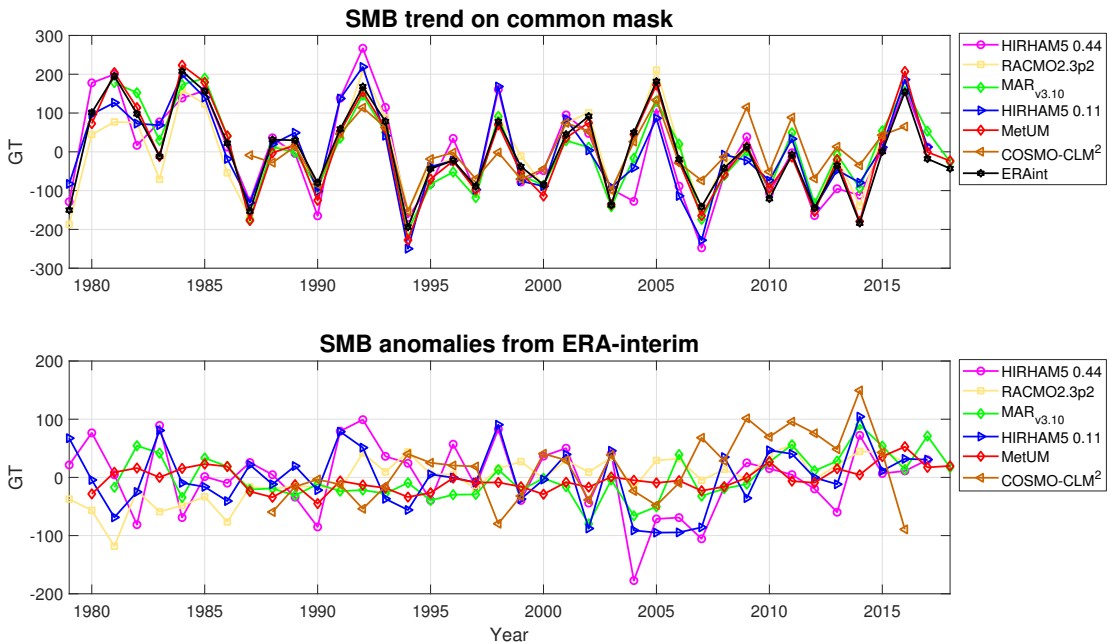

**Figure 8.** The upper figure shows the variability in surface mass balance over the common mask for each of the different RCMs, in the period 1979-2018. Calculated by subtracting the respective model mean from each RCM's SMB time series. The bottom figure displays how modelled SMB from each RCM deviates from the ERA-interim SMB.

Figure 8 emphasises the large variability in SMB on an annual to decadal scale by plotting the variation from the mean for each model and the variation from ERA-Interim for each model. Both HIRHAM5 0.11°and 0.44°shows the highest values of variability, probably due to the unconstrained nature of the runs, but in different years different models show higher variability than the others. The lower panel in Figure 8 shows that MetUM is by far the closest to the driving model with much less

5  variability than the others, HIRHAM5 again shows the highest difference compared to the driving model but from year to year the model showing maximum difference varies and there appears to be no systematic pattern as to whether or not modelled SMB is higher or lower than the ERA-Interim reanalysis when quanitifed on the common mask and over the whole of Antarctica.





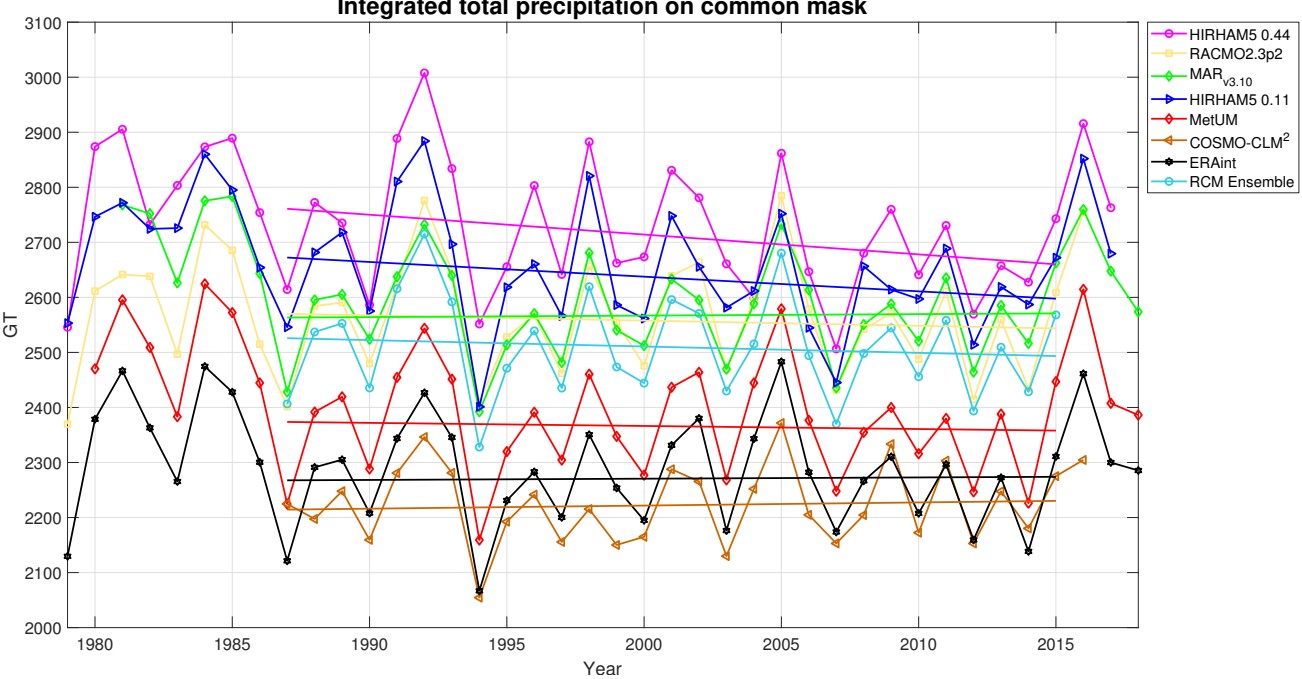

**Figure 9.** Annually resolved precipitation integrated over the common mask for the different RCMs, in the period 1979-2018. All RCMs use ERA-Interim. The ensemble is a mean calculated from all 5 RCMs in the period 1987-2015 where all models have data.

Since SMB is made up of accumulation and ablation components, and in Antarctica precipitation is the dominant term, Figure 9 shows the precipitation component only over the common mask for the different models and ERA-Interim. There is a very similar pattern to that in Figure 7 but compensating effects from melt and sublimation explain the bigger offset between HIRHAM5 0.11°and MAR$_{v3.10}$, which in turn is closer to RACMO2.3 in terms of precipitation. The mean values

for the SMB components of precipitation, evaporation and sublimation as well as SMB for the common period 1987-2015 over the common ice mask are also displayed in Table 4. These values confirm that the very much higher precipitation in both HIRHAM5 runs compared to the other models is to some extent compensated for by higher values of sublimation. The higher sublimation rates in the RACMO2.3 model results from the drifting snow scheme, as the RACMO2.3 model includes sublimation from ventilated snow, while MAR$_{v3.10}$ only includes wind erosion of surface snow. This also bring the SMB

further away from that of MAR$_{v3.10}$, even though the total precipitation is rather similar in the two models, even if differently distributed. MetUM, which performs similarly to RACMO2.3 when compared with SMB observations has lower precipitation and higher sublimation rates than RACMO2.3 however, suggesting that ventilation of drifting snow alone does not explain the higher sublimation rates. MAR$_{v3.10}$ has the lowest of all sublimation rates and COSMO-CLM$^2$ the highest. In fact our results suggest that the dry bias in COSMO-CLM$^2$ is a result in part of the lower precipitation values, which are very close to those of

the driving ERA-Interim model, but also a consequence of the much higher sublimation values. The RACMO2.3 model is still



closest to the ensemble mean annual precipitation but the MAR$_{v3.10}$ model mean values are only 10 Gt different to RACMO2.3 and in some years shown in fig. 9 it is actually even closer.

| Model | SMB (Gt yr$^{-1}$) | Precipitation (Gt yr$^{-1}$) | Sublimation (Gt yr$^{-1}$) |
|---|---|---|---|
| HIRHAM5(0.44°, 0.11°) | 2519 ± 118, 2452 ± 107 | 2715 ± 117, 2635 ± 107 | 192 ± 12, 183 ± 10 |
| MAR$_{v3.10}$ | 2445 ± 91 | 2567 ± 87 | 122 ± 11 |
| RACMO2.3p2 | 2397 ± 101 | 2557 ± 100 | 158 ± 7 |
| MetUM | 2191 ± 101 | 2366 ± 100 | 175 ± 9 |
| COSMO-CLM$^2$ | 1961 ± 70 | 2222 ± 72 | 262 ± 10 |
| ERA-Interim | 2016 ± 99 | 2271 ± 95 | 255 ± 18 |
| Ensemble mean | 2329 ± 94 | 2498 ± 93 | 194 ± 9 |

**Table 4.** Mean annual SMB and components on common mask for each model averaged over the 1987-2015 period where all the models overlap, standard deviations are also show.

## 4 Discussion

### 4.1 The Surface Mass Budget of Antarctica

The range of models in this intercomparison study allows us to estimate not only the likely range of SMB over Antarctica, but also to identify sources of disagreement and bias within and between models. Accounting for differences in ice mask, the ensemble mean annual SMB integrated over the whole of Antarctica between 1987 and 2018 is 2329 ± 94 Gt per year. The RACMO2.3p2 model has a value closest to the ensemble mean with the high resolution HIRHAM5 model 190 GT over this number and the COSMO-CLM$^2$ model 368 Gt below. The HIRHAM5 0.11° and MAR$_{v3.10}$ numbers are almost exactly the

same however at 2452 and 2445 respectively around 150 Gt above the mean but MetUM, like COSMO-CLM$^2$, is much lower at about 138 Gt and 368 Gt below the mean respectively. Given that the models perform fairly similarly when evaluated against SMB observations we here give all models equal weight, although we suspect that there is a dry bias in COSMO-CLM$^2$ and a wet bias in HIRHAM5 0.44°. With an identical forcing from ERA-Interim, the present day estimate of the surface mass budget of Antarctica ranges from 2519 Gt to 1961 Gt per year, a 558 Gigatonne range that alone is equivalent to around 1.5 mm of

global mean sea level rise. Clearly narrowing this range for the purposes of estimate sea level change at present and in the future is an important task and for this reason we have evaluated the models against observations in Antarctica (see below).
There is no obvious strong trend in the modelled SMB in any of the models or in the driving ERA-Interim model. However, shorter periods within the time series can appear to have quite strong trends, for example a steady declining trend is apparent through the 1990s and 2000s but appears to have reversed since 2014, suggesting that distinguishing noise from signal will be

challenging in coming decades and emphasising the importance of long time series of observations. SMB variability is a result of low- and mid-latitude weather variability, interannual variability is particularly large at the beginning of the ERA-Interim period up to 1990 and we hypothesise this is related to improved data assimilation in the southern hemisphere in the period



between 1979 and 1989 (Dee et al., 2011). The models disagree on both the magnitude and the sign of the overall trend in the 1987 to 2018 common period of all models. It is clear from figure 8 that the external forcing model, in this case ERA-Interim, is extremely important in determining both the total SMB and the year-to-year variability in the SMB trend, even though the absolute values are somewhat dependent on the individual RCM. This is not an unexpected result given that these

are all limited area models forced at the boundaries but it has important implications for estimates of future projections of SMB in Antarctica. Decadal and multidecadal scale climate variability expressed in global climate models will have a strong influence on Antarctica mass budget (including the dynamical components via ocean forcing) that may suppress or enhance the anthropogenic forcing in ways that are difficult to predict given the large internal variability in the system. Long climate simulations with large ensembles will be necessary to define the likely range of internal climate variability and this poses

challenges of computing resources when regional downscaling is required to represent the spatial patterns of SMB over the ice sheet at high resolution.

Even between models with similar values for the integrated SMB there is substantial spatial variability in the pattern of SMB, as shown by the basin level breakdown in Figure 5 and the variation from the ensemble mean in Figure 6. These together show a nuanced picture. Over most of Antarctica, particularly in the east, the variation between models is rather small, the biggest

deviations are largely around the coast. These small areas have a disproportionate influence on the continental integrated SMB values due to high accumulation rates. Basins in west Antarctica, and particularly on the Antarctica peninsula have very large differences where for example, HIRHAM5 0.11°shows an average annual SMB of 176 Gigatonnes but COSMO-CLM$^2$ has the lowest estimate of 46 Gigatonnes in the same basin. The MAR model which shows an integrated SMB value similar to HIRHAM5 over the whole continent gives 130 Gt in the same basin, closer to the RACMO2.3p2 value of 134 Gt while

MetUM is again lower at 96 Gigatonnes. Clearly, averaging SMB over the whole continent smooths out a good deal of the spatial variability which in turn is also important for driving ice dynamics. Equally, as some basins especially in west Antarctica have very high precipitation rates, differences between models in a relatively small areas here can make a large contribution to the difference in the integrated numbers over the whole continent.

Similarly, relatively small differences in ice masks that are primarily in coastal regions with high accumulation rates can lead

to relative large differences in SMB estimates (see Figure A1) as Vernon et al. (2013) have also shown in Greenland. Figure A1 in the Appendix compares the ice masks of all the models. We found that although the variation looks quite small, the grid points affected include some of the highest precipitation points within the domain and thus small differences can have large effects. This is one of the main differences between the earlier RACMO2.1, with one of the smallest ice masks, and RACMO2.3 for example. Almost all the other models were larger around the entire coastline. The total SMB integrated over the continent

is therefore highly sensitive to the size of the common mask. For example, the SMB for HIRHAM5 0.11°is computed on its native mask gives an integrated SMB on average 9.95% higher compared to the common mask result, even though the native mask is only 2.93% larger than the common mask. These differences suggest that the CORDEX community should agree a common protocol to calculate the ice mask to reduce uncertainties in Antarctic ice sheet SMB. The deviation from ensemble mean SMB shown in Figure 6 suggests that while over the high plateau of east Antarctica there is little deviation in general,

much bigger differences occur between model SMB estimates around the Transantarctic mountains where the effect of higher



resolution becomes obvious in resolving the topography but model physics also likely plays a role. We see a similar effect in the high relief topography of west Antarctica. Finally, our results show that between 14% (COSMO-CLM$^2$) and 19% (MetUM) of the SMB is accounted for by the ice shelves around Antarctica.

A comparison of the high and low resolution HIRHAM5 simulations is interesting here as the models are identical other
than resolution. There is a substantial difference in the location of the maximum upslope precipitation as well as the downslope precipitation shadow. We attribute these differences to resolution that allows high resolution simulations to better represent steep topography. A similar but less marked impact is seen between the earlier RACMO2.1P and newer RACMO2.3p2 though in this case changes in model physics may also be responsible.

## 4.2   Model Evaluation with Observations

Clearly, evaluating the models against observations is very important for assessing where there are significant biases, but evaluation of model performance is significantly hampered by the lack of observations in key regions. Nonetheless we are able to show that the models have some skill in simulating surface climate, particularly temperature and pressure, as well as SMB. The skill in simulating climate does not however translate perfectly to simulating SMB, partly due to the difficulties of modelling and evaluating precipitation, as our analysis shows, where e.g. COSMO-CLM$^2$ better simulated surface climate
than HIRHAM5 but has an lower skill in SMB. The difference can be explained as variables such as temperature and pressure are more easily measured and models have been optimised to give good performance. Antarctic SMB is dominated by the precipitation term that is much harder to measure accurately and also has much higher uncertainty in models.

SMB observations themselves, are not always very reliable and sub-grid scale surface snow processes, such as the build up of sastrugi can give substantially different results over short spatial scales (Andersen et al., 2006). It is therefore important to
break down the data into different regions and elevation classes to see where models have better or weaker performance. We note the scatter in both models and observations within the different elevation bins and that the two polar optimised models (MAR and RACMO) perform, broadly speaking better than the others (see also Figures A2 to A7 in the appendix), though the differences are rather small in some of the elevation bins and not always very significant. It is clear that more work needs to be done to understand exactly how SMB varies spatially over the continent in order to better optimise parameterisations. The use
of nudging in models does however seem to make it easier to replicate both observed climate and SMB in RCMs, we discuss further below the use of nudging in regional climate simulations.

## 4.3   Processes Important for Ice Sheet SMB

Evaluation against observations helps to identify missing and mischaracterised processes within RCMs. Models that have not undergone specific adjustments for Antarctica clearly represent the SMB in Antarctica more poorly than those that have bee
adjusted. Other biases are also clear in this analysis. The driest model COSMO-CLM$^2$, underestimates SMB close to the coast, a region very relevant for total ice sheet mass balance. This is due to an overestimated sublimation amplified by an underestimated snowfall rate close to the coast. High values for the sublimation originate from an underestimated albedo due to aging of the snow that occurs too fast in the model (Souverijns et al., 2019). The low values for the snowfall rate is likely



related to cloud microphysics, namely a too slow conversion of ice to snow or a too slow deposition of water vapor on the solid hydrometeors. Currently efforts are ongoing to improve the coastal SMB performance in COSMO-CLM$^2$. The HIRHAM5 climate simulations both appear to have a wet bias, likely again related to the cloud microphysics and precipitation schemes but also probably a result of a diagnostic precipitation scheme commonly used in hydrostatic models. The models typically

have a wet bias on the upslope of steep topography and a dry bias on the downslope. The RACMO2.3 model shows a similar, though less pronounced effect that derives also from the IFS physical schemes (Hermann et al., 2018; Schmidt et al., 2017). New prognostic precipitation schemes have been developed in numerical weather prediction models to solve this problem (R. Forbes, Tompkins, A.M., Ungatch, A., 2011) and implementation of a similar prognostic scheme in MAR probably explains the different pattern of SMB in areas with steep topography (Agosta et al., 2019). As RACMO and MAR are the only two models

that have a specific subsurface scheme for ice sheets in this model comparison we have excluded discussion of melt and runoff and this will likely be the subject of future work. Given the high amount of precipitation over Antarctica this runoff is still very small in absolute and relative senses but as a warming future climate is expected to bring increasing amounts of melt a more sophisticated treatment that includes refreezing within the snowpack will become increasingly important. More importantly, with respect to the radiative schemes within the models, adding an ice sheet specific snowpack to the surface module in MAR

and RACMO does improve the surface temperature (and 10 m snow temperature) and therefore the air temperature. This is clear in fig. 2 and may also be a factor in some of the biases shown in fig. 1. Improving these surface schemes is therefore important not just for future projections of SMB but also to improve the near-surface climate.

## 4.4   Model Topography and Resolution

The inclusion of two simulations with the HIRHAM5 model, varying only the resolution allows us to assess the impact that

higher resolution has on the results as shown in 7 and Table 4. The added value from a higher resolution is higher spatial variability that should better capture local topography and associated weather phenomena. This is especially important in areas of high relief such as in the coastal areas and around the Transantarctic Mountains. These are also the areas where models vary from each other and the ensemble mean the most. While there are very few observations to confirm the better performance on a local scale, the pattern of SMB suggests that the high relief rugged topography is better captured in HIRHAM5 $0.11°$ than $0.44°$.

However, there is a cost of a high resolution model also. Not only is the higher resolution more computationally expensive, in a set-up like the one described here where there is no nudging, the larger number of grid points gives increased degrees of freedom for the model to evolve freely and thus introduces more internal variability. While this is not necessarily a problem for climate simulations in the future, the enhanced internal variability is inevitably punished when compared with observations and models that have been internally nudged.

Nudged models (MAR, RACMO, COSMO-CLM$^2$) show a generally lower variance from the ERA-Interim mean SMB compared to the unnudged models (HIRHAM5, MetUM), though MetUM, run as a hindcast, shows the closest values to ERA-Interim overall. They also show a closer match to observed climate than the unnudged model runs. The advantages of nudged runs are thoroughly explored in van de Berg and Medley (2016) who run two versions of RACMO2 for Antarctica one nudged and one not nudged. They find that RACMO2 nudged gives SMB results that better represent the temporal variability





of the observations, because the top of the atmosphere is constrained, thus avoiding the model deviating too far from large scale systems in the mod-latitudes. The nudging as applied in RACMO is not spectral nudging but relaxation of temperature, pressure and wind fields and this leads to some systematic mid-tropospheric warming, and hence to slightly lower SMB in the interior of Antarctica also. Other studies (Alexandru et al., 2009; Berg et al., 2013) show that spectral nudging can also lead to

lower precipitation extremes and reduced vorticity while Akperov et al. (2018) shows better representation of Arctic cyclones in nudged models. The daily reinitialisation and close forcing by ERA-Interim also explains why the MetUM modelled SMB is closest to the ERA-Interim values when integrated over the common mask.The MetUM simulation is a hindcast series where the full prognostic model state is replaced daily or twice-daily. The series is technically made continuous by construction, but it is in fact likely to be discontinuous in terms of energy, momentum and moisture budgets and like all nudged models, they are in

general not energy, moisture or momentum conserving. Berg et al. (2013), argue for caution in applying nudging during climate simulations as while it compensates for the RCM's deficiencies in meso and large scale circulation, the assumption is that the driving model represents the large-scale circulation well. In the ERA-Interim re-analysis dataset, this is a minor problem, but for free-running GCMs, large-scale circulation may well be more poorly simulated. As the external forcing controls what s delivered on the boundaries, future projections of Antarctic climate and ice sheet change will be highly controlled by the quality

of the forcing on the RCM boundaries. Models nudged internally within the domain will be further constrained in estimates of SMB by the driving models, implying rigorous assessment of global climate models should be performed before downscaling GCMs for future projections to determine which biases will be introduced (Agosta et al., 2015; Barthel et al., 2019).

## 5   Conclusions

The Polar CORDEX regional climate simulations for Antarctica are a valuable and freely available dataset for climate re-
searchers. In this paper we have compared the models against each other and against observational datasets. Much more analysis is possible and will be followed up by this group. We hope also to encourage other scientists to make use of the CORDEX dynamically downscaled models. Analysis and model intercomparison is a useful technique to evaluate models and to show directions for model improvements. Our results can be summarised as showing that the RCMs in this analysis produced skillful climate simulations over the Antarctic continent though with more uncertainty surrounding estimates of SMB. There is

a high annual and decadal and variability in SMB across Antarctica and no clear long-term trend. Model resolution and model dynamics interact in interesting ways in areas with high relief and complex topography that make it important to focus on observational campaigns in these regions.There is closer model agreement on SMB for the interior of the Antarctic ice sheet than there is in the margins and on the Antarctic peninsula. The largest areas of disagreement between models are primarily in west Antarctica. In this paper we focus mostly on precipitation and sublimation/evaporation, but reliable subsurface snow

and firm schemes will become increasingly important, particularly when making projections of SMB in the future. Models that have been optimised for the Antarctic climate and which incorporate nudging, typically demonstrate more model skill than those which do not.





*Data availability.* Model outputs used in this paper are available to download from the CORDEX archive, see https://www.cordex.org/data-access/how-to-access-the-data/ for instructions. In addition The COSMO-CLM[2] monthly output of key variables is open-access available (https://doi.org/10.5281/zenodo.2539147). Output for key variables from the high resolution HIRHAM5 simulations are available here: http://ensemblesrt3.dmi.dk/data/prudence/temp/RUM/HIRHAM/ANTARCTICA/ERAI/, further data is available on request. MAR3.10
5   monthly outputs are available here: ftp://ftp.climato.be/climato/ckittel/MARv3.10/ while all other variables are available on request.

## Appendix A: Appendix A: Ice Masks

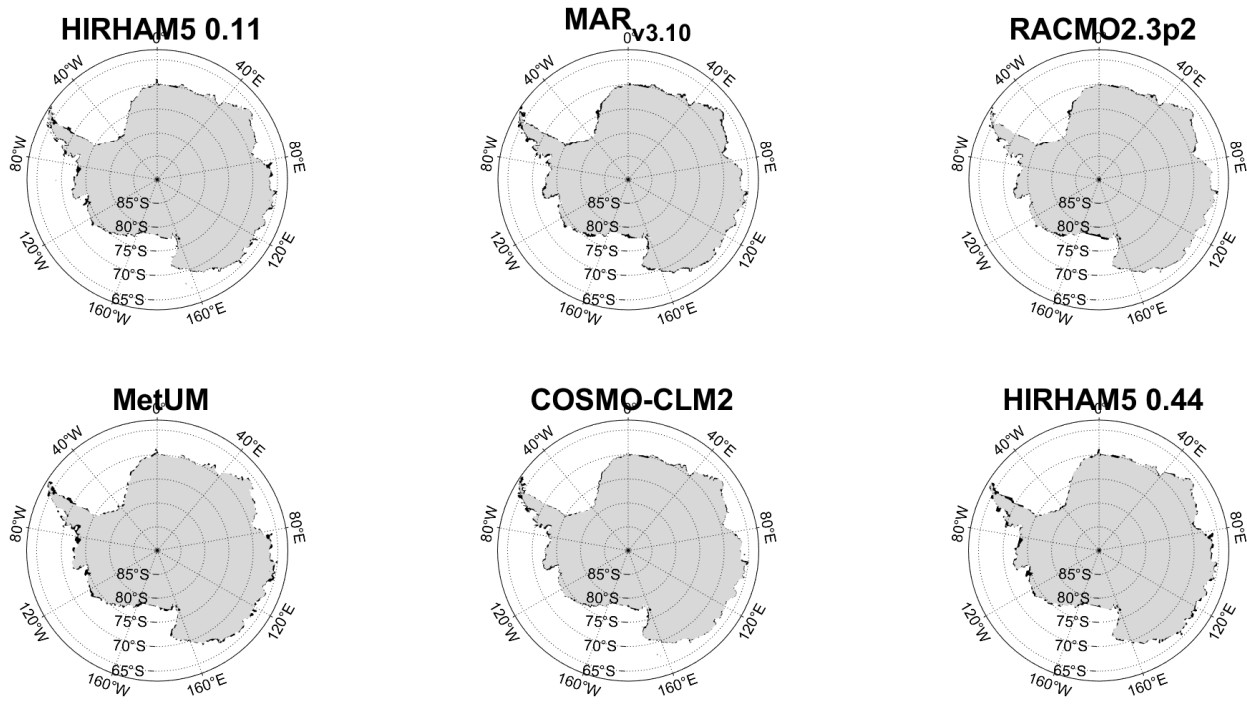

**Figure A1.** Each sub-figure shows where the common mask and the individual model masks are identical, grey is ice sheet or land in both masks. Black shows variation between model and common masks.

All title masks are larger than the common mask, HIRHAM5 0.11°is 2.43 % larger, MAR$_{v3.10}$ is 2.89 % larger, RACMO2.3P2 is 1.85 % larger, MetUM is 2.49 % larger, COSMO-CLM[2] is 1.94 % larger, and HIRHAM5 0.44°is 2.49 % larger. Some of the differences is due to inclusion of nunataks and mountain ranges within the continent. The common mask also includes
10   nunataks.





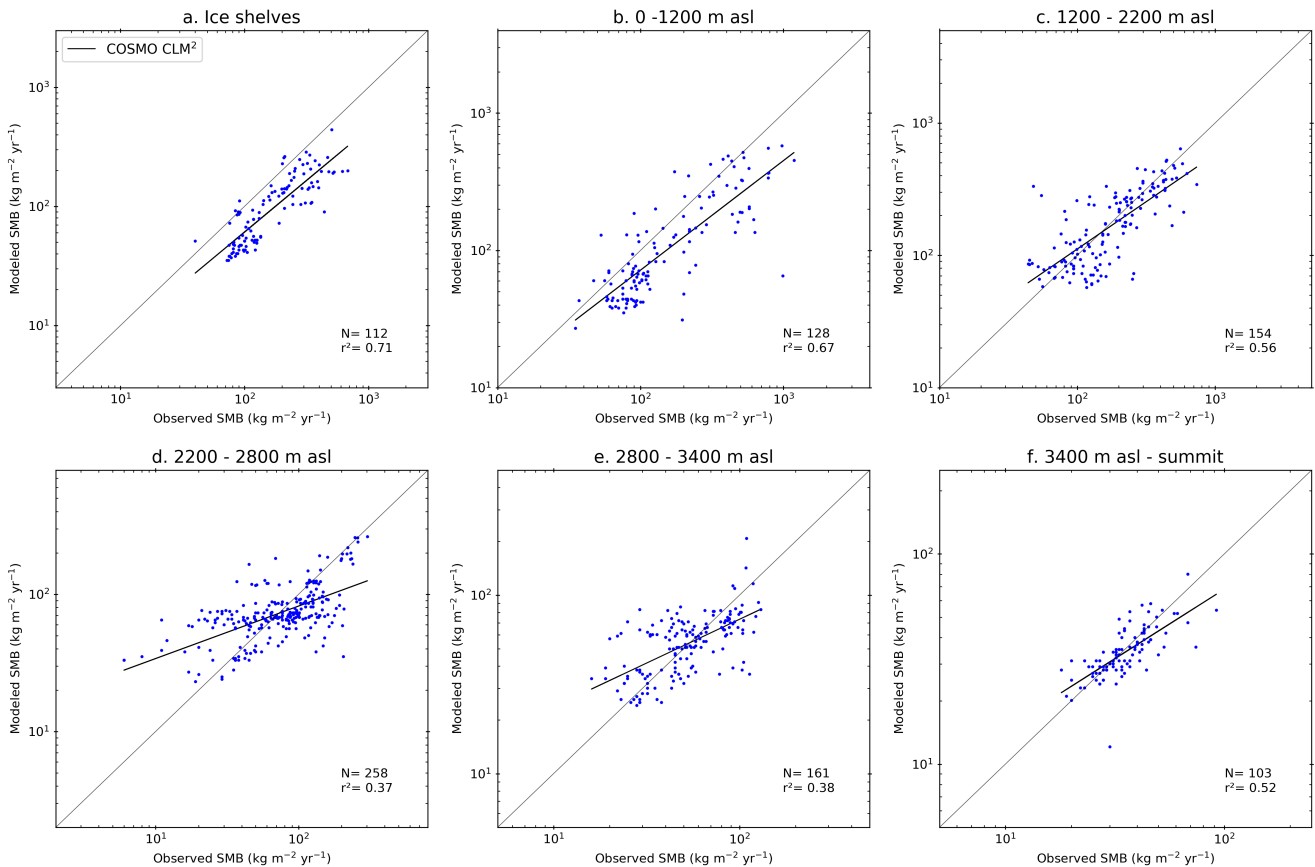

**Figure A2.** Comparison between COSMO-CLM$^2$ and observed SMB (units= kg m$^{-2}$ yr$^{-1}$) over the ice shelves (a) and by elevation classes (b-f). Due to the use of logarithmic axes, only positive values for the observed and modelled SMB from all the RCMs in this study are used (number for each bin N). Finally, the regression coefficient of each regression line is also shown (r$^2$).



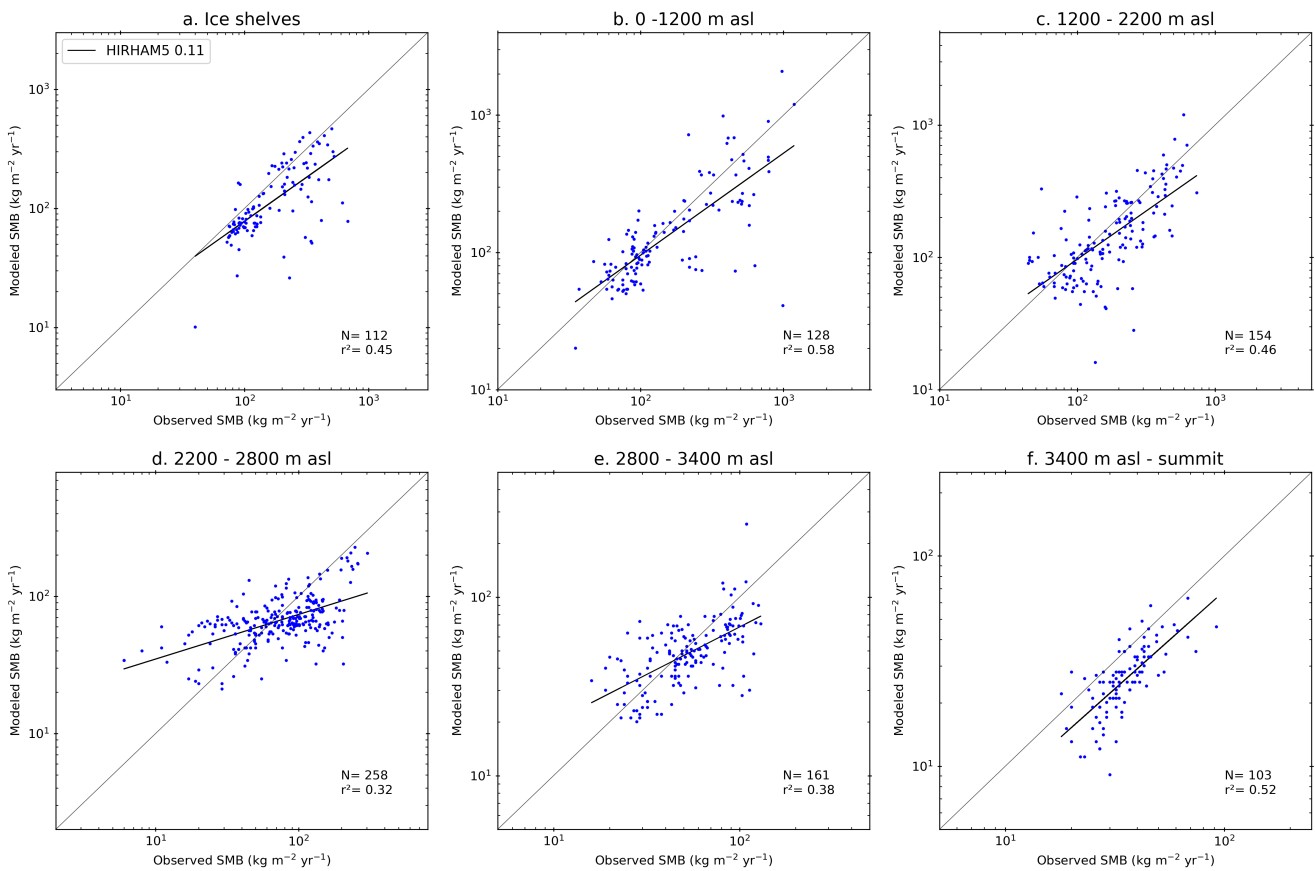

**Figure A3.** Comparison between HIRHAM5 0.11° and observed SMB (units= kg m$^{-2}$ yr$^{-1}$) over the ice shelves (a) and by elevation classes (b-f). Due to the use of logarithmic axes, only positive values for the observed and modelled SMB from all the RCMs in this study are used (number for each bin N). Finally, the regression coefficient of each regression line is also shown (r$^2$).



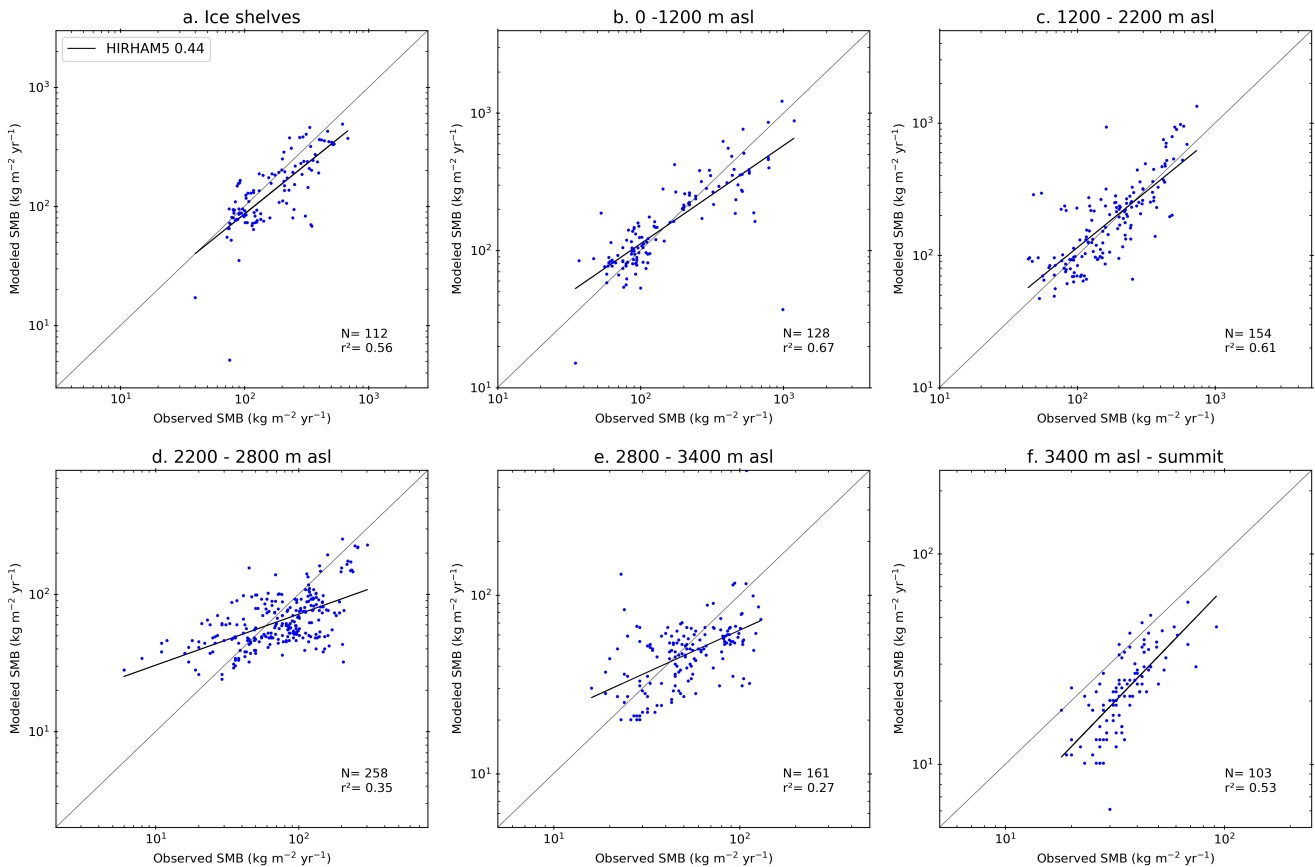

**Figure A4.** Comparison between HIRHAM5 $0.44°$ and observed SMB (units= kg m$^{-2}$ yr$^{-1}$) over the ice shelves (a) and by elevation classes (b-f). Due to the use of logarithmic axes, only positive values for the observed and modelled SMB from all the RCMs in this study are used (number for each bin N). Finally, the regression coefficient of each regression line is also shown (r$^2$).



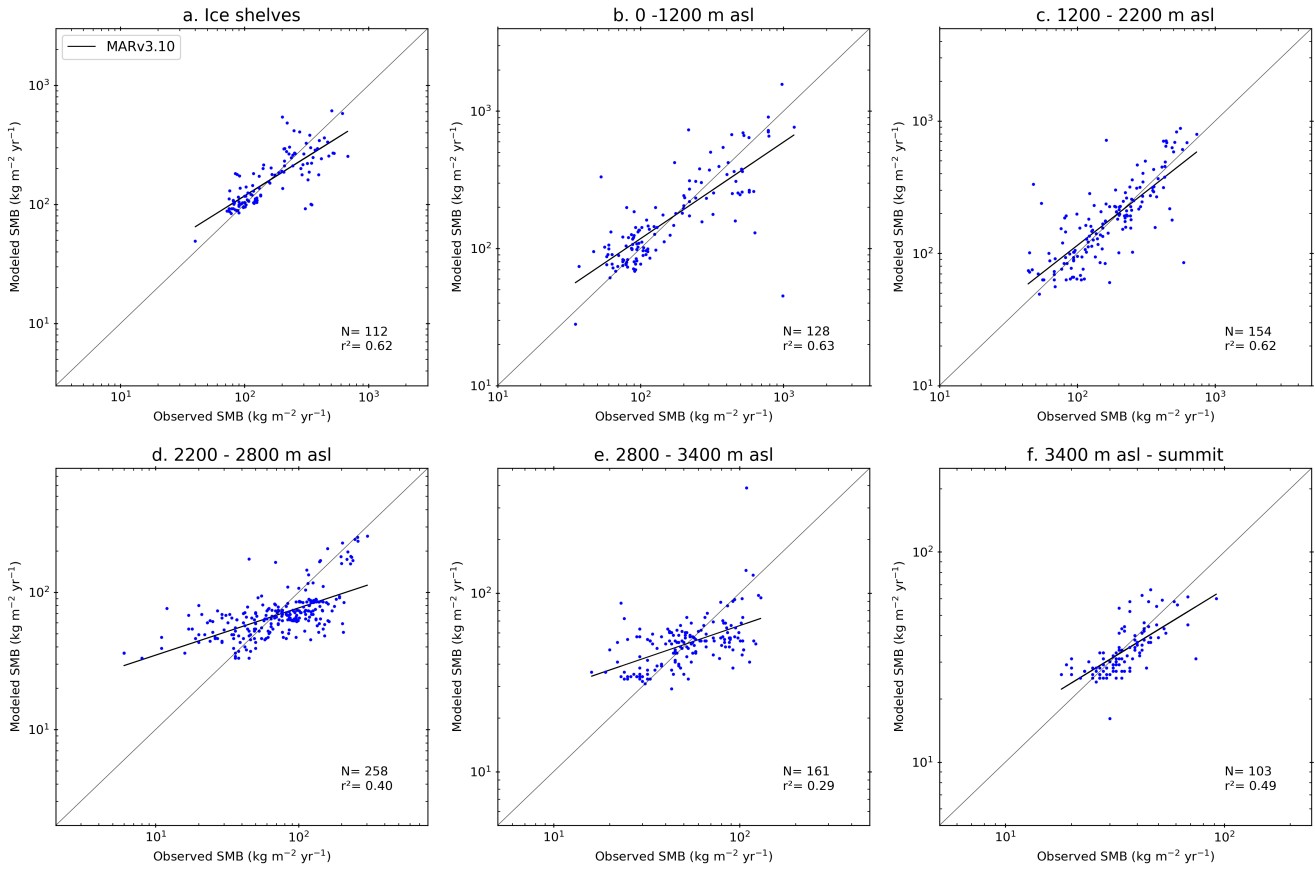

**Figure A5.** Comparison between MAR$_{v3.10}$ and observed SMB (units= kg m$^{-2}$ yr$^{-1}$) over the ice shelves (a) and by elevation classes (b-f). Due to the use of logarithmic axes, only positive values for the observed and modelled SMB from all the RCMs in this study are used (number for each bin N). Finally, the regression coefficient of each regression line is also shown (r$^2$).





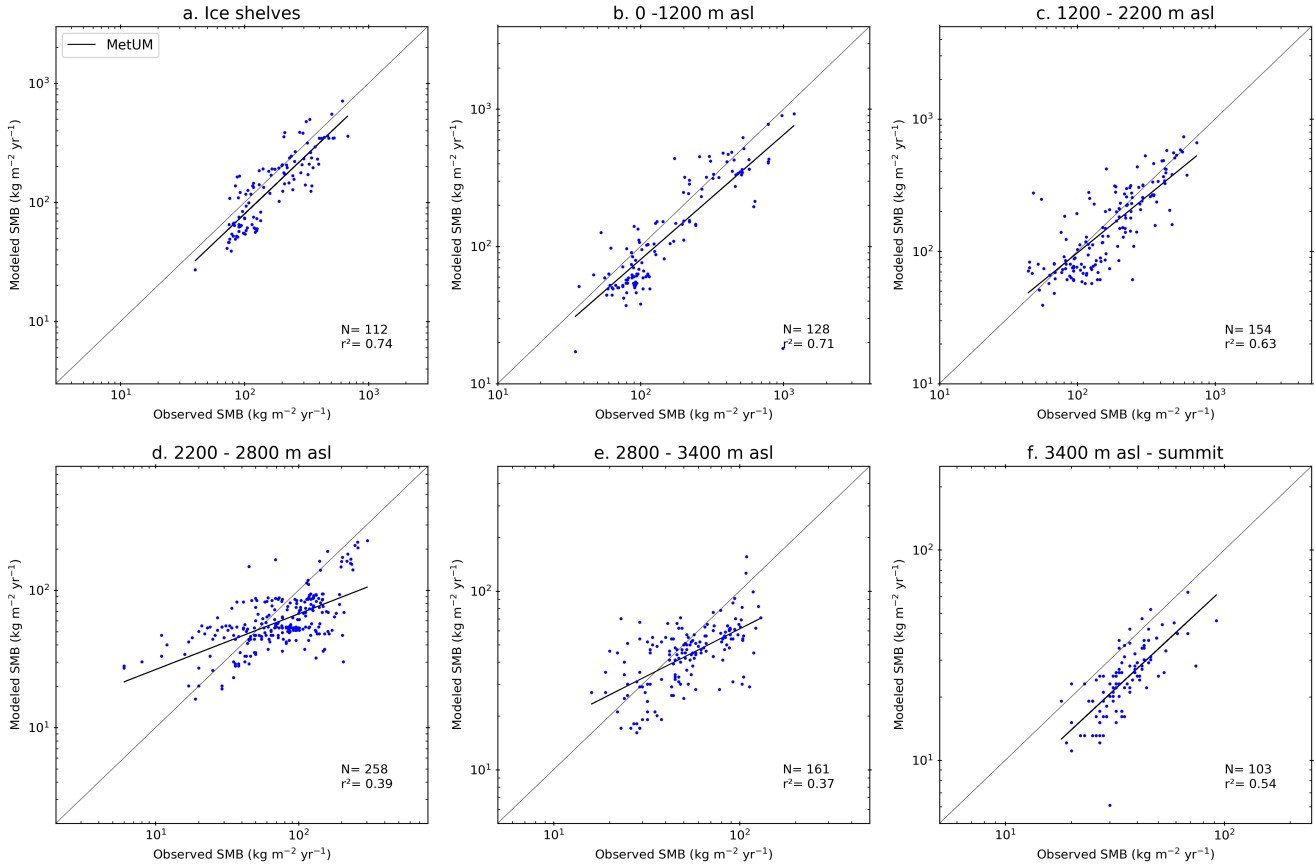

**Figure A6.** Comparison between MetUM and observed SMB (units= kg m$^{-2}$ yr$^{-1}$) over the ice shelves (a) and by elevation classes (b-f). Due to the use of logarithmic axes, only positive values for the observed and modelled SMB from all the RCMs in this study are used (number for each bin N). Finally, the regression coefficient of each regression line is also shown (r$^2$).

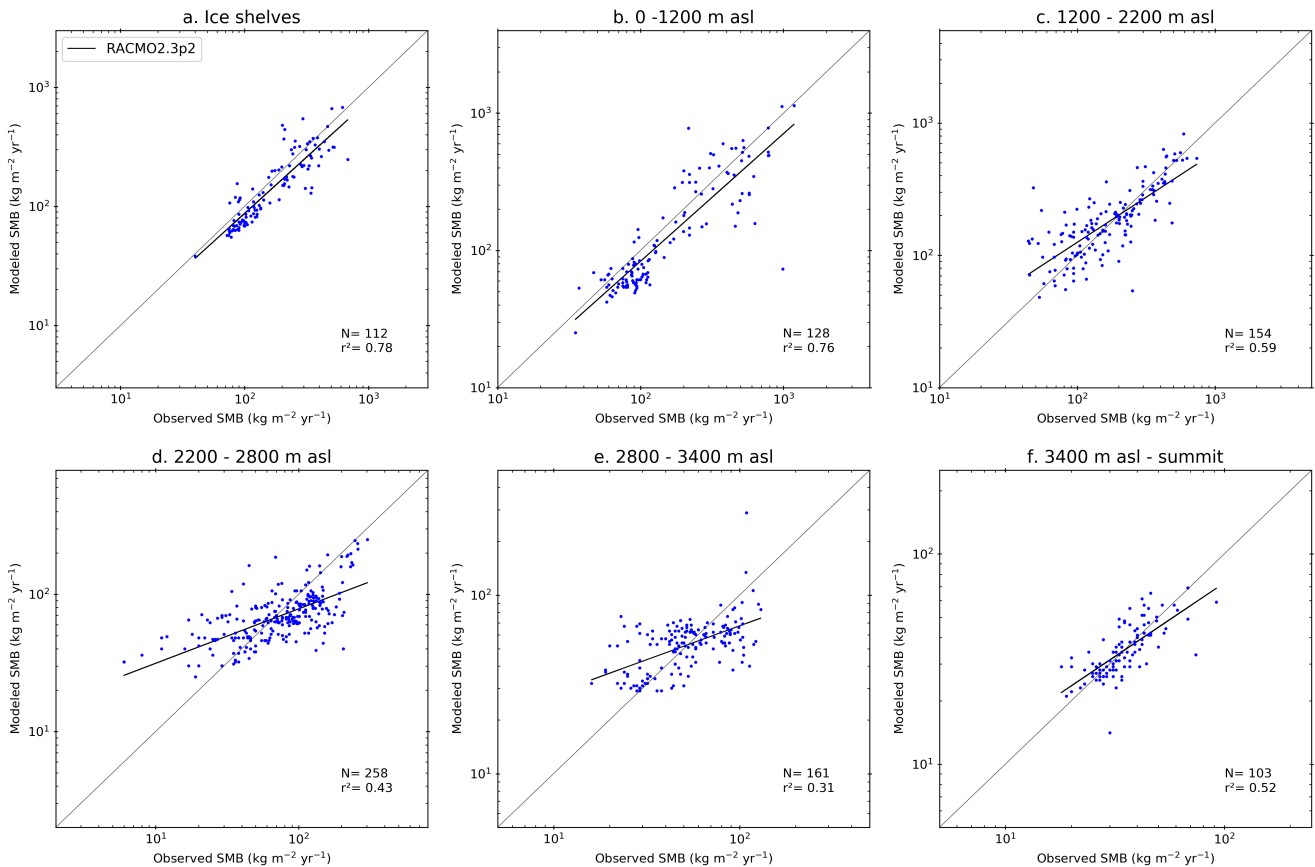

**Figure A7.** Comparison between RACMO2.3p2 and observed SMB (units= kg m$^{-2}$ yr$^{-1}$) over the ice shelves (a) and by elevation classes (b-f). Due to the use of logarithmic axes, only positive values for the observed and modelled SMB from all the RCMs in this study are used (number for each bin N). Finally, the regression coefficient of each regression line is also shown (r$^2$).

## A1

*Author contributions.* RM and SBS conceived the study, analysis of simulations was carried out by CK, JMVW, NH and RM. Model simulations were carried out by FB, CK, NS, AG, AO, SW, TP, JWVM, and EVM. All authors contributed to the manuscript.

*Competing interests.* We declare no competing interests.

5   *Disclaimer.* TEXT



*Acknowledgements.* Acquisition of meteorological data in Adélie Land has been made with the financial and logistical support of the French Polar Institute IPEV (programme CALVA-1013).

The COSMO-CLM$^2$ integrations were supported by the Belgian Science Policy Office (BELSPO; grant number 747 BR/143/A2/AEROCLOUD) and the Research Foundation Flanders (FWO; grant number 748 G0C2215N and GOF5318N (EOS ID: 30454083)). Computational resources
and services were provided by the Flemish Supercomputer Center, funded FWO and the Flemish Government - department EWI. Matthias Demuzere, Jan Lenaerts, Irina Gorodetskaya and Sam Vanden Broucke are gratefully acknowledged for supporting the COSMO-CLM$^2$ integrations. COSMO-CLM$^2$ is the community model of the German regional climate research jointly further developed by the CLM-Community. Computational resources for MAR simulations have been provided by the Consortiumdes Équipements de Calcul Intensif (CÉCI), funded by the Fondsde la Recherche Scientifique de Belgique (F.R.S. – FNRS) undergrant no. 2.5020.11 and the Tier-1 supercom-
puter (Zenobe) of theFédération Wallonie Bruxelles infrastructure funded by the WalloonRegion under grant agreement no. 1117545. C. Kittel's work was suppported by the Fonds de la Recherche Scientifique – FNRS undergrant no. T.0002.16

R Mottram, N. Hansen and S.B. Simonsen acknowledge the ESA Climate change initiative for the Greenland ice sheet funded via ESA-ESRIN contract number 4000104815/11/I-NB and the Sea Level Budget Closure CCI Project funded via ESA-ESRIN contract number 4000119910/17/I-NB. HIRHAM5 regional climate model simulations were carried out by R.M. and F.B. as part of the ice2ice project,
a European Research Council project under the European Community's Seventh Framework Programme (FP7/ 2007-2013)/ ERC grant agreement 610055.



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
