# Peer review of "What is the Surface Mass Balance of Antarctica? An Intercomparison of Regional Climate Model Estimates"

_The Cryosphere, 2019_

## Referee Comment (RC1) · Jan Lenaerts (Referee) · 25 Feb 2020

Review of Mottram et al: "What is the Surface Mass Balance of Antarctica? An Intercomparison of Regional Climate Model Estimates"

*By Tessa Gorte and Jan Lenaerts, University of Colorado Boulder.*

Mottram and co-authors present an intercomparison of different regional climate models regarding their performance in simulating Antarctic Ice Sheet surface mass balance. They show that the RCMs, all forced by ERA-Interim at their boundaries, show overall satisfying (but to a varying degree) correspondence with available weather and SMB observations, and that many remaining biases are common between the different models. The integrated ice sheet SMB varies widely from model to model, but interannual variability is very similar. Overall, we think that this is an interesting paper, containing relevant and important results for the climate, SMB, and ice sheet modeling communities, and very fitting for potential publication in The Cryosphere. However, this paper lacks a bit of context and broader impacts in its current form, and it suffers from some internal inconsistencies, ambiguities, and poor figure and language quality in places. We would invite the authors to consider our general and more specific comments, highlighted below.

**General comments**

First of all, in many places it is not clear to what the ice sheet integrated SMB numbers refer to, i.e. grounded ice sheet or full ice sheet (including ice shelves)? That's an important issue to improve, not only to enhance clarity, but also since the former is directly translatable to sea level equivalent, the latter is not. An obvious place to start is the abstract (e.g. page 1, line 7 and 11). Using appropriate labels and explanations, and clearly separating grounded and full ice sheet throughout the paper would be essential.

Second, although we understand that the authors want to refrain from 'ranking' the models, we would argue that, based on the input-output method of determining mass balance (in e.g. the IMBIE assessments), one could qualify the new RACMO2 and MARv3 models more realistic than other models. Using other models would draw a completely different picture of AIS mass balance; based on Table 3, using e.g. COSMO-CLM would more than double current AIS mass loss, and HIRHAM would suggest AIS mass gain, both of which cannot be reconciled with other methods that determine AIS mass balance (GRACE, altimetry, etc.). A discussion on this would strengthen the impact of this paper beyond a straightforward intercomparison, and inform the community on strengths and weaknesses of the different models.

Thirdly, many of the figures are very difficult to read, and colors showing different models are difficult to separate. Moreover, the figures could use a bit more explanation in the text as well as in the caption. A lot is left to the reader to decipher these figures (which potentially convey very interesting information).

Lastly, language needs to be improved throughout. A few places are consistently lacking commas: after/around thus, therefore, moreover, etc. Several sentences were a bit long and could be broken up to make them easier to read. The authors switch between active and passive voice quite often throughout the text (i.e. "parameterizations are included" instead of "the models

include parameterizations), suggesting that various authors have contributed to the writing and the end result is somewhat heterogeneous. We have pointed out a few locations below, but there are many more in the paper. Try to avoid phrases like 'clearly' throughout the paper. This is a subjective statement, and findings may not be so clear to the reader as it is to the authors.

**Specific Comments**

P1: Why are SMB and Gt given as abbreviations but not AIS which is abbreviated later?
P1L1-2: Technically, Antarctica loses mass through enhanced ice discharge across the grounding line into ice shelves (not compensated by an increase in SMB), and ice shelves lose mass by *enhanced* calving and basal melt (not compensated by an increase in ice shelf SMB and/or solid ice influx). Separating these various processes can help to separate the grounded and full ice sheet (see General Comment 1).
P1L3-4: "… of crucial importance…" → "crucially important"
P2L1: "compar" → "compare"
P2L12: "… a significant part of the climate system" is a bit vague and could be expanded upon
P2L15: Is "submarine melting" a common phrase for basal melting?
P2L20-21: Scambos and Shuman maybe shouldn't be in all caps.
P2L27-28: "In the future… climate change" → this sentence requires a change in punctuation for readability for me. For instance, consider changing to "In the future, a "greenlandification" of the ice sheet climate (increased melt and refreezing within the snowpack) is projected due …"
P3L12-16: "Souverijns et al… peer review literature" → this is quite a long sentence. Perhaps consider breaking it up for readability.
P3L27: It might be good to list all 5 RCMs at the beginning of the Methods section
P7L4: "Parameterizations are included…" → "The models include parameterizations…"
P8L2: "… nudging whether spectral or with simpler techniques keeps…" → "nudging, whether spectral or with simpler techniques, keeps…"
P9L6: "Weather observations are used…" → change to active voice
P9L22-27: Change paragraph to active voice
P10L10-11: "Observations between… 5 years" → consider rephrasing for readability
P10L15-20: Authors say SMB was computed in 3 steps but only two seem to be explicitly mentioned.
P11L10: So you're saying that the higher resolution the model, the poorer skill it will show due to increased internal variability? Please clarify, since this is essentially contradicting many other studies that are suggesting enhanced performance with resolution.
P11L17: What causes you to suspect "…biases in cloud cover and long-wave radiation…" are the leading factors in divergence from observation? How would you expect a model bias that overestimates cloud cover to effect observations, for example?
P12L7-8: "The models can be divided into two groups…" → how are you dividing these groups? Not sure we understand the origin or the purpose of having two groups here.
P12L22: Extra parenthesis.
P12L23: "…in the colder, and therefore higher elevation locations, while…" → is this supposed

to be "…in the colder, and therefore higher elevation, locations while…"? Also, perhaps consider changing the order to "in the higher elevation, and therefore colder, …" such that it seems like temperature is a function of elevation and not the other way around.

P14L14: What do you mean by "good results" exactly?

P16L2: "…we here show…" → "…here we show…"

P17, Table 2: Arguable showing an RMSE with absolute SMB numbers that decrease rapidly from the coast to the interior is not justified, since the RMSE will tend to decrease along with the SMB itself. Adding relative RMSE (i.e. as a ratio to the mean) is required to compare apples to apples across the elevation bins.

P20: When looking at the ensemble mean, have you considered how your results may change if you calculate the mean on different grids? What grid did you use for this (i.e. how does this common grid resolution compare to that of any of the given models)?

P24L9: "bring" → "brings"

P25L9-11: "The HIRHam5 … below the mean respectively" → This sentence is long and difficult to read due to the lack of commas.

P25L17-20: The authors address the period of the "1990s and 2000s" for SMB trend, but since SMB is so highly variable, can you really say that this is significant/important?

P26L16: "west Antarctica" and "Antarctic peninsula" → "West Antarctica" and "Antarctic Peninsula"

P27L29: "bee" → "been"

Figure Comments:

Figure 1: These Taylor diagrams are very interesting way to convey information, but many readers will have never seen something like this before. It will be important to better clarify the metrics conveyed by the figure. For instance, we are unsure what the curved lines (i.e. ranging 1.60 to 13.50 in the left panel) are supposed to represent.

Figure 2: Could you perhaps also include a table of correlation and/or bias for each model?

Figure 4: The same comment as figure 3, but with the color bars

Figure 5: Is this meant to be rotated? Also, increase the font size again.

Figure 6: Increase label size

---

## Referee Comment (RC2) · Anonymous Referee #2 · 27 Feb 2020

Remarks to the Authors

Review of "What is the Surface Mass Balance of Antarctica? An Intercomparison of Regional Climate Model Estimates" by Ruth Mottram et al.

The Cryosphere Discuss. Manuscript Number: tc-2019-333
* * *
General comments:

This paper presents results from a first intercomparison of polar regional climate models (RCM) applied in the Antarctic Ice Sheet (AIS). The model performances were

compared and assessed in terms of surface pressure, near-surface air temperature, near-surface wind speed, surface temperature, and surface mass balance (SMB) of the AIS. The models that participate in this intercomparison project are COSMO-CLM2, HIRHAM5, MetUM, MAR, and RACMO. For some models, results from different versions are provided additionally.

My first honest impression after reading through this manuscript is that the current title "What is the Surface Mass Balance of Antarctica?" is a bit misleading, because meltwater runoff is not considered in the most participating models except for MAR and RACMO. It is true that a contribution by runoff to the changes in the present-day AIS SMB is relatively small than contributions from precipitation and sublimation/evaporation. But, runoff in the present-day AIS already cannot be neglected as presented by several studies cited in this manuscript. In the future in a warming world, the contribution by runoff to the changes in AIS SMB will become much higher almost certainly as pointed out by the authors (P. 2, L. 18 ∼ 19). Therefore, this reviewer expected that all the models calculated runoff in the present study, and as a result, I was a bit disappointed when I found the relevant description in Sect. 2.2.1.

Related to the point indicated above, the intercomparison procedure for SMB sounds a bit inadequate to me, because the authors employ different definitions of SMB (Sect. 2.2.1). If the authors focus intercomparisons only for precipitation and sublimation/evaporation (in addition to the three surface meteorological properties as well as the surface temperature), it makes sense and highlights key differences in model physics employed by these participating models more clearly. This reviewer recommends the authors to reconsider the title of this manuscript: maybe something like "intercomparison of Antarctic ice sheet surface meteorological conditions simulated by five different regional climate models" would be appropriate.

However, the intercomparison of RCMs performed in the AIS itself is a considerable new challenge, so provides the latest comprehensive information related to these RCMs, which is very informative for readers certainly, so deserved to be published

in the journal The Cryosphere.

In the following part, this reviewer gives specific comments to be considered before the publication. Please note that page and line numbers are denoted by "P" and "L", respectively.
* * *
Specific comments (major)

P. 2, L. 7 ∼ 8: What kind of measurements do the authors think here (observational campaigns)? Maybe it is not necessary to indicate explicitly here; however, please suggest something at least in the discussion and/or conclusion sections.

P. 12, L. 2 ∼ 3: What kind of physical mechanisms do the authors think here? Please detail more.

P. 12, L. 5: How large do the authors think the uncertainties are?

P. 12, L. 5 ∼ 6: Readers cannot know the difference in turbulent heat schemes, because they are not described in this manuscript.

P. 24, L. 13 ∼ 15: It is interesting to see the model-simulated precipitation integrated over the common ice sheet mask by COSMO-CLM2 tends to be lower than that by the parent data ERA. It is because, precipitation in a dynamically downscaled data is higher than precipitation in its parent data in general. Please discuss.

P. 29, L. 27: Please suggest what kind of measurements do the authors think necessary in the "observational campaigns"?
* * *
Specific comments (minor)

P. 4, L. 13 ∼ 18: Is it OK to understand MAR 3.6 is older than MAR v3.10? If yes, it is a bit confusing isn't it?

P. 8, L. 6: What do the authors mean by "cloud physics"? To me, it is difficult to understand why "cloud physics" is resolved better in nudged models.

P. 12, L. 24 ∼ 25: It is not clear why the authors think so. Please explain more.

P. 12, L. 31: "For the warmer costal regions": From which data can we see this argument?

Figure 2: This figure is a bit difficult to see. Please provide a table indicating ME, RMSE, and R2.

P. 23, L. 4 ∼ 7: What is an interesting point here? I don't think the lower panel of Fig. 8 is necessary; however, if the authors think it is necessary, please discuss more about the figure. Maybe, inter-annual variations of these model results should be discussed more.
* * *
Technical corrections

P. 2, L. 1: "compar": typo

P. 2, L. 11: "a potentially important potential contributor" -> "an important potential contributor"?

P. 2, L. 22: surface mass balance -> SMB; Note this term is already defined.

P. 3, L. 22 ∼ 25: This sentence especially after "and better understand drive sea level rise . . ." is a bit difficult to understand. Please reformulate it.

P. 5, L. 11: "regional mesoscale model e.g." -> "regional mesoscale model as presented by e.g."?

P. 7, L. 14: Define $SU_{ds}$ and $ER_{ds}$.

P.9, L. 2: "GrIS": typo, right?

P. 10, L. 27: "assessing" -> "assess"

P. 12, L. 7: Indicate the publication year for Zentek and Heinemann.

P. 12, L. 25: "downwelling longwave and surface albedo" -> "downwelling longwave radiation flux and surface albedo"

P. 14, L. 18: "HIRHAM5.011": typo

---

## Referee Comment (RC3) · Anonymous Referee #3 · 5 Mar 2020

Review of Mottram et al., What is the Surface Mass Balance of Antarctica? An Intercomparison of Regional Climate Model Estimates

Summary

The authors present an intercomparison exercise of five different regional climate model surface mass balance estimates, as well as the near surface climate, over Antarctica. The authors find a large spread in total SMB (1961 to 2519 Gt year-1), which largely stems from differences in West Antarctica and the Antarctic Peninsula. Variability is quite consistent between models, which is unsurprisingly since they are all forced by ERA-Interim, but the trends differ in sign and magnitude and are quite sensitive to the time period selected. Also, not surprisingly, the nudged models simulate the near-surface climate better as they are not allowed to deviate as substantially from ERA-Interim as the un-nudged models. Finally, the authors discover that the biases are typically consistent between models. The paper presents a significant amount of work but still requires improvements. First, the manuscript has numerous mistakes throughout and needs refinement of the language in several places (see Minor Comments). More importantly, there are several major issues with the analysis that need to be addressed to improve the scientific rigor of the paper.

Major Comments

1. Throughout the manuscript, it is not clear what time periods are being used. There is the common model interval, climatological mean, etc. The authors need to be very clear throughout the paper because it often seems that different intervals are being confused in nomenclature. It's not clear to me why the common reference interval is not the common period between all models: 1987 – 2015. Throughout the paper sometimes its 1980-2010, 1987-2015, and 1987-2018. I recommend using the same interval through to avoid confusion. If the authors have a reason to use different intervals, then please make it clear what interval is being used. It is additionally unclear why 1980-2010 is representative of the climatological period, please explain.

2. Similarly, there is no discussion of significance for the statistics presented. There are claims within the text that certain models perform better than, but without significance levels, these claims lack strength and are more speculative. Trends are discussed at both long (1987-2015) and short (decadal) time intervals, but the significance is never discussed. I would caution the authors descriptions of trends, especially at short time scales, since it is very hard to observe a significant trend in SMB since its highly variable year to year. Furthermore, because this is an intercomparison paper, its important for the authors to be very clear concerning the metrics of how they conclude one model outperforms the other. Is it RMSE? R2? Bias? And what is the threshold? Is an RMSE of 93 better than 97? What if one model performs differently at different el-

evation bands? I did not find the argument compelling that the models tuned to specific Antarctic conditions outperformed the others because there was not a clear framework for comparison. The authors need to make clear the evaluation metrics and how they evaluate model performance, which will require more detailed statistical analysis throughout. Model means are compared, but its not clear if the paper considers even a simple statistic of the standard error of the mean. The Student's t-test can be used to evaluate whether the means are different. Please be transparent with the limitations of the analysis and provide meaningful significance tests on all of the comparisons, otherwise the conclusions are speculative rather than significant.

3. All of the RCMs presented are forced by the ERA-Interim reanalysis product. I find it concerning that there is no discussion of the role of using a single reanalysis to force all of the RCMs. Thus, this is not a definitive evaluation of the full range of possibilities in SMB, but rather a range due to RCM differences alone. There should be more discussion about how there would be additional spread due to varying choice of forcing; specifically, what is the impact of comparing models that are all forced by the same reanalysis. I think the paper needs to tone done the claims about the work providing the "likely range of SMB" in the first sentence of section 4.1, as it is more the likely range of RCMs forced by ERA-Interim. Basically, this explores the range in RCM space, but not reanalysis forcing space.

4. It's obviously quite a challenge to compare these models, which have differing levels of complexity. But it seems that the comparison would be better suited by comparing all the variables consistent between models (Precip-Evap-Subl). Otherwise, an inter-comparison doesn't shed much light on direct model to model differences. In fact, it appears that the authors could investigate whether these Antarctic specific physics actually provide improvement, which would be of great interest to the community. Therefore, the paper should do an ideal comparison of all 5 models with common variables $(P - E - S)$ and evaluate performance. Then evaluate the models with extra physics (RACMO/MAR) to see if and how much model performance improves. Otherwise, its

difficult to untangle whether those additional processes provide any more information.

5. The manuscript needs to justify the use of SMB observations starting in 1950. There are regions of strong trends in snow accumulation that might end up biasing the comparison. If the issue relates mainly to reducing the number, the authors could present a comparison of only coincident SMB observations with the data, but then also provide the more liberal comparison as it currently exists in the text.

6. Finally, the paper needs to discuss the impacts of its findings. With the present day mass loss from Antarctica on the order of 100 Gt per year, this is quite concerning finding the differences in SMB from RCM choice alone are several hundred Gt per year. Please contextualize the findings in regard to how we can measure the mass balance of the ice sheets.

Minor Comments

Several model names and versions are discussed before they are described, which makes it quite hard to follow. Please reorder the sections to ease. For instance, section 2.1 and the end of Section 1 mention several models and different version, but there is no description, so its hard for the reader to follow. It would also be appropriate to cite the papers that refer to these model versions.

P1, Line 7: Is this for grounded ice only? Does it include islands and ice shelves?

P1, Line 7-8: What do the values after the $\pm$ represent? The standard deviation of all the models?

P1, Line 10-11: Why is 1980-2010 chosen as the climatological period? Later in section 2.3.3, it appears that 1987-2015 is the common modeling period that is used to "compute a climatological mean" (P10, Line 12). Please rectify.

P2, Line 1: change "compar" to "compare"

P2, Line 11: remove either "potentially" or "potential" since its repetitive

P2, Line 13: add "and" after "2002,"

P3, Line 16: remove the comma after "published"

P3, Line 23: remove "drive"

P4, Line 1: please describe what a "reinitialized hindcast" is

P4, Lines 5-7: While this is true, it might have a limit. See Lenaerts et al., 2018, which shows that often the snow is not dumped in the proper place when moving from 27 km to 5.5 km. Please add a sentence clarifying this.

P4, Line 8: add "is used" after "ensemble mean"

P5, Line 6: change "developed in" to "developed for"

P5, Line 18: the end of this sentence needs to be reworded

P6, Line 4: Do you mean "processes" not "process"?

P6, Line 22: change "includes no" to "does not include"

Table 1: What does SMB scheme mean?

P7, Line 10: change "schemes" to "scheme"

P10, Line 10: add "that are" after "2015 and 2018"

P10, Lines 14-20 need clarification

P12, Lines 4-7: This sentence is very long and needs to be split in two.

P12, Line13-14: Please reword the sentence as its confusing.

P13, Line 6: Remove "In"

Figure 3. Please add the statistics to these plots (RMSE, etc.). Also, its very difficult to distinguish the colors here. Maybe large dots would help.

Figure 4. This figure should be much bigger. It's very hard to see the colors. Also, in the caption there are "a", "b", etc., but they do not exist on the plots.

P18, Lines 9-11: are these values consistent with what is listed in the abstract?

Figure 5. This needs to be in landscape orientation. The numbers are much too small to read.

P20, Lines 5-6: What does "much clearer mean," please clarify

Figure 6. Again, these plots are too small, and the numbers are nearly impossible to read. P22, Line 2: remove "below"

There should be significance values associated with the trends. It looks like none would be statistically significant and thus are effectively no different than zero.

P24, Line 6: remove "very"

P24, Line 9: change "bring" to "brings"

P25, Line 7: Should the interval be 1987-2015?

P27, Line29-30: Do your results actually support "Models that have not undergone specific adjustments for Antarctica clearly represent the SMB in Antarctica more poorly". Look at the RMSE value in Table 2, it looks liked sometimes they perform better. Please clarify.

P 28, Line 11-12: please give values in Gt of these processes to show that they are effectively negligible

P 28, Line 20: add "fig." before "7"

P28, Lines 20-21: the sentence needs to be improved.

P29, Line 2: replace "mod-latitudes" with "mid-latitudes"

---

## Author Comment (AC3) · 26 Jun 2020

Review of Mottram et al., What is the Surface Mass Balance of Antarctica? An Inter-comparison of Regional Climate Model Estimates

Summary
The authors present an intercomparison exercise of five different regional climate model surface mass balance estimates, as well as the near surface climate, over Antarctica. The authors find a large spread in total SMB (1961 to 2519 Gt year-1), which largely stems from differences in West Antarctica and the Antarctic Peninsula.Variability is quite consistent between models, which is unsurprisingly since they are all forced by ERA-Interim, but the trends differ in sign and magnitude and are quite sensitive to the time period selected.
Also, not surprisingly, the nudged models simulate the near-surface climate better as they are not allowed to deviate as substantially fromERA-Interim as the un-nudged models. Finally, the authors discover that the biases are typically consistent between models. The paper presents a significant amount of work but still requires improvements.

We thank the reviewer for their careful reading and thoughtful comments, which have led to significant improvements in the manuscript.. We address the specific points in the text below.

First, the manuscript has numerous mistakes throughout and needs refinement of the language in several places (see Minor Comments).

This point has also been made by the other reviewers. The whole manuscript has been thoroughly proof-read and made simpler to read and more consistent in language and structure.

More importantly, there are several major issues with the analysis that need to be addressed to improve the scientific rigor of the paper.

Major Comments

1. Throughout the manuscript, it is not clear what time periods are being used. There Is the common model interval, climatological mean, etc. The authors need to be very clear throughout the paper because it often seems that different intervals are being confused in nomenclature. It's not clear to me why the common reference intervals are not the common period between all models: 1987 – 2015. Throughout the paper sometimes its 1980-2010, 1987-2015, and 1987-2018. I recommend using the same interval through to avoid confusion. If the authors have a reason to use different intervals, then please make it clear what interval is being used. It is additionally unclear why 1980-2010 is representative of the climatological period, please explain.

We agree it is a bit confusing that the models were run for slightly different periods and this also makes comparisons between them more complicated. Unfortunately we were constrained to use simulations that were already available for this analysis. We have

added a paragraph explaining the different periods used to the Methods section and explicitly mentioned all the way through the paper which periods are being used during the results and discussion when relevant. We have also done additional analysis that shows the results of the observational comparison to the different models does not substantially change when using the shorter period. This is added to the supplementary material in order to avoid further lengthening the paper.

2. Similarly, there is no discussion of significance for the statistics presented. There are claims within the text that certain models perform better than, but without significance levels, these claims lack strength and are more speculative.
Trends are discussed at both long (1987-2015) and short (decadal) time intervals, but the significance is never discussed. I would caution the authors' descriptions of trends, especially at short time scales, since it is very hard to observe a significant trend in SMB since its highly variable year to year.

We absolutely agree that detecting a significant trend in SMB is almost impossible and in fact that is partly why we include Figures 7-9. However, given all reviewers comments we have clearly not described this well enough. We have added a paragraph making this point explicitly and setting the SMB in context.

"Unlike previous studies, we detect no obvious strong trend in the modelled SMB in any of the models or in the driving ERA-Interim model. Shorter periods within the time series appear on first look to have quite strong trends, for example a steady declining trend is apparent through the 1990s and 2000s but appears to have reversed since 2014. Our results suggest that strong interannual and decadal variability makes the identification of meaningful trends over short periods very difficult. Distinguishing noise from signal will be challenging in coming decades and this also emphasises the importance of long time series of observations."

Furthermore, because this is an intercomparison paper, it's important for the authors to be very clear concerning the metrics of how they conclude one model outperforms the other. Is it RMSE? R2? Bias? And what is the threshold? Is an RMSE of 93 better than 97? What if one model performs differently at different elevation bands?
I did not find the argument compelling that the models tuned to specific Antarctic conditions outperformed the others because there was not a clear frame-work for comparison. The authors need to make clear the evaluation metrics and how they evaluate model performance, which will require more detailed statistical analysis throughout.
Model means are compared, but its not clear if the paper considers even a simple statistic of the standard error of the mean. The Student's t-test can be used to evaluate whether the means are different. Please be transparent with the limitations of the analysis and provide meaningful significance tests on all of the comparisons,otherwise the conclusions are speculative rather than significant.

This is a very important point and in part one of the drivers for this paper. We do not attempt to rank the models because it is clear from our results that on different measures, (bias, RMSE etc) the different models perform quite differently for different variables (both meteorological) and there is also a spatial component as the reviewer points out with different biases apparent at different elevation bands and in different locations. This means that most likely different models should be used for different purposes. It is also an aim of this paper however, to give clarity on exactly how the models compare, for which reason we give extensive statistics in figure 1 and table 2, which we have also expanded to include the mean value of SMB observations for the elevation bands as requested by reviewer 1.

As the paper is already very long, we propose to add 2 new figures and associated table in the supplementary section showing how the relative RMSE and mean bias compares between models for the different elevation bands. These for instance show that at high elevation COSMO_CLM, MAR and RACMO better represent SMB but HIRHAM, MetUM and COSMO-CLM have a lower mean bias in the middle elevations and MAR, HIRHAM and RACMO have a lower mean bias in the lowest elevations. In addition we have added some extra discussion comparing the different statistical methods and their use in evaluating the models in the discussion of table 2.

[Figure]

[Figure]

3. All of the RCMs presented are forced by the ERA-Interim reanalysis product. I find it concerning that there is no discussion of the role of using a single reanalysis to force all of the RCMs. Thus, this is not a definitive evaluation of the full range of possibilities in SMB, but rather a range due to RCM differences alone. There should be more discussion about how there would be additional spread due to varying choice of forcing; specifically, what is the impact of comparing models that are all forced by the same reanalysis. I think the paper needs to tone done the claims about the work providing the "likely range of SMB" in the first sentence of section 4.1, as it is more the likely range of RCMs forced by ERA-Interim. Basically, this explores the range in RCM space, but not reanalysis forcing space.

The point of this study is to determine the RCM uncertainty space rather than different boundary conditions. We specifically excluded models that ran different reanalyses as we would like to determine how different models compare with each other. However, having said that, analysis by Agosta et al., 2019 used different reanalyses to force the same model and found that the results were quite similar. We have added two extra sentences and this reference stating this in the methods section.

"All models were forced on the lateral boundaries with the ERA-Interim climate reanalysis (Dee et al., 2011) but downscaling used different grids, over slightly different domains and at different resolutions with slightly different ice masks used in the different model versions (see A1 in the appendix). Simulations with MAR forced by different reanalyses (e.g. Agosta et al., 2019) found that results were rather similar to ERA-Interim. However, in order to exclude additional variability potentially introduced by using different boundary forcings, we chose to use a single common reanalysis only"

4. It's obviously quite a challenge to compare these models, which have differing levels of complexity. But it seems that the comparison would be better suited by comparing all the variables consistent between models (Precip-Evap-Subl). Otherwise, an inter-comparison doesn't shed much light on direct model to model differences. In fact, it appears that the authors could investigate whether these Antarctic specific physics actually provide improvement, which would be of great interest to the community. Therefore, the paper should do an ideal comparison of all 5 models with common variables (P – E – S) and evaluate performance. Then evaluate the models with extra physics (RACMO/MAR) to see if and how much model performance improves. Otherwise, it is difficult to untangle whether those additional processes provide any more information.

We have also addressed this point in our response to the second reviewer. Melt is likely to become more important in the future, but at the present day melt and runoff are only observed at a few very specific locations. However, while all of the models simulate melt, they have varying degrees of complexity to calculate refreezing so purely to simplify the comparison here we focus only on the precipitation,

evaporation and sublimation terms. Then, RACMO and MAR include sublimation from blowing snow subroutines, while it would be ideal to separate these out, the physical parameterisations have been developed and tuned with these processes in the model, so it is difficult to remove them without negatively and unfairly affecting the results- we have therefore also used the sublimation from snow schemes in calculating SMB. We note that both RACMO and MAR groups have published articles demonstrating the improvement from the enhanced snow schemes (Van Wessem et al and Agosta et al., 2019).  We have added extra text to make this point in the description of the SMB in the methods section.

"As the RACMO and MAR models have been developed to include the wind blown snow sublimation terms, they cannot easily be removed without retuning the models, and for this reason we have opted to include these within the SMB calculation for these two models."

5. The manuscript needs to justify the use of SMB observations starting in 1950.There are regions of strong trends in snow accumulation that might end up biasing the comparison. If the issue relates mainly to reducing the number, the authors could present a comparison of only coincident SMB observations with the data, but then also provide the more liberal comparison as it currently exists in the text.

Unfortunately there are relatively few observations in Antarctica and including only those that were taken during the period of the simulations would make the model - observation comparison less robust. Including observations taken from 1950 increases the number of observations available to 923 comparisons from 469. More importantly, while the total number still sounds substantial, the main benefit is in fact in spatial representativeness. The 1987-2015 observations cover only a very small part of Antarctica. We discuss this problem in section 2.3.3 but we have expanded the discussion and we propose to include a new figure showing the spread of observations by date in the supplementary material. As we point out in the discussion and conclusions, the difficulty is also that the places where the models disagree most are also the areas with the sparsest observations of SMB.

Furthermore, observations starting before 1987 are often ice cores and are therefore the average of this long period (there is also a selection criterion which requires that they must last more than 5 years to be considered in the comparison. This obviously smoothes out the results of a strange year (for example due to sastrugi or scour) in addition to reducing measurement error. At the same time, we compare the average smb of the models over our entire period, so we also smooth out variability between simulated years. This means that using the observations before 1987, we likely have higher quality obs (lower measurement error and better spatial representativeness) and also slightly smoothed models. We have mitigated the problem of unrepresentativeness as much as possible by for example excluding observations before 1987 that cover too short a period (less than 5 years) in order to keep only observations representative of a mean climate. However, we are not immune to introducing biases because these

observations include biases arising from regional circulation trends that the models cannot represent, nonetheless the comparison is more robust if it represents a larger area. We have added these points in the expanded methods section describing the comparison with observations.

6. Finally, the paper needs to discuss the impacts of its findings. With the present day mass loss from Antarctica on the order of 100 Gt per year, this is quite concerning finding the differences in SMB from RCM choice alone are several hundred Gt per year.Please contextualize the findings in regard to how we can measure the mass balance of the ice sheets.

We have added a paragraph in response to reviewer 1's comments along these lines, where we relate the modelled SMB to the latest analysis of Antarctic mass budget from altimetry and GRACE observations:

"It is interesting to compare our results with those used in the IMBIE study of Antarctic mass budget (Shepherd et al., 2018). When taking into account the published uncertainties on the observational mass budget estimates from the input-output method, only the COSMO-CLM and MetUM estimates are outside the range defined by the IMBIE study based only on altimetry and GRACE data. However, as these two models, particularly MetUM, perform well in comparison to meteorological observations, the source of the mismatch is unclear and an area that requires significant future work. It may also indicate either that some of the components of SMB are poorly captured by the models or that there are compensating errors in the modelled SMB components and/or their spatial variability. Nevertheless it is therefore also important to consider the wide uncertainties in both observations and the likely biases in models discussed in this paper, in assessing the contribution to sea level rise from Antarctica"

Minor Comments
Several model names and versions are discussed before they are described, which makes it quite hard to follow. Please reorder the sections to ease.
For instance, section2.1 and the end of Section 1 mention several models and different version, but there is no description, so it's hard for the reader to follow. It would also be appropriate to cite the papers that refer to these model versions.

We have reordered and expanded this whole section to make it easier to read and to follow which models are under discussion and how they relate to each other and to give further details on the different schemes and parameterisations.

P1, Line 7: Is this for grounded ice only? Does it include islands and ice shelves?
This was for the whole ice sheet including ice shelves. We have added a paragraph in the introduction discussing how SMB is calculated and the differences between grounded ice sheet and ice shelves to clarify our ice mask definitions. The abstract has been completely rewritten to summarise the conclusions of the paper more effectively.

P1, Line 7-8: What do the values after the ± represent? The standard deviation of all the models?

The values represent the standard deviation of the annual ensemble mean including all models, but see comment above also.

P1, Line 10-11: Why is 1980-2010 chosen as the climatological period? Later in section 2.3.3, it appears that 1987-2015 is the common modeling period that is used to"compute a climatological mean" (P10, Line 12). Please rectify.

We realise that it is confusing that the models were run for slightly different periods and this also makes comparisons between them more complicated. We have added a paragraph explaining the different periods to the Methods section and clarified all the way through the paper which periods are being used during the results and discussion. The abstract has been rewritten to reflect this also.

P2, Line 1: change "compar" to "compare"

Fixed

P2, Line 11: remove either "potentially" or "potential" since its repetitive

Fixed

P2, Line 13: add "and" after "2002,"

Fixed

P3, Line 16: remove the comma after "published"

Fixed

P3, Line 23: remove "drive"

Fixed

P4, Line 1: please describe what a "reinitialized hindcast" is

A reinitialised hindcast is a model run in weather forecast mode that is reinitialised by observations every 48 hours. We have added a line to explain this.

P4, Lines 5-7: While this is true, it might have a limit. See Lenaerts et al., 2018, which shows that often the snow is not dumped in the proper place when moving from 27 km to 5.5 km. Please add a sentence clarifying this.

This is a good point and in fact one of the reasons we have undertaken this comparison. We have added a line mentioning this
"Lucas-Picher et al. (2012); Lenaerts et al. (2012b); Franco et al. (2012); van Wessem et al. (2018) among others have found that a higher spatial model resolution gives more physically plausible results, especially with respect to precipitation processes

in areas with steep terrain. However, there is also evidence that moving to high resolution (~5.5km) can lead to precipitation falling in the wrong place due to e.g. upslope effects (e.g. Lenaerts et al., 2018; Schmidt et al., 2017)."

P4, Line 8: add "is used" after "ensemble mean"
Fixed

P5, Line 6: change "developed in" to "developed for"
Fixed

P5, Line 18: the end of this sentence needs to be reworded
Edited to:
"Although the mesoscale version includes a multi-layer snowscheme (Walters et al., 2019), in these simulations we used a simplified single-layer scheme with for example, no refreezing (Cox et al., 1999). SMB was calculated based on output precipitation and sublimation and evaporation."

P6, Line 4: Do you mean "processes" not "process"?
Fixed

P6, Line 22: change "includes no" to "does not include"
Fixed

Table 1: What does SMB scheme mean?
SMB scheme refers to whether or not the regional climate model has been modified to take into account atmosphere - ice sheet interactions, or if it is run in a standard mode without explicitly calculating SMB.

P7, Line 10: change "schemes" to "scheme"
Fixed

P10, Line 10: add "that are" after "2015 and 2018"
Fixed

P10, Lines 14-20 need clarification
We have added more explanation of the technique used to make the comparison between dataset and observation and added more detail on the dataset used.

P12, Lines 4-7: This sentence is very long and needs to be split in two.
Fixed

P12, Line13-14: Please reword the sentence as its confusing.
Fixed

P13, Line 6: Remove "In"
Fixed

Figure 3. Please add the statistics to these plots (RMSE, etc.). Also, its very difficult to distinguish the colors here. Maybe large dots would help.

The statistics in these plots are given in Table 2 for clarity, we also present the models separately in the supplement and (in response to a request by reviewer 1) we have expanded this analysis to include a relative RMSE plot that we intend to include in the supplementary material (see above)

Figure 4. This figure should be much bigger. It's very hard to see the colors. Also, in the caption there are "a", "b", etc., but they do not exist on the plots.

Figure 4 has been revised and enlarged to make it easier to read, we have removed the superfluous letters from the caption.

P18, Lines 9-11: are these values consistent with what is listed in the abstract?

We have revised the text of the manuscript to be more clear about which periods the SMB figures refer to. The abstract has been completely rewritten to simplify it and summarise the conclusions further

Figure 5. This needs to be in landscape orientation. The numbers are much too small to read.

We have edited this figure to make it larger and the labels clearer, however we prefer to keep this orientation as it makes it easier to read and interpret the figure when printed.

P20, Lines 5-6: What does "much clearer mean," please clarify

The topography in the regions noted in the text have a substantial influence on the modelled SMB, this allows physical features such as the Transantarctic mountains to be picked out in the SMB maps. We have updated the sentence to reflect this.

Figure 6. Again, these plots are too small, and the numbers are nearly impossible to read.

Figure 6 has been made bigger and restructured for ease of reading with larger font on the labels.

P22, Line 2: remove "below" There should be significance values associated with the trends. It looks like none would be statistically significant and thus are effectively no different than zero.

We fully agree about the lack of significance of the trends and have added a paragraph discussing this point as discussed above.

P24, Line 6: remove "very"

Fixed

P24, Line 9: change "bring" to "brings"

Fixed

P25, Line 7: Should the interval be 1987-2015?

Yes, here we discuss the overlapping period.

P27, Line 29-30: Do your results actually support "Models that have not undergone specific adjustments for Antarctica clearly represent the SMB in Antarctica more poorly".Look at the RMSE value in Table 2, it looks like sometimes they perform better. Please Clarify.

As discussed above, assessing how well the models perform is complex. The new figures discussed above help to clarify this somewhat but we have modified the text here to take into account the spatial and process variability.
"Models that have not undergone specific adjustments for Antarctica clearly represent the SMB in Antarctica more poorly than those that have been adjusted in some regions . However table 2 shows this is not unambiguous as in some elevation bands the unmodified models have lower bias and RMSE (see section 3.3).

P 28, Line 11-12: please give values in Gt of these processes to show that they are effectively negligible
Fixed

P 28, Line 20: add "fig." before "7"
Fixed

P28, Lines 20-21: the sentence needs to be improved.
Modified to: "The added value from a higher resolution model is that it better captures local topography and associated weather phenomena that in turn leads to more representative outputs."

P29, Line 2: replace "mod-latitudes" with "mid-latitudes"
Fixed

---

## Author Response (AR1)

Response to Review of Mottram et al: "What is the Surface Mass Balance of Antarctica? An Intercomparison of Regional Climate Model Estimates"

Review 1:

By Tessa Gorte and Jan Lenaerts, University of Colorado Boulder.

Mottram and co-authors present an intercomparison of different regional climate models regarding their performance in simulating Antarctic Ice Sheet surface mass balance. They show that the RCMs, all forced by ERA-Interim at their boundaries, show overall satisfying (but to a varying degree) correspondence with available weather and SMB observations, and that many remaining biases are common between the different models. The integrated ice sheet SMB varies widely from model to model, but interannual variability is very similar. Overall, we think that this is an interesting paper, containing relevant and important results for the climate, SMB, and ice sheet modeling communities, and very fitting for potential publication in The Cryosphere.

However, this paper lacks a bit of context and broader impacts in its current form, and it suffers from some internal inconsistencies, ambiguities, and poor figure and language quality in places. We would invite the authors to consider our general and more specific comments, highlighted below.

We thank the reviewers for their very considerate and thoughtful review, we agree with many of their comments and in the process of addressing these we feel that the paper has been considerably improved.

**General comments**

First of all, in many places it is not clear to what the ice sheet integrated SMB numbers refer to, i.e. grounded ice sheet or full ice sheet (including ice shelves)? That's an important issue to improve, not only to enhance clarity, but also since the former is directly translatable to sea level equivalent, the latter is not. An obvious place to start is the abstract (e.g. page 1, line 7 and 11). Using appropriate labels and explanations, and clearly separating grounded and full ice sheet throughout the paper would be essential.

This is a very good point and we have added a paragraph clarifying the difference between grounded SMB and SMB on ice shelves in the introduction section. The abstract has been completely revised and rewritten to reflect this and several other points raised by reviewers. Where SMB is discussed throughout the paper we clarify if we refer to the whole continent including ice shelves or only the grounded part. See below:

p2, line 25 "It is important to distinguish between the continental  grounded ice sheet and ice shelves when considering values for SMB integrated over a wider area whether regional or continent wide. Snowfall and melt on ice shelves is not directly relevant to sea level rise contributions as they are already floating but precipitation and ablation on grounded parts of the ice sheet is.  As the models used in this study by and large do not distinguish between

grounded and floating ice in their ice masksm in this paper, when we refer to SMB over an area we include ice shelves unless otherwise specifically noted."

Second, although we understand that the authors want to refrain from 'ranking' the models, we would argue that, based on the input-output method of determining mass balance (in e.g. the IMBIE assessments), one could qualify the new RACMO2 and MARv3 models more realistic than other models. Using other models would draw a completely different picture of AIS mass balance; based on Table 3, using e.g. COSMO-CLM would more than double current AIS mass loss, and HIRHAM would suggest AIS mass gain, both of which cannot be reconciled with other methods that determine AIS mass balance (GRACE, altimetry, etc.). A discussion on this would strengthen the impact of this paper beyond a straightforward intercomparison, and inform the community on strengths and weaknesses of the different models.

The reviewers are correct that the aim of this study is not to rank the models. Our analysis shows that the different models tend to have different strengths both spatially and in terms of different processes in reproducing climate and weather in Antarctica. However, it is also an important point that the modelled SMB should be consistent with observational constraints from the input-output method and we have therefore explored this further. We have added a new short section in the discussion where we analyse the model output on the same ice sheet mask and over the same time period as that used in the IMBIE (Shepherd et al., 2019) study of Antarctic mass budget and discuss the implications. Our analysis shows that, given the published uncertainties on the observational estimates from the input-output method, the COSMO-CLM and MetUM estimates are outside the range defined by the IMBIE study based on altimetry and GRACE data. However, as these models, particularly MetUM, perform well in comparison to meteorological observations, the source of the mismatch is less clear and indicates that some of the components of SMB are being poorly captured by the models and/or that there are compensating errors in the modelled SMB. This is an important point and we have therefore also included it in the discussion as below and in the summary conclusions.

P27, L17 "It is interesting to compare our results with those used in the IMBIE study of Antarctic mass budget \citep{shepherd2018}. When taking into account the published uncertainties on the observational mass budget estimates from the input-output method, only the COSMO-CLM and MetUM estimates are outside the range defined by the IMBIE study based only on altimetry and GRACE data. However, as these two models, particularly MetUM, perform well in comparison to meteorological observations, the source of the mismatch is unclear and an area that requires significant future work. It may also indicate either that some of the components of SMB are poorly captured by the models or that there are compensating errors in the modelled SMB and ice dynamic components and/or their spatial variability."

Thirdly, many of the figures are very difficult to read, and colors showing different models are difficult to separate. Moreover, the figures could use a bit more explanation in the text as well as

in the caption. A lot is left to the reader to decipher these figures (which potentially convey very interesting information).

We agree that more explanation of the figures is necessary and as well as revising them to make them clearer we have added additional explanatory text for each throughout the paper.

Lastly, language needs to be improved throughout. A few places are consistently lacking commas: after/around thus, therefore, moreover, etc. Several sentences were a bit long and could be broken up to make them easier to read. The authors switch between active and passive voice quite often throughout the text (i.e. "parameterizations are included" instead of "the models include parameterizations), suggesting that various authors have contributed to the writing and the end result is somewhat heterogeneous. We have pointed out a few locations below, but there are many more in the paper. Try to avoid phrases like 'clearly' throughout the paper. This is a subjective statement, and findings may not be so clear to the reader as it is to the authors.

A multi-author paper of this type is indeed vulnerable to inconsistent language and we have therefore proof-read and thoroughly revised all text and reverted all passive voice to active to make the paper more readable. We have also removed more subjective language and (also with reference to Reviewer 3's comments) tightened up the statistical basis of statements where necessary.

Specific Comments

P1: Why are SMB and Gt given as abbreviations but not AIS which is abbreviated later?

We have added the AIS acronym and made use consistent throughout.

P1L1-2: Technically, Antarctica loses mass through enhanced ice discharge across the grounding line into ice shelves (not compensated by an increase in SMB), and ice shelves lose mass by enhanced calving and basal melt (not compensated by an increase in ice shelf SMB and/or solid ice influx). Separating these various processes can help to separate the grounded and full ice sheet (see General Comment 1).

We have revised the abstract considerably to make it shorter and easier to read, the separation of the mass budget components, including this point has now been included in the introduction section (see first comment above).

P1L3-4: "... of crucial importance..." → "crucially important"

Removed - see previous comment

P2L1: "compar" → "compare"

Fixed

P2L12: "... a significant part of the climate system" is a bit vague and could be expanded upon

Adjusted to: "The Antarctic Ice Sheet (AIS) is the largest body of freshwater on the planet and thus a potentially important contributor to global sea level rise as well as a significant part of the climate system contributing freshwater to the ocean and with it's high relief significantly affecting atmospheric circulation."

P2L15: Is "submarine melting" a common phrase for basal melting?

We use submarine melting here to distinguish from basal melting at the bed of the ice sheet generated by e.g. geothermal flux, friction processes etc.

P2L20-21: Scambos and Shuman maybe shouldn't be in all caps.

Fixed

P2L27-28: "In the future... climate change" → this sentence requires a change in punctuation for readability for me. For instance, consider changing to "In the future, a "greenlandification" of the ice sheet climate (increased melt and refreezing within the snowpack) is projected due ..."

Changed to: "In the future, a "greenlandification" of the ice sheet climate is projected due to anthropogenically induced climate change \citep{trusel2018nonlinear}. This will lead to more melt with more refreezing in the snowpack as well as increasing runoff."

P3L12-16: "Souverijns et al... peer review literature" → this is quite a long sentence. Perhaps consider breaking it up for readability.

Changed to 2 sentences "In the polar regions, CORDEX simulations can also be used to assess the mass budget of the large polar ice sheets, but have not yet been evaluated together for Antarctica. \citet{Souverijns2019} made a 30 years hindcast with COSMO-CLM$^{2}$, and \citet{Agosta2019} estimated the SMB using MAR, while various versions of RACMO2 have been used to estimate the SMB of the AIS  \citep{van2014improved, VanWessem2018}. Both MetUM and HIRHAM5 have been run for the Antarctic domain but evaluation of the SMB results have not yet been published in peer review literature \citep{hansen2019}"

P3L27: It might be good to list all 5 RCMs at the beginning of the Methods section

Added in brackets on first line.

P7L4: "Parameterizations are included..." → "The models include parameterizations..."

Fixed

P8L2: "... nudging whether spectral or with simpler techniques keeps..." → "nudging, whether spectral or with simpler techniques, keeps..."

Fixed

P9L6: "Weather observations are used..." → change to active voice

Fixed

P9L22-27: Change paragraph to active voice

Fixed to : "As the different models have different ice masks and topographies we only retain stations on the common mask where the difference in elevation is lower than 500 m for each model, this gives a total of 184 AWS (See the supplementary material for locations of AWS used in this study). We compute the modelled surface pressure, near-surface temperature and wind speed, as well as the model elevation, using a four-nearest inverse-distance-weighted method. Finally, since the measurement height is not known for every station, we use the vertical level closest to the surface (10 m or 2 m) of the models for all comparisons with the observations."

P10L10-11: "Observations between... 5 years" → consider rephrasing for readability

Fixed to "Observations between 1950 and 1987, or 2015 and 2018 that are not fully included in the common modelling period of 1987 to 2015, were used for evaluation only if they covered more than 5 years."

P10L15-20: Authors say SMB was computed in 3 steps but only two seem to be explicitly mentioned.

Typo, FIXED

P11L10: So you're saying that the higher resolution the model, the poorer skill it will show due to increased internal variability? Please clarify, since this is essentially contradicting many other studies that are suggesting enhanced performance with resolution.

The main issue here is that the Antarctic domain is very large, without nudging or relaxation the higher resolution models have many more degrees of freedom to evolve, we have clarified this here:

P12, L4 "Without nudging, the large domain size in Antarctica means that synoptic scale systems have more degrees of freedom to evolve away from the observed quantities. This is likely to be a particular problem for higher resolution models where there are more grid points between the boundary and a given station, compared to a lower resolution model with fewer grid points. Our results show that the high resolution (0.11\degree) version of HIRHAM5 that has many more grid cells than the low resolution (0.44\degree) version has a higher divergence due to internal variability. MetUM is not nudged by surface relaxation but is run in daily reinitialisation mode and while this probably also helps to keep surface pressure close to observed it is also likely that the large number of atmospheric levels in MetUM also improves modelled surface pressures."

P11L17: What causes you to suspect "...biases in cloud cover and long-wave radiation..." are the leading factors in divergence from observation? How would you expect a model bias that overestimates cloud cover to effect observations, for example?

The analysis of Van Wessem et al 2014 shows that significant improvement of the RACMO2.3 model was derived from improved cloud microphysics parameterisations. We have clarified this further.

P13 L2 "However, biases in cloud cover and long-wave radiation reaching the surface are likely the main explanation for divergence from observations and should be investigated for all RCMs run for Antarctica as shown by \citet{vanWessem2014}. IN theri study, significant improvements in the RACMO2.3p2 model were obtained by adjustments to the cloud microphysics."

P12L7-8: "The models can be divided into two groups..." → how are you dividing these groups? Not sure we understand the origin or the purpose of having two groups here.

Here we were referring to a visual contrast in the placement of the models on the Taylor diagrams. We have clarified it to "The models appear to fall in two groups on the Taylor Diagram"
P12L22: Extra parenthesis.

Fixed

P12L23: "...in the colder, and therefore higher elevation locations, while..." → is this supposed to be "...in the colder, and therefore higher elevation, locations while..."? Also, perhaps consider changing the order to "in the higher elevation, and therefore colder, ..." such that it seems like temperature is a function of elevation and not the other way around.

Fixed to: "the other models overestimate temperature in the higher elevation, colder locations, while underestimating temperature at lower elevations in the coastal regions"

P14L14: What do you mean by "good results" exactly?

In this case we mean that compared to the other models, RACMO2.3p2 has a lower bias in the SMB and a higher correlation, however as the word good is unclear we have changed the sentence to read

P15 L9 "The blowing snow module included in RACMO2.3p2 may explain the lower bias in results between 0 and 1200 and especially 1200 and 2200 m (Table \ref{tab:samba_smb} b and c), compared to the other models."

P16L2: "...we here show..." → "...here we show..."

Fixed

P17, Table 2: Arguable showing an RMSE with absolute SMB numbers that decrease rapidly from the coast to the interior is not justified, since the RMSE will tend to decrease along with the SMB itself. Adding relative RMSE (i.e. as a ratio to the mean) is required to compare apples to apples across the elevation bins.

This is a fair comment, we have updated table 2 to include the relative RMSE for each model by elevation bin, however it does not alter our results substantially. We propose

to add the following new plot showing this relative RMSE for each model in the supplementary information as it shows visually how the percentage RMSE varies for each model according to different elevation bins.

[Figure]

P20: When looking at the ensemble mean, have you considered how your results may change if you calculate the mean on different grids? What grid did you use for this (i.e. how does this common grid resolution compare to that of any of the given models)?

We computed the ensemble mean SMB of the 9 models using each model's own grid. First we calculate the basin averaged SMB for all the models on their own grids and then we take the common grid points that fall within the defined basins. This means that numbers are independent of the model grids, and can be averaged into an ensemble mean.

We opted to use the RACMO2.3p2 grid to present the ensemble mean as it is an intermediate resolution for all the models and we compare it with the Shepherd et al 2019 study that also used this grid. We have updated the caption to reflect this.

"Table 3.Integrated mean annual SMB for the six models used in this study, for the period 1980 to 2010 except for COSMO-CLM2 where the period was 1987 to 2010. Three older model versions, ensemble mean and standard deviation as shown in Figure 5. All calculations done on

the original grid of the individual models using a common set of drainage basins and ice mask defined by IMBIE2 Shepherd et al. (2018b). The ensemble mean was calculated by transforming all models to the RACMO2.3p2 grid. GIS denotes grounded ice sheet, IS denotes ice shelves and ToTIS denotes the full Antarctic ice sheet including ice shelves"

P24L9: "bring" → "brings"

Fixed

P25L9-11: "The HIRHam5 ... below the mean respectively" → This sentence is long and difficult to read due to the lack of commas.

Fixed to read:

P27 L9 "The HIRHAM5 0.11∘ and MARv3.10 numbers are almost exactly the same at 2452 Gt and 2445 Gt respectively around 150 Gt above the mean. MetUM, like COSMO-CLM2, is much lower at about 138 Gt and 368 Gt below the mean respectively."

P25L17-20: The authors address the period of the "1990s and 2000s" for SMB trend, but since SMB is so highly variable, can you really say that this is significant/important?

This is actually one of our main points, that it's almost meaningless to suggest significant trends over short periods given the large variability which Figures 7-9 clearly show. As all reviewers have a similar comment here we have added a section in the discussion where we explicitly state this.

P28 L4 "Unlike previous studies, we detect no obvious strong trend in the modelled SMB in any of the models or in the driving ERA-Interim model. Shorter periods within the time series appear to have quite strong trends, for example a steady declining trend is apparent through the 1990s and 2000s but appears to have reversed since 2014. Our results suggest that strong interannual and decadal variability makes the identification of meaningful trends over short periods very difficult. Distinguishing noise from signal will be challenging in coming decades and this also emphasises the importance of long time series of observations."

P26L16: "west Antarctica" and "Antarctic peninsula" → "West Antarctica" and "Antarctic Peninsula"

Fixed

P27L29: "bee" → "been"

Fixed

Figure Comments:

Figure 1: These Taylor diagrams are a very interesting way to convey information, but many readers will have never seen something like this before. It will be important to better clarify the

metrics conveyed by the figure. For instance, we are unsure what the curved lines (i.e. ranging 1.60 to 13.50 in the left panel) are supposed to represent.

We have expanded the explanation of the Taylor diagrams in the caption and the analysis of the results in the Results section as below

"Figure 1: Taylor diagrams showing model performance compared to daily observations of surface pressure (a), near-surface temperature (b), and observed wind speeds (c).The horizontal and vertical axes represent the standard deviation, the dashed line in bold shows the standard deviation of the observations . The Taylor plot also shows the correlation which is measured by the angle with the x-axis. Finally, the CRMSE is represented by the curved lines in light grey. A perfect model should therefore be in the same place as the observations (black star, correlation of 1, same standard deviation, and zero CRMSE). Similarly, the further away a model is from the observations, the worse it is. Mean biases and observation mean are also indicated. The units of standard deviation, CRMSE, mean bias and mean of the observations are the same (hPa for surface pressure, K for near-surface temperature, and m/s for wind speed)."

P11L6 "In figure \ref{fig:taylor} we show Taylor diagrams for pressure, temperature and wind velocities. Taylor diagrams offer an efficient way to assess model skill by comparing the Pearson correlation coefficient, the centred root mean square error (RMSE) and the standard deviation of the modelled output with the observed values. CRMSE is equivalent to the RMSE but systematic biases are removed by subtracting the mean observation and mean modelled values from each value. A perfect model should be in the same place as the observations (black star, correlation of 1, same standard deviation, and zero CRMSE). The further away a model is from the observations, the worse it matches the observed weather. Mean biases and observation mean are also indicated. In this case modelled values closest to the dashed line have a more correct representation of the standard deviation and the closer to the black reference star the closer the model correlates to the observations values. We also list the bias below the diagrams."

Figure 2: Could you perhaps also include a table of correlation and/or bias for each model?

As the paper is already very long and we have also added substantial new material in response to the reviewers comments we don't want to add further tables or figures unless absolutely necessary. Figure 2 is not a central figure for understanding the paper and we have included Table 2 with substantial statistics relating to the SMB.

Figure 4: The same comment as figure 3, but with the color bars

There appears to be a comment about Figure 3 missing that makes this comment difficult to answer - we have however revised Figure 4 to make it easier to read along the lines suggested by the other reviewers

Figure 5: Is this meant to be rotated? Also, increase the font size again.

This figure has been deliberately rotated to make it easier to fit on the page. We have increased the font size.

Figure 6: Increase label size

The labels have been increased and the plot has been restructured and enlarged to enhance readability.

Reviewer 2

General Comments:

This paper presents results from a first intercomparison of polar regional climate models (RCM) applied in the Antarctic Ice Sheet (AIS). The model performances were compared and assessed in terms of surface pressure, near-surface air temperature, near-surface wind speed, surface temperature, and surface mass balance (SMB) of the AIS. The models that participate in this intercomparison project are COSMO-CLM2, HIRHAM5, MetUM, MAR, and RACMO. For some models, results from different versions are provided additionally.

We thank the reviewer for their thoughtful comments and have addressed in detail the points they raise below.

My first honest impression after reading through this manuscript is that the current title "What is the Surface Mass Balance of Antarctica?" is a bit misleading, because meltwater runoff is not considered in the most participating models except for MAR and RACMO. It is true that a contribution by runoff to the changes in the present- day AIS SMB is relatively small than contributions from precipitation and sublimation/evaporation. But, runoff in the present-day AIS already cannot be neglected as presented by several studies cited in this manuscript. In the future in a warming world, the contribution by runoff to the changes in AIS SMB will become much higher almost certainly as pointed out by the authors (P. 2, L. 18 ~ 19). Therefore, this reviewer expected that all the models calculated runoff in the present study, and as a result, I was a bit disappointed when I found the relevant description in Sect. 2.2.1.

Related to the point indicated above, the intercomparison procedure for SMB sounds a bit inadequate to me, because the authors employ different definitions of SMB (Sect. 2.2.1). If the authors focus intercomparisons only for precipitation and

sublimation/evaporation (in addition to the three surface meteorological properties as well as the surface temperature), it makes sense and highlights key differences in model physics employed by these participating models more clearly. This reviewer recommends the authors to reconsider the title of this manuscript: maybe something like "intercomparison of Antarctic ice sheet surface meteorological conditions simulated by five different regional climate models" would be appropriate. However, the intercomparison of RCMs performed in the AIS itself is a considerable new challenge, so provides the latest comprehensive information related to these RCMs, which is very informative for readers certainly, so deserved to be published

In this paper we focus on the Surface Mass Budget of Antarctica and the uncertainties introduced by using different regional climate models to calculate it, even when those models are forced by the same global model. Overall, precipitation dominates SMB to such an extent at the present day that even subtracting all runoff from the models that calculate melt and refreezing leaves the overall SMB virtually unchanged. The difference is negligible even on a basin scale. We agree that melt is likely to become more important in the future, but at the present day melt and runoff are only observed at a few very specific locations.

However, while all of the models simulate melt, they have varying degrees of complexity to calculate refreezing so purely to simplify the comparison here we focus on precipitation, evaporation and sublimation terms. Two of the models include sublimation from blowing snow subroutines, and as the physical parameterisations have been developed with these processes in the model, we have also used the drifting snow schemes in the results.

We absolutely agree that there are important questions around melt extent that have important implications for future SMB projections and we plan to extend this study with a detailed look at how the models simulate melt and runoff in a paper currently in preparation. For now, we have however added in the introduction and discussion sections more detail on how SMB is computed (see response to reviewer 1) and the processes that can be and are included and address the issue the reviewer raises in more details.

We note also that we do compare modelled SMB with measured SMB from stakes and from other studies such as the IMBIE study (see reply to reviewer 1 here also) and we think therefore it is justified to keep the title as it is.

**Specific comments (major)**

P. 2, L. 7 ~ 8: What kind of measurements do the authors think here (observational campaigns)? Maybe it is not necessary to indicate explicitly here; however, please suggest something at least in the discussion and/or conclusion sections.

Our results suggest that in particular stake measurements of SMB are crucial. These need to cover locations where there are very few recent observations and where there are large disagreements between the models. We propose adding a new figure (see below) to the supplementary materials that show the locations of SMB observations as well as locations of weather stations in order to demonstrate the significant data gaps. We have made this clearer in the conclusions.

[Figure]

Figure A8. Location of automatic weather stations and SMB observations in Antarctica and used in this study

P. 12, L. 2 ~ 3: What kind of physical mechanisms do the authors think here? Please detail more.

In locations with melt we expect that the lack of refreezing will affect the latent heat release in the snowpack which will in turn affect observed 10m depth temperatures. Furthermore, the diffusion of and conductivity of the surface snow layers is affected by the presence or absence of ice layers and by density which is dependent on

densification schemes that this version of HIRHAM and COSMO-CLM2 do not have. We have clarified this to read:

P13, L2 "However, biases in cloud cover and long-wave radiation reaching the surface are likely the main explanation for divergence from observations and should be investigated for all RCMs run for Antarctica as shown by van Wessem et al. (2014). In their study, significant improvements in the RACMO2.3p2 model were obtained by adjustments to the cloud microphysics. Furthermore, the lack of detailed subsurface snow pack schemes including processes such as refreezing (and subsequent latent heat release) and densification also likely has an impact on the temperature bias in HIRHAM5 and MetUM (see also figure 2)"

P. 12, L. 5: How large do the authors think the uncertainties are?

It is very difficult to quantify uncertainties on these observations, particularly wind, as they are made at mostly automatic unstaffed stations and are subject to different biases depending on location from effects such as burial by snow, changes in orientation due to wind and breakdown of sensors among others.

Modified to:
P13, L8. "This is likely in part due to large uncertainties in the observations especially at unattended stations where burial by snow, changes in orientation and sensor breakdown are more likely. However, the effects of different resolution and differences in turbulent schemes between the models may also be important. In particular the extremely stable boundary layer over most of Antarctica is hard to represent in models particularly at lower resolutions"

P. 12, L. 5 ~ 6: Readers cannot know the difference in turbulent heat schemes, because they are not described in this manuscript.

A detailed description of all of the models turbulent energy schemes is beyond the scope of this paper, we have however added relevant references to these for each model to support our interpretations of model biases in an expanded section 3.1.

P. 24, L. 13 ~ 15: It is interesting to see the model-simulated precipitation integrated over the common ice sheet mask by COSMO-CLM2 tends to be lower than that by the parent data ERA. It is because precipitation in a dynamically downscaled data is higher than precipitation in its parent data in general. Please discuss.

Our analysis suggests that the COSMO-CLM2 model used in this simulation has indeed a dry bias compared to the other models and the reasons behind this are the subject of ongoing work. The bias was first identified by Souverijns et al., 2019 and seems in part

to be a consequence of a particularly dry bias on the coast, especially in the peninsular and west Antarctica but there is conversely an overestimate of precipitation in the interior. We have added these details to the paper.

P30, L7. "The driest model COSMO-CLM2 underestimates SMB close to the coast, a region very relevant for total ice sheet mass balance. This is due to an overestimated sublimation amplified by an underestimated snowfall rate close to the coast. High values for the sublimation originate from an underestimated albedo due to aging of the snow that occurs too fast in the model (Souverijns et al., 2019). The low values for the snowfall rate is likely related to cloud microphysics, namely a too slow conversion of ice to snow or a too slow deposition of water vapor on the solid hydrometeors. Currently efforts are ongoing to improve the coastal SMB performance in COSMO-CLM2."

P. 29, L. 27: Please suggest what kind of measurements do the authors think necessary in the "observational campaigns"?

Given the importance of precipitation and snow processes to the SMB in Antarctica, stake and radar measurements, supplemented with firn cores are a clear priority. New observations should focus where possible in regions where there is a lack of measurements, but also where there is strong disagreement between models, as identified in the Results section. We have added these points explicitly to the conclusions.

P32, L6 "In particular, we argue that given the importance of precipitation for SMB, new observational programmes are needed that focus on accumulation and snow processes, e.g. stakes, firn cores and radar. Furthermore, focusing new observations in regions (see for example,A8 ) where there is both a lack of current data and strong disagreement between models will be valuable for understanding climate in Antarctica"

**Specific comments (minor)**

P. 4, L. 13 ~ 18: Is it OK to understand MAR 3.6 is older than MAR v3.10? If yes, it is a bit confusing isn't it?

MAR v.3.10 is the more recent version of MAR

P. 8, L. 6: What do the authors mean by "cloud physics"? To me, it is difficult to understand why "cloud physics" is resolved better in nudged models.

As nudged models better represent cyclones when compared to observations, the presence or absence of clouds is more likely to be closer to observed, however we

agree saying "cloud physics" is not quite technically correct so we have modified to "the presence of clouds"

P. 12, L. 24 ~ 25: It is not clear why the authors think so. Please explain more.

We have added the reference to van Wessem et al., 2014 and updated the text to:

P13, L2 "However, biases in cloud cover and long-wave radiation reaching the surface are likely the main explanation for divergence from observations and should be investigated for all RCMs run for Antarctica as shown by van Wessem et al. (2014). In their study, significant improvements in the RACMO2.3p2 model were obtained by adjustments to the cloud microphysics. Furthermore, the lack of detailed subsurface snow pack schemes including processes such as refreezing (and subsequent latent heat release) and densification also likely has an impact on the temperature bias in HIRHAM5 and MetUM (see also figure 2)"

P. 12, L. 31: "For the warmer coastal regions": From which data can we see this argument?

As this sentence is a bity unclear we have changed to

 P.14, L3 "For the lower elevation, mostly coastal regions most models have a cold bias."

Figure 2: This figure is a bit difficult to see. Please provide a table indicating ME, RMSE, and R2.

We have included table 2.

P. 23, L. 4 ~ 7: What is an interesting point here? I don't think the lower panel of Fig. 8 is necessary; however, if the authors think it is necessary, please discuss more about the figure. Maybe, inter-annual variations of these model results should be discussed more.

We have added more discussion in the section following Figure 8 to make explicit the finding that while all the models have the same anomaly when compared to their own mean, the sign of the anomaly compared to the ERA-Interim value can be different. Since the most highly constrained models show the lowest anomaly compare to ERA-Interim, we suggest that most of the variation is related to internal variability (weather) within the domain.

P25, Line 1 "Figure 8 emphasises the large variability in SMB on an annual to decadal scale by plotting the variation from the mean for each model and the variation from ERA-Interim for each model. This shows that while all the models have more or less the same anomaly when compared to their own mean, the sign of the anomaly compared to the ERA-Interim value can be different. Since the most highly constrained models show the lowest anomaly compared to ERA-Interim, we suggest that most of the variation is related to internal variability (weather) within the domain.Both HIRHAM5 0.11\degree and 0.44\degree shows the highest values of variability, probably due to the unconstrained nature of the runs, but in different years different models show higher variability than the others. The lower panel in Figure \ref{fig_SMB_trend} shows that MetUM is by far the closest to the driving model with much less variability than the others, HIRHAM5 again shows the highest difference compared to the driving model but from year to year the model showing maximum difference varies and there appears to be no systematic pattern as to whether or not modelled SMB is higher or lower than the ERA-Interim reanalysis when quantified on the common mask and over the whole of Antarctica. The implication is that while the driving model controls broad scale pattern of SMB, the downscaling model adds its own weather variability to the broad scale pattern. The variability, or weather noise is unsurprisingly, largest in un-nudged models. The effect of this noise on ice sheet dynamics may be small overall but as for example, Mikkelsen et al. (2018) show, small variations in SMB can have a non-negligible impact on ice sheet dynamics."

**Technical corrections**

P. 2, L. 1: "compar": typo

P. 2, L. 11: "a potentially important potential contributor" -> "an important potential contributor"?

Fixed

P. 2, L. 22: surface mass balance -> SMB; Note this term is already defined.

Fixed

P. 3, L. 22 ~ 25: This sentence especially after "and better understand drive sea level rise . . ." is a bit difficult to understand. Please reformulate it.

Fixed

P. 5, L. 11: "regional mesoscale model e.g." -> "regional mesoscale model as presented by e.g."?

Fixed

P. 7, L. 14: Define $SU_{ds}$ and $ER_{ds}$.

Fixed

P.9, L. 2: "GrIS": typo, right?

Fixed

C4P. 10, L. 27: "assessing" -> "assess"

Fixed

P. 12, L. 7: Indicate the publication year for Zentek and Heinemann.

Fixed

P. 12, L. 25: "downwelling longwave and surface albedo" -> "downwelling longwave radiation flux and surface albedo"

Fixed

P. 14, L. 18: "HIRHAM5.011": typo

Fixed

Review of Mottram et al., What is the Surface Mass Balance of Antarctica? An Inter-comparison of Regional Climate Model EstimatesSummaryThe authors present an intercomparison exercise of five different regional climate model surface mass balance estimates, as well as the near surface climate, over Antarctica. The authors find a large spread in total SMB (1961 to 2519 Gt year-1), which largely stems from differences in West Antarctica and the Antarctic Peninsula.Variability is quite consistent between models, which is unsurprisingly since they are all forced by ERA-Interim, but the trends differ in sign and magnitude and are quite sensitive to the time period selected.
Also, not surprisingly, the nudged models simulate the near-surface climate better as they are not allowed to deviate as substantially fromERA-Interim as the un-nudged models. Finally, the authors discover that the biases are typically consistent between models. The paper presents a significant amount of work but still requires improvements.

We thank the reviewer for their careful reading and thoughtful comments, which have led to significant improvements in the manuscript.. We address the specific points in the text below.

First, the manuscript has numerous mistakes throughout and needs refinement of the language in several places (see Minor Comments).

This point has also been addressed by the other reviewers and the whole manuscript has been thoroughly proof-read and made simpler to read and more consistent in language and structure.

More importantly, there are several major issues with the analysis that need to be addressed to improve the scientific rigor of the paper.

Major Comments

1. Throughout the manuscript, it is not clear what time periods are being used. There Is the common model interval, climatological mean, etc. The authors need to be very clear throughout the paper because it often seems that different intervals are being confused in nomenclature. It's not clear to me why the common reference intervals are not the common period between all models: 1987 – 2015. Throughout the paper sometimes its 1980-2010, 1987-2015, and 1987-2018. I recommend using the same interval through to avoid confusion. If the authors have a reason to use different intervals, then please make it clear what interval is being used. It is additionally unclear why 1980-2010 is representative of the climatological period, please explain.

We realise that it is confusing that the models were run for slightly different periods and this also makes comparisons between them more complicated. We have added a paragraph explaining the different periods to the Methods section and clarified all the way through the paper which periods are being used during the results and discussion when relevant.

P4. L13 "Unfortunately, as we are constrained to using existing simulations, the models cover slightly differing periods (see \ref{tab:model_overview} for details). We have therefore defined a common 30 year climatological period of 1980 to 2010 for all models to simplify the integrated mass budget comparison, except for COSMO-CLM$^{2}$ where the period covers 1987 to 2010. Figures that show time series of data show the full period relevant for each model."

2. Similarly, there is no discussion of significance for the statistics presented. There are claims within the text that certain models perform better than, but without significance levels, these claims lack strength and are more speculative.

Trends are discussed at both long (1987-2015) and short (decadal) time intervals, but the significance is never discussed. I would caution the authors' descriptions of trends, especially at short time scales, since it is very hard to observe a significant trend in SMB since its highly variable year to year.

We absolutely agree that detecting a significant trend in SMB is almost impossible and in fact that was one of the points of Figures 7-9. However, given all reviewers comments we have clearly not described this well enough. We have added a paragraph making this point explicitly and setting the SMB in context.

P28, L4 "Unlike previous studies, we detect no obvious strong trend in the modelled SMB in any of the models or in the driving ERA-Interim model. Shorter periods within the time series appear on first look to have quite strong trends, for example a steady declining trend is apparent through the 1990s and 2000s but appears to have reversed since 2014. Our results suggest that strong interannual and decadal variability makes the identification of meaningful trends over short periods very difficult. Distinguishing noise from signal will be challenging in coming decades and this also emphasises the importance of long time series of observations."

Furthermore, because this is an intercomparison paper, it's important for the authors to be very clear concerning the metrics of how they conclude one model outperforms the other. Is it RMSE? R2? Bias? And what is the threshold? Is an RMSE of 93 better than 97? What if one model performs differently at different elevation bands?
I did not find the argument compelling that the models tuned to specific Antarctic conditions outperformed the others because there was not a clear frame-work for comparison. The authors need to make clear the evaluation metrics and how they evaluate model performance, which will require more detailed statistical analysis throughout.

Model means are compared, but its not clear if the paper considers even a simple statistic of the standard error of the mean. The Student's t-test can be used to evaluate whether the means are different. Please be transparent with the limitations of the analysis and provide meaningful significance tests on all of the comparisons,otherwise the conclusions are speculative rather than significant.

This is a very important point and in part one of the drivers for this paper. We do not attempt to rank the models because it is clear from our results that on different measures, (bias, RMSE etc) the different models perform quite differently for different variables (both SMB and the meteorological variables). There is also a spatial component as the reviewer points out with different biases apparent at different elevation bands and in different locations. This means that most likely different models should be used for different purposes. It is also an aim of this paper however, to give clarity on exactly how the models compare, for which reason we give extensive statistics in figure 1 and table 2, which we have also expanded to

include the mean value of SMB observations for the elevation bands as requested by reviewer 1. As the paper is already very long, we have added 2 new figures and associated statistics in the supplementary section showing how the relative RMSE and mean bias compares between models for the different elevation bands. These for instance show that at high elevation COSMO_CLM, MAR and RACMO better represent SMB but HIRHAM, MetUM and COSMO-CLM have a lower mean bias in the middle elevations and MAR, HIRHAM and RACMO have a lower mean bias in the lowest elevations. In addition we have added some extra discussion comparing the different statistical methods and their use in evaluating the models in the discussion of table 2.

[Figure]

3. All of the RCMs presented are forced by the ERA-Interim reanalysis product. I find it concerning that there is no discussion of the role of using a single reanalysis to force all of the RCMs. Thus, this is not a definitive evaluation of the full range of possibilities in SMB, but rather a range due to RCM differences alone. There should be more discussion about how there would be additional spread due to varying choice of forcing; specifically, what is the impact of comparing models that are all forced by the same reanalysis. I think the paper needs to tone done the claims about the work providing the "likely range of SMB" in the first sentence of section 4.1, as it is more the likely range of RCMs forced by ERA-Interim. Basically, this explores the range in RCM space, but not reanalysis forcing space.

The point of this study is to determine the RCM uncertainty space rather than different boundary conditions. We specifically excluded models that ran different reanalyses as we would like to determine how different models compare with each other. However, having said that, analysis by Agosta et al., 2019 used different reanalyses to force the same model and found that the results were quite similar. We have added two extra sentences and this reference stating this in the methods section.

P3, L31. "All models were forced on the lateral boundaries with the ERA-Interim climate reanalysis (Dee et al., 2011) but downscaling used different grids, over slightly different

domains and at different resolutions with slightly different ice masks used in the different model versions (see A1 in the appendix). Simulations with MAR forced by different reanalyses (e.g. Agosta et al., 2019) found that results were rather similar to ERA-Interim. However, in order to exclude additional variability potentially introduced by using different boundary forcings, we chose to use a single common reanalysis only"

4. It's obviously quite a challenge to compare these models, which have differing levels of complexity. But it seems that the comparison would be better suited by comparing all the variables consistent between models (Precip-Evap-Subl). Otherwise, an inter-comparison doesn't shed much light on direct model to model differences. In fact, it appears that the authors could investigate whether these Antarctic specific physics actually provide improvement, which would be of great interest to the community. Therefore, the paper should do an ideal comparison of all 5 models with common variables (P – E – S) and evaluate performance. Then evaluate the models with extra physics (RACMO/MAR) to see if and how much model performance improves. Otherwise, it is difficult to untangle whether those additional processes provide any more information.

We have also addressed this point in our response to the second reviewer. Melt is likely to become more important in the future, but at the present day melt and runoff are only observed at a few very specific locations. However, while all of the models simulate melt, they have varying degrees of complexity to calculate refreezing so purely to simplify the comparison here we focus only on the precipitation, evaporation and sublimation terms. Then, RACMO and MAR include sublimation from blowing snow subroutines, while it would be ideal to separate these out, the physical parameterisations have been developed and tuned with these processes in the model, so it is difficult to remove them without negatively and unfairly affecting the results- we have therefore also used the sublimation from snow schemes in calculating SMB. We note that both RACMO and MAR groups have published articles demonstrating the improvement from the enhanced snow schemes (Van Wessem et al and Agosta et al., 2019). We have added extra text to make this point in the description of the SMB in the methods section.
P4, L26 "As the RACMO and MAR models have been developed to include the wind blown snow sublimation terms, they cannot easily be removed without retuning the models, and for this reason we have opted to include these within the SMB calculation for these two models."

5. The manuscript needs to justify the use of SMB observations starting in 1950.There are regions of strong trends in snow accumulation that might end up biasingthe comparison. If the issue relates mainly to reducing the number, the authors could present a comparison of only coincident SMB observations with the data, but then also provide the more liberal comparison as it currently exists in the text.

Unfortunately there are relatively few observations in Antarctica and including only those that were taken during the period of the simulations would make the model - observation comparison less robust. Including observations taken from 1950 increases the number of observations available to 923 comparisons from 469. More importantly, while the total number still sounds substantial, the main benefit is in fact in spatial representativeness. The 1987-2015 observations cover only a very small part of Antarctica. We discuss this problem in section 2.3.3 but we have expanded the discussion and we propose to include a new figure showing the spread of observations by date in the supplementary material. As we point out in the discussion and conclusions, the difficulty is also that the places where the models disagree most are also the areas with the sparsest observations of SMB.

We have mitigated the problem of unrepresentativeness as much as possible by for example excluding observations before 1987 that cover too short a period (less than 5 years) in order to keep only observations representative of a mean climate. However, we are not immune to introducing biases because these observations include biases arising from trends that the models cannot represent, nonetheless the comparison is more robust if it represents a larger area. We have added these points in the expanded methods section.

P10, L14 "To evaluate the models, we selected observations of SMB belonging to the common ice mask and for which the measurement period began after 1950 to 2018. These conditions reduced the total number of observations used in the comparison to 3671. We used the observations between 1950 and 1987, or 2015 and 2018 that are not fully included in the common modelling period of 1987 to 2015, for evaluation only if they covered more than 5 years. These 1849 SMB observations are compared to modelled values averaged over the common modelling period in order to compute a climatological mean while we averaged modelled SMB values over the exactly same period for the observations between 1987 to 2015 (1822 observations). Since the models have different resolutions and grids, we do not directly compare the modelled SMB values to the observations."

6. Finally, the paper needs to discuss the impacts of its findings. With the present day mass loss from Antarctica on the order of 100 Gt per year, this is quite concerning finding the differences in SMB from RCM choice alone are several hundred Gt per year.Please contextualize the findings in regard to how we can measure the mass balance of the ice sheets.

We have added a paragraph in response to reviewer 1's comments along these lines, where we relate the modelled SMB to the latest analysis of Antarctic mass budget from altimetry and GRACE observations:

P27, L17 "It is interesting to compare our results with those used in the IMBIE study of Antarctic mass budget (Shepherd et al., 2018). When taking into account the published uncertainties on the observational mass budget estimates from the input-output method, only the COSMO-CLM

and MetUM estimates are outside the range defined by the IMBIE study based only on altimetry and GRACE data. However, as these two models, particularly MetUM, perform well in comparison to meteorological observations, the source of the mismatch is unclear and an area that requires significant future work. It may also indicate either that some of the components of SMB are poorly captured by the models or that there are compensating errors in the modelled SMB components and/or their spatial variability. Nevertheless it is therefore also important to consider the wide uncertainties in both observations and the likely biases in models discussed in this paper, in assessing the contribution to sea level rise from Antarctica"

Minor Comments
Several model names and versions are discussed before they are described, which makes it quite hard to follow. Please reorder the sections to ease.
For instance, section2.1 and the end of Section 1 mention several models and different version, but there is no description, so it's hard for the reader to follow. It would also be appropriate to cite the papers that refer to these model versions.

We have reordered and expanded this whole section to make it easier to read and to follow which models are under discussion and how they relate to each other and to give further details on the different schemes and parameterisations.

P1, Line 7: Is this for grounded ice only? Does it include islands and ice shelves?
This was for the whole We have added a section discussing SMB and discharge and the differences between grounded ice sheet and ice shelves to clarify our ice mask definitions.

P2, L25 "It is important to distinguish between the continental grounded ice sheet and ice shelves when considering values for SMB integrated over a wider area whether regional or continent wide. Snowfall and melt on ice shelves is not directly relevant to sea level rise contributions as they are already floating but precipitation on grounded parts of the ice sheet is. In this paper when we refer to SMB over an area we include ice shelves, unless otherwise specified as the models used in this study by and large do not distinguish between grounded and floating ice in their ice mask"

P1, Line 7-8: What do the values after the $\pm$ represent? The standard deviation of all the models?
The values represent the standard deviation of all models and this has been clarified in the paper.

P1, Line 10-11: Why is 1980-2010 chosen as the climatological period? Later in section 2.3.3, it appears that 1987-2015 is the common modeling period that is used to"compute a climatological mean" (P10, Line 12). Please rectify.

We realise that it is confusing that the models were run for slightly different periods and this also makes comparisons between them more complicated. We have added a paragraph explaining the different periods to the Methods section and clarified all the way through the paper which periods are being used during the results and discussion (see reply to comment 1)

"Unfortunately, as we are constrained to using existing simulations, the models cover slightly differing periods (see \ref{tab:model_overview} for details). We have therefore defined a common 30 year climatological period of 1980 to 2010 for all models to simplify the integrated mass budget comparison, except for COSMO-CLM$^{2}$ where the period covers 1987 to 2010. Figures that show time series of data show the full period relevant for each model."

P2, Line 1: change "compar" to "compare"
Fixed

P2, Line 11: remove either "potentially" or "potential" since its repetitive
Fixed

P2, Line 13: add "and" after "2002,"
Fixed

P3, Line 16: remove the comma after "published"
Fixed

P3, Line 23: remove "drive"
Fixed

P4, Line 1: please describe what a "reinitialized hindcast" is
A reinitialised hindcast is a model run in weather forecast mode that is reinitialised by observations every 48 hours. We have added a line to explain this.

P4, Lines 5-7: While this is true, it might have a limit. See Lenaerts et al., 2018, which shows that often the snow is not dumped in the proper place when moving from 27 km to 5.5 km. Please add a sentence clarifying this.

This is a good point and in fact one of the reasons we have undertaken this comparison. We have added a line mentioning this and also included this point in the discussion section P4, L9 "Lucas-Picher et al. (2012); Lenaerts et al. (2012b); Franco et al. (2012); van Wessem et al. (2018) among others have found that a higher spatial model resolution gives more physically plausible results, especially with respect to precipitation processes in areas with steep terrain. However, there is also evidence that moving to high resolution (~5.5km)

can lead to precipitation falling in the wrong place due to e.g. upslope effects (e.g. Lenaerts et al., 2018; Schmidt et al., 2017)."

P4, Line 8: add "is used" after "ensemble mean"
Fixed

P5, Line 6: change "developed in" to "developed for"
Fixed

P5, Line 18: the end of this sentence needs to be reworded
Edited for clarity
P5, L26"Although the mesoscale version includes a multi-layer snowscheme (Walters et al., 2019), in these simulations we used a simplified single-layer scheme with for example, no refreezing (Cox et al., 1999). SMB was calculated based on output precipitation and sublimation and evaporation."

P6, Line 4: Do you mean "processes" not "process"?
Fixed
P6, Line 22: change "includes no" to "does not include"
Fixed

Table 1: What does SMB scheme mean?
SMB scheme refers to whether or not the regional climate model has been modified to take into account atmosphere - ice sheet interactions, or if it is run in a standard mode without explicitly calculating SMB.

P7, Line 10: change "schemes" to "scheme"
Fixed

P10, Line 10: add "that are" after "2015 and 2018"
Fixed

P10, Lines 14-20 need clarification
P10, L21 "As in Kittel et al. (2018) and Agosta et al. (2019), we compute modelled and observed SMB values in 2 steps. Firstly, the original resolution modelled SMB values were interpolated, as for AWS observations, to the observation location using a four-nearest inverse-distance-weighted method. Secondly, all the interpolated SMB values contained in the same grid cell from the common ice mask were averaged as well as the observations for finally creating 923 comparison pairs. This leads to a fair comparison for each model that takes into account the benefit of using a higher resolution for a specific model and removing the very high spatial variability of the observations that cannot be reproduced by the models."

P12, Lines 4-7: This sentence is very long and needs to be split in two.

Fixed

P12, Line13-14: Please reword the sentence as its confusing.
Fixed

P13, Line 6: Remove "In"
Fixed

Figure 3. Please add the statistics to these plots (RMSE, etc.). Also, its very difficult to distinguish the colors here. Maybe large dots would help.
The statistics in these plots are given in Table 2 for clarity, we also present the models separately in the supplement and (in response to a request by reviewer 1) we have expanded this analysis to include a relative RMSE plot that we include in the supplementary material.

Figure 4. This figure should be much bigger. It's very hard to see the colors. Also, in the caption there are "a", "b", etc., but they do not exist on the plots.
Figure 4 has been revised and enlarged to make it easier to read, we have removed the superfluous letters from the caption.

P18, Lines 9-11: are these values consistent with what is listed in the abstract?
We have revised the text of the manuscript to be more clear about when values are consistent with a given dataset. The abstract has been completely rewritten to simplify it and summarise the conclusions further

Figure 5. This needs to be in landscape orientation. The numbers are much too small to read.
We have edited this figure to make it larger and the labels clearer, and we have kept it in landscape orientation as it makes it easier to read and interpret the figure.

P20, Lines 5-6: What does "much clearer mean," please clarify
The topography in the regions noted in the text have a substantial influence on the modelled SMB, this allows physical features such as the Transantarctic mountains to be picked out in the SMB maps. We have updated the sentence to reflect this.

P22. L 4 "The figure shows quite substantial agreement between models over large areas of Antarctica but also some considerable local variability. Features such as the Transantarctic mountains and the rugged coastal topography in West Antarctica both of which substantially influence local weather patterns are picked out in the spatial pattern of the SMB. These features are more clearly delineated in the higher resolution runs."

Figure 6. Again, these plots are too small, and the numbers are nearly impossible to read.
Figure 6 has been made bigger and restructured for ease of reading with larger font on the labels.

P22, Line 2: remove "below" There should be significance values associated with the trends. It looks like none would be statistically significant and thus are effectively no different than zero.

We fully agree about the lack of significance of the trends and have added a paragraph discussing this point.

P24, Line 6: remove "very"

Fixed

P24, Line 9: change "bring" to "brings"

Fixed

P25, Line 7: Should the interval be 1987-2015?

Yes, good catch! Here we show results for the common model period 1987-2015

P27, Line 29-30: Do your results actually support "Models that have not undergone specific adjustments for Antarctica clearly represent the SMB in Antarctica more poorly".Look at the RMSE value in Table 2, it looks like sometimes they perform better. Please Clarify.

As discussed above, assessing how well the models perform is complex. The new figures discussed above help to clarify this somewhat but we have modified the text here to take into account the spatial and process variability.

P30, L4. "Models that have not undergone specific adjustments for Antarctica clearly represent the SMB in Antarctica more poorly than those that have been adjusted in some regions . However table 2 shows this is not unambiguous as in some elevation bands the unmodified models have lower bias and RMSE (see section 3.3)."

P 28, Line 11-12: please give values in Gt of these processes to show that they are effectively negligible

Fixed

P 28, Line 20: add "fig." before "7"

Fixed

P28, Lines 20-21: the sentence needs to be improved.

Modified to:

P30, L30 "The higher resolution version adds value with higher spatial variability that should better capture local topography and associated weather phenomena. This is especially important in areas of high relief such as in the coastal areas and around the Transantarctic Mountains. These are also the areas where models vary from each other and the ensemble mean the most. While there are very few observations to confirm the better performance on a local scale, the pattern of SMB suggests that the high relief rugged topography is better captured in HIRHAM5 0.11\degree than

0.44\degree. However, the higher resolution model is not only more computationally expensive, in a simulation where there is no nudging, like here, the larger number of grid points gives increased degrees of freedom for the model to evolve freely and thus introduces more internal variability.

P29, Line 2: replace "mod-latitudes" with "mid-latitudes"
Fixed

**What is the Surface Mass Balance of Antarctica? An Intercomparison of Regional Climate Model Estimates**

The Polar CORDEX Community[1], Ruth Mottram[1], Nicolaj Hansen[1,2], Christoph Kittel[3], Melchior van Wessem[4], Cécile Agosta[5], Charles Amory[3], Fredrik Boberg[1], Willem Jan van de Berg[5], Xavier Fettweis[3], Alexandra Gossart[6], Nicole P.M. van Lipzig[6], Erik van Meijgaard[7], Andrew Orr[8], Tony Phillips[8], Stuart Webster[9], Sebastian B. Simonsen[2], and Niels Souverijns[6,10]

[1]DMI, Lyngbyvej 100, Copenhagen, 2100, Denmark
[2]DTU-Space, Kongens Lyngby, Denmark
[3]Laboratory of Climatology, Department of Geography, SPHERES, University of Liège, Liège, Belgium
[4]Institute for Marine and Atmospheric Research Utrecht, Utrecht University, Utrecht, the Netherlands
[5]Laboratoire des Sciences du Climat et de l'Environnement, LSCE-IPSL, CEA-CNRS-UVSQ, Université Paris-Saclay, Gif-sur-Yvette, France
[6]Department of Earth and Environmental Sciences, KU Leuven, Belgium
[7]Royal Netherlands Meteorological Institute, De Bilt, the Netherlands
[8]British Antarctic Survey, High Cross, Madingley Road, Cambridge, UK
[9]UK Met Office, FitzRoy Road, Exeter, Devon, EX1 3PB, UK
[10]Unit Remote Sensing and Earth Observation Processes, Flemish Institute for Technological Research (VITO), Mol, Belgium

**Correspondence:** Ruth Mottram (rum@dmi.dk)

**Abstract.** ~~Antarctic ice sheet mass loss is currently equivalent to around 1 mm year$^{-1}$ of global mean sea level rise. Most mass is lost due to sub-ice shelf melting and calving of icebergs. Ice sheet models of the Antarctic ice sheet have thus largely concentrated on parameterising sub-shelf and calving processes. However, surface mass balance (SMB) is also of crucial importance in controlling the stability and evolution of the vast Antarctic ice sheet. In this paper weOur results show that , when regional climate models (RCMs) are forced by the ERA-Interim reanalysis, the integrated Antarctic ice sheet~~ Evaluation of the models shows that they simulate Antarctic climate well when compared with daily observed temperature and pressure though nudged models perform slightly better than un-nudged models. The ensemble mean annual SMB over the AIS including ice shelves

is 2329 $\pm$ 94 Gigatonnes (Gt) year$^{-1}$ over the common 1987 to 2015 period covered by all models. However, mean annual SMB is sensitive to the chosen period with large interannual variability. Over a defined 30 year climatological mean period of 1980 to 2010, the ensemble mean is 2486 Gt year$^{-1}$. However, individual model estimates vary from an annual mean of 1961 $\pm$ 70 to 2519 $\pm$ 118 Gt year$^{-1}$. The  largest spatial differences between model SMB estimates are in West Antarctica and the  Antarctic Peninsula as well as around the

Transantarctic mountains.  Interannual variability is consistent between  models and dominated by variability in the driving ERA-Interim reanalysis.

 We find no significant trend in Antarctic SMB.
 We compare modelled surface mass balance with a large dataset of observations which, though biased by undersampling in some regions, indicates that many of the biases in modelled SMB are common between models.  Drifting-snow schemes improve modelled SMB on ice sheet slopes between 1000 and 2000 m where strong katabatic winds form but other regions where precipitation rates are high lack observations needed for evaluation. Different ice masks have a substantial impact on the integrated total SMB and along with model resolution is  factored into our analysis.  Antarctic ice sheet (AIS) mass loss is currently equivalent to around half a millimetre year$^{-1}$ of global mean sea level rise (Shepherd et al., 2018b) and our results indicate some substantial uncertainty in the surface mass balance (SMB) contribution based on regional climate models. Targeting coastal areas for  observations is key to improving  estimates of the  
[revised manuscript text omitted]

 It is interesting to compare our results with those used in the IMBIE study of Antarctic mass budget (Shepherd et al., 2019). When taking into account the published uncertainties on the observational mass budget estimates from the input-output

20 method, only the COSMO-CLM and MetUM estimates are outside the range defined by the IMBIE study based only on altimetry and GRACE data. However, as these two models, particularly MetUM, perform well in comparison to meteorological observations, the source of the mismatch is unclear and an area that requires significant future work. It may also indicate either

that some of the components of SMB are poorly captured by the models or that there are compensating errors in the modelled SMB components and/or their spatial variability. Nevertheless it is therefore also important to consider the wide uncertainties in both observations and the likely biases in models discussed in this paper, in 
[revised manuscript text omitted]

---

## Referee Report (RR1)

We thank the authors for addressing our comments and making changes. We recognize that work, especially group work that requires coordination, is difficult given the current global pandemic and so we appreciate the authors' efforts in this regard. This paper has improved markedly from the previous version to the current. However, we think substantial revisions remain necessary for this paper to be ready for publication. Below are some general comments that discuss some issues that recur throughout the paper, as well as specific comments that address individual points.

*Tessa Gorte & Jan Lenaerts*

**General Comments**

While the authors have added some additional discussion that addresses our initial comments (and that of the other reviewers), we would encourage them to be more specific and quantitative. For example, the discussion about the relevance to Antarctic mass balance (input-output method) is useful but currently very vague, qualitative, and not supported by any visual evidence (a figure would be useful). More generally, the comparisons between the models in the text remain rather qualitative in many instances, sometimes subjective, and or not supported by numbers and their associated uncertainties. There are numerous examples of this throughout the text, e.g. 'overestimate/underestimate' (by how much?); 'lower bias'; 'best comparison'; 'rather small'; etc. etc.

While much better than the previous iteration of the manuscript, we find that the authors still switch from passive to active voice (and back), sporadically. For instance:
"The model was used to run a series of consecutive twice-daily 24-hour forecasts at 00 and 12 UTC 25 from the beginning of 1980 to the end of 2018." (P5L24-25)
"… in these simulations we used a simplified single-layer scheme with for example, no refreezing (Cox et al., 1999)." (P5L27-28).
"SMB was calculated base on output precipitation and sublimation and evaporation." (P5L28)

The figure quality needs additional improvements. Several figures are still hard to read (see 'Specific Comments' for more details). The font size for some figures, particularly figures with maps, could be increased a bit to help with readability. Figure 3 is especially problematic, since the individual dots are not readable, the lines are difficult to separate, and most importantly, the comparison of observed and modeled SMB is plotted on a double-logarithmic axis, which hides a lot of the scatter between observations and models. A linear fit that is shown on a double logarithmic axis (as is the case in Figure 3), does not imply a linear relation between the observations and models – it might just show that their exponential behavior is linearly related. We would highly encourage the authors to revise Figure 3 substantially to provide a fair and visually engaging and understandable.

Lastly, the third reviewer brought up a very important point, and we feel that hasn't yet been adequately dealt with. If a true intercomparison of models is presented, the common P-E-S budget should be evaluated. So that means that, while blowing snow sublimation can be included, runoff (for MAR and RACMO2) and $ER_{ds}$ (for RACMO2) should not. The author's argument that 'these cannot be easily removed without retuning the models', is only relevant for blowing snow sublimation, not for runoff or erosion.

**Specific Comments**

P1L2: SMB is introduced without spelling out the acronym. Later in the abstract, the phrase is entirely spelled out, though.

P1L5: Similar to SMB, AIS should be spelled out the first time.

P1L11: Consider adding over what period you refer to finding no trend (i.e. 1987-2015 or 1980-2010). It may be over both, in which case you can add "over either period."

P1L14: "between 1000 and 2000 m" what? I'm not sure what that scale is referring to. Is it the length of the slope?

P2L9: fine to use submarine melting here so as not to confuse it with basal melting but still think this term should be defined either within this sentence or in a following sentence.

P2L18: We are a bit confused why the authors introduce the term "surface mass budget" here. If it is synonymous with surface mass balance, surely just using SMB throughout the paper is sufficient. If the authors want to note this term and its equivalence to surface mass balance or climate mass balance, perhaps it would be best to do so when SMB is first introduced.

P2L30: "Currently runoff…" —> "Currently, runoff…"

P2L30-34: This is one of the few remaining run-on sentences that I think could be broken up.

P3L23: "In this paper we seek…" —> "In this paper, we seek…"

P3L29: "…backwards continuity we also…" —> "…backwards continuity, we also…"

P4L9: There should be an and at the end of this list and commas around "among others."

P4L28: Missing space after the end of the sentence starting with "As these terms cannot easily…"

P4L33: "Additionally this model…" —> "Additionally, this model…"

P5L21: "Here we run…" —> "Here, we run…"

P11L2: "…taking note of the differences…" —> "…take note of the differences…"

P11L6: "In figure 1 we show…" —> "In figure 1, we show…"

P12L1: "Figure 1 analysis shows that depending on the variable the models…" —> "Figure 1 analysis shows that, depending on the variable, the models…"

P12L12-13: "…show that although the models perform well (…) on average…" —> "…show that, although the models perform well (…), on average…"

P13L8: Avoid "It is clear." This is a complex figure that, while you describe what's going on well, is still fairly high-level.

P13L23: "This comparison therefore is…" —> "This comparison, therefore, is…"

P15L8-P16L2: This sentence 1) requires commas around "to show… models clearly" and 2) switches from passive voice in the first half to active voice in the second ("are given in Table 2", "we show all models").

P16L3: "Apart from COSMO-CLM2 and HIRHAM5 0.11∘the…" —> "Apart from COSMO-CLM2 and HIRHAM5 0.11∘, the…"

P16L5: "In general all models…" —> "In general, all models…"

P16L6-9: This is another long sentence that could benefit from being broken into two.

P18L3: Consider introducing the concept of SSMB as well as the acronym to help the reader.

P18L10: "…and therefore orographic precipitation." —> "…and, therefore, orographic precipitation."

P22L8: "For example there is an…" —> "For example, there is an…"

P25L5: Missing spaces after the degree symbols.

P27L5: "…allows us to estimate not only the likely range of SMB over Antarctica, but also to identify…" —> "…allows us not only to estimate the likely range of SMB over Antarctica, but also to identify…"

P28L1: "Nevertheless it is therefore also important…" —> "Nevertheless, it is also important…"

P28L5: "…have quite strong trends, for example a steady…" —> "…have quite strong trends: for example, a steady…" or "…have quite strong trends (for example, a steady…)."

P28L7: I believe the comma after "very difficult" is meant to be a period.

P28L12: Again, try to avoid "it is clear" as it may not be clear to all readers.

P28L26: "Basins in West Antarctica, and particularly on the Antarctic peninsula have very large…" —> "Basins in West Antarctica, and particularly on the Antarctic peninsula, have very large…"

P29L1: "We found that although the variation…" —> "We find that, although the variation…"

P29L21: "Nonetheless we are able…" —> "Nonetheless, we are able…"

P29L22-25: This is a long and confusing sentence that could be reworked.

P29L29: "It is therefore important…" —> "Therefore, it is important…"

P29L32: "…perform, broadly speaking better than…" —> "…perform, broadly speaking, better than…"

P30L6: "However table 2 shows…" —> "However, table 2 shows…"

P30L7: Consider changing "clear" to "evident" (or something similar).

P30L12: "Currently efforts are…" —> "Currently, efforts are…"

P31L2: Missing space after the degree symbol.

P21L18: Missing space after the period.

P30L21: "Berg et al. (2013), argue…" —> "Berg et al. (2013) argue…"

P31L31: "In this paper we have compared…" —> "In this paper, we have compared…"

Figure 1 caption: Toward the end of the first sentence, there is an extra space in the (c) notation and a missing space after the period. The curved, centric lines are not explained (do they show CRMSE? What is CRMSE?).

Figure 2: I realize this is in contrast from our previous review so at the risk of sounding contradictory and incredibly nitpick-y, would it be possible to make this figure a touch smaller such that it fits onto the previous page? The font size increase will help significantly if the figure size is reduced and (I think) the figure will still be readable. Also, the y-axis reads "Modeled" which is fine American English but the authors use "modelled" (which is fine British English) throughout the rest of the paper.

Figure 4: This seems like a potentially really great figure but the images are a bit too small to glean a lot of high detail information. Perhaps consider rearranging to make the individual subplots larger.

Table 3: Is the last column meant to have sigma spelled out?

Figure 6: The text on this figure (both the title and the numbers on the scale) are still quite small and hard to read.

Figures 7, 8, & 9: Shouldn't the units for SMB (and precipitation) be Gt/yr and SMB trend Gt/yr2, respectively?

Table 4 caption: "…standard deviations are also show." —> "…standard deviations are also shown."

---

## Author Response (AR2)

**Response to Reviewers Round 2:**

We thank the reviewers and editor again for their very helpful feedback and have endeavoured to implement their suggestions where appropriate, including adding a new figure and substantially revising others to improve readability. Please see below for detailed responses to the comments in blue text..

Editor:
Although still quantitative words are missing as pointed out by the referees,

We have conducted a thorough proofread and inserted more quantitative description of our findings throughout to give a more quantitative understanding. .

figures are providing substantial amount of new information in this work.
Figure 4 compares model SMB and observed SMB. It shows both the mean and the bias with the color bar reflecting the wide range of SMB of 10s kg m-2 yr-1 in the broad inland (above 3000m altitude) to nearly 1000 kg m-2 yr-1 by using unequally spaced color scale, nicely. On the other hand, Figure 6 summarizes the ensemble mean of SMB reanalysis with yet another color bar, without using unequally space color scale and even the other SMB unit (mm yr -1 water equivalent, although the number is the same). Figure 6 will provide more information by using the same unequally spaced color scales as in Figure 4 both for mean and bias. Not only the coast region but also the inland information is very useful for the ice sheet modellers once this work is published.
I am looking forward to seeing the nearly finalized improved version reflecting the comments by the referees.

Thank you very much for this suggestion. We have implemented the suggested changes to Figure 6 to bring it into line with Figure 4 and we feel that it really improves the information provided in this figure. Please see below:

[Figure]

Reviewer 1:

While the authors have added some additional discussion that addresses our initial comments (and that of the other reviewers), we would encourage them to be more specific and quantitative. For example, the discussion about the relevance to Antarctic mass balance (input-output method) is useful but currently very vague, qualitative, and not supported by any visual evidence (a figure would be useful).

We have added more detail in this section of the discussion and a new figure showing the comparison with IMBIE derived mass budget estimate for Antarctic see below:

[Figure]

**Figure 10.** Modelled SMB minus discharge calculated from IMBIE2 results (Shepherd et al., 2018b) (filled circles indicate mean, light grey box indicates IMBIE2 uncertainty range of $\pm 56$ Gt year$^{-1}$ ) and Rignot et al. (2019) (mean showed in filled square, uncertainty range of 142 Gt year$^{-1}$ shown by narrow blue shaded box). The uncertainty for is taken from table 1 in Rignot et al. (2019), assuming the same uncertainty range for the period 2009 to 2017 is applicable over the longer 1992 to 2017 period. The total mass budget estimated by IMBIE2 is also shown by the horizontal dark grey shaded box for ease of comparison. Numbers are mean annual SMB-D for the 1992 to 2017 IMBIE period for each model.

We can compare our results for the total mass budget of Antarctica with those produced by the IMBIE2 study (Shepherdet al., 2018b). In figure 10 we show the SMB-Discharge for two different datasets, where the IMBIE2 (Shepherd et al., 2018b) reconciled estimate of mean annual discharge is 2103±56Gt year−1and the Rignot et al. (2019) estimated discharge of 2247±140Gt year−1 for the same period is subtracted from SMB calculated from each model. We use the simple SMB calculation in equation 1 for the period 1992 to 2017 over the grounded ice sheet only. The Rignot et al. (2019) dataset has a wider uncertainty range than the Shepherd et al. (2018b) estimate and a larger discharge that gives a lower total mass budget overall, but in all cases the two overlap within the uncertainty ranges. Note that the RACMO2.3p2 model was used to produce both the IMBIE2 and Rignot et al. (2019) estimates and it is thus not a truly independent comparison. The earlier MARv3.6 model was

also included in the Shepherd et al. (2018b) study. When taking into account the published uncertainties on the observational mass budget estimates of discharge, only the COSMO-CLM2 and MetUM estimates are completely outside the range defined by the IMBIE study (109±56Gt year−1) for the total mass budget of Antarctica. However, both models perform well compared to the weather station observation, particularly MetUM, and both have higher correlations and lower biases than the two HIRHAM simulations (see figure 1) for pressure and temperature. Comparison with the SMB observations shows that COSMO-CLM2 has a large dry bias (of 40%) over ice shelves and at lower elevations, this bias is larger than the other models over the ice shelves and up to 1200m in elevation, but at higher elevations the mean bias is close to zero for the COSMO-CLM2 model and in fact much lower than the other models in the 2800 to 3400m elevation range (see figure 4). MetUM on the other hand has a middle of the range mean bias at low elevations compared to other models but a much higher (-25 to -30%) mean bias as shown in figure 4 at the upper elevations. The combination of these results, bearing in mind also the undersampling in the dataset, thus indicate either that some of the components of SMB are poorly captured by the models or that there are compensating errors in the modelledSMB components and/or their spatial variability and most likely a combination of factors. This means that there are large uncertainties in both observations as well as the biases in models that we discuss in this paper, that complicate assessing the contribution to sea level rise from Antarctica from SMB processes.

More generally, the comparisons between the models in the text remain rather qualitative in many instances,sometimes subjective,and or not supported by numbers and their associated uncertainties.There are numerous examples of this throughout the text, e.g. 'overestimate/underestimate' (by how much?); 'lower bias'; 'best comparison'; 'rather small'; etc. etc.

We have conducted a thorough proofread and revised text where necessary to provide more quantitative description of our findings with statistics to support qualitative statements.

While much better than the previous iteration of the manuscript, we find that the authors still switch from passive to active voice (and back), sporadically. For instance:

Thankyou for the very thorough comments on our text. We have again proofread the final draft and we have hopefully managed to remove all instances of passive voice now. See below for specific edits.

"The model was used to run a series of consecutive twice-daily 24-hour forecasts at 00 and 12 UTC 25 from the beginning of 1980 to the end of 2018." (P5L24-25)

Fixed to "For this study we ran a series of consecutive twice-daily 24-hour forecasts.."

"... in these simulations we used a simplified single-layer scheme with for example, no refreezing (Cox et al., 1999)." (P5L27-28).

"SMB was calculated based on output precipitation and sublimation and evaporation." (P5L28)

"We therefore calculate SMB based on output precipitation and sublimation and evaporation."

The figure quality needs additional improvements. Several figures are still hard to read(see 'Specific Comments' for more details). The font size for some figures, particularly figures with maps, could be increased a bit to help with readability.

Figure 3 is especially problematic, since the individual dots are not readable, the lines are difficult to separate, and most importantly, the comparison of observed and modeled SMB is plotted on a double-logarithmic axis, which hides a lot of the scatter between observations and models. A linear fit that is shown on a double logarithmic axis(as is the case in Figure 3), does not imply a linear relation between the observations and models–it might just show that their exponential behavior is linearly related. We would highly encourage the authors to revise Figure 3 substantially to provide a fair and visually engaging and understandable.

We have revised all figures in line with the specific comments (below) to make them more readable. It is unfortunately difficult to include all of the information we would like on all the plots we would like when including so many models. However, we agree it is difficult to read some of the text and hopefully the revisions have substantially improved.

Figure 3 is now as shown below with larger labels and point markers and is therefore we hope much clearer. As before, we have presented the individual model comparisons in the appendix to avoid filling the paper up with too many figures.We prefer to keep the double log axes because the distribution of both the observed and simulated SMB is not Gaussian. This means a linear regression will be strongly influenced by the extreme values, especially the $r^2$, which is skewed by the errors for the largest values but only weakly influenced by the errors on the smallest values. This means that in the absolute numbers, the error is greater for large (observed and simulated) SMB values than for small values, while in relative terms, it may be exactly the same error. Using log/log axes makes the distributions Gaussian and enables us to calculate a linear regression with unbiased coefficients. Using linear axes would mean that the comparison is biased by the errors for the largest values and we can now see how the models reproduce the low values. The associated statistics show both r and rlog so the linear relationship is also given.

[Figure]

**Figure 3.** Comparison between modelled SMB and observed SMB in a gridded dataset. Trend lines and points are plotted for each model in a different colour, note different x and y axes for different elevation bins. The figures are plotted on logarithmic axes because the datasets are not gaussian distributed and this better represents the relative error in both high and low SMB regions.

We have clarified this in the figure caption (see above) and inserted the following text :

"Note that Figure 3 is plotted on logarithmic axes because the distribution of both the observed and simulated SMB is not Gaussian. As linear regression is strongly influenced by the extreme values, which skew r2 errors in both modelled and observed SMB for the largest values but is only weakly influenced by the errors on the smallest absolute values, a logarithmic plot better displays how well models reproduce SMB in both high and low SMB regions."

Lastly, the third reviewer brought up a very important point, and we feel that hasn't yet been adequately dealt with. If a true intercomparison of models is presented, the common P-E-S budget should be evaluated. So that means that, while blowing snow sublimation can be included, runoff (for MAR and RACMO2) and ER$_{ds}$(for RACMO2) should not. The author's argument that 'these cannot be easily removed without retuning the models', is only relevant for blowing snow sublimation, not for runoff erosion.

We apologise for causing some confusion here. It is true that the versions of the MAR and RACMO models presented here have fully optimised SMB schemes including runoff, and in the case of RACMO the wind blown erosion of snow,compared to the other models. The SMB from using these full schemes is presented in Table 3. However, in table 4 we also present SMB as calculated using only the simple SMB as described in equation 1 along with the components precipitation and sublimation for all models and the ERA-Interim reanalysis in order to enable exactly this kind of comparison. However we had not noted this very

clearly in the text. This has now been clarified with the following text in the discussion and updated in the table caption.

$$SMB = precipitation - evaporation - sublimation \qquad (1)$$

"We calculate the mean annual SMB and components across the continent including ice shelves, as given in Table 4, over the period 1987 to 2015 for which outputs are available for all the models. Note that this is calculated over a common ice mask and a common simulation period and using the simple SMB calculation given in equation 1 and results are therefore slightly different to those already published for different models or shown in Figure 5 or Table 3."

**Specific Comments**

P1L2: SMB is introduced without spelling out the acronym. Later in the abstract, the phrase is entirely spelled out, though.
Fixed

P1L5: Similar to SMB, AIS should be spelled out the first time.
Fixed

P1L11: Consider adding over what period you refer to finding no trend (i.e. 1987-2015 or 1980-2010). It may be over both, in which case you can add "over either period."
Fixed

P1L14: "between 1000 and 2000 m" what? I'm not sure what that scale is referring to. Is it the length of the slope?
It refers to the surface elevation of the ice sheet. This has been clarified "Drifting-snow schemes improve modelled SMB on ice sheet surface slopes with an elevation between 1000 and 2000 m where strong katabatic winds form"

P2L9: fine to use submarine melting here so as not to confuse it with basal melting but still think this term should be defined either within this sentence or in a following sentence.
Fixed
"Most ice loss in Antarctica occurs as a result of submarine melting, that is melt at the water-ice interface underneath ice shelves, or by the calving of icebergs from ice shelves."

P2L18: We are a bit confused why the authors introduce the term "surface mass budget" here. If it is synonymous with surface mass balance, surely just using SMB throughout the paper is sufficient. If the authors want to note this term and its equivalence to surface mass balance or climate mass balance, perhaps it would be best to do so when SMB is first introduced.

This was a relic of an earlier draft. We have moved the phrase " also known as surface mass balance or climate mass balance" to where the abbreviation SMB is described for the first time for the sake of completeness.

P2L30: "Currently runoff..." —> "Currently, runoff..."
Fixed

P2L30-34: This is one of the few remaining run-on sentences that I think could be broken up.
Adjusted to: Currently, runoff is a relatively minor contribution (Lenaerts et al., 2019) to mass loss in Antarctica. Increasing snowfall, associated with higher saturated vapour pressure is expected to dominate future changes in SMB, compensating for the projected increase in surface runoff (Krinner et al., 2008; Lenaerts et al., 2016) but the balance between these processes is still a matter of debate.

P3L23: "In this paper we seek..." —> "In this paper, we seek..."
Fixed

P3L29: "...backwards continuity we also..." —> "...backwards continuity, we also..."
Fixed

P4L9: There should be an and at the end of this list and commas around "among others."
Fixed

P4L28: Missing space after the end of the sentence starting with "As these terms cannot easily..."
Fixed
P4L33: "Additionally this model..." —> "Additionally, this model..."
Fixed

P5L21: "Here we run..." —> "Here, we run..."

Fixed

P11L2: "...taking note of the differences..." —> "...take note of the differences..."
Fixed
P11L6: "In figure 1 we show..." —> "In figure 1, we show..."
Fixed

P12L1: "Figure 1 analysis shows that depending on the variable the models..." —> "Figure 1 analysis shows that, depending on the variable, the models..."
Fixed

P12L12-13: "...show that although the models perform well (...) on average..." —> "...show that, although the models perform well (...), on average..."
Fixed

P13L8: Avoid "It is clear." This is a complex figure that, while you describe what's going on well, is still fairly high-level.
Fixed to: "Figure 1 shows that all of the models perform less well for wind speeds than for temperature or pressure observations. The wind speed plot shows all models have higher CRMSE, higher standard deviation and lower correlation values when compared with observations."

P13L23: "This comparison therefore is..." —> "This comparison, therefore, is..."
Fixed

P15L8-P16L2: This sentence 1) requires commas around "to show... models clearly" and 2) switches from passive voice in the first half to active voice in the second ("are given in Table 2", "we show all models").
Fixed: "We show detailed statistics for the SMB comparison in Table 2. In order to show the large scatter in the observations and the models clearly, we also plot all modelled SMB values against observed SMB values in Figure"

P16L3: "Apart from COSMO-CLM2 and HIRHAM5 0.11∘the..." —> "Apart from COSMO-CLM2 and HIRHAM5 0.11∘, the..."
Fixed

P16L5: "In general all models..." —> "In general, all models..."

Fixed

P16L6-9: This is another long sentence that could benefit from being broken into two.
Fixed: "In general, all models underestimate SMB over the ice shelves and at the low elevation coastal regions of Antarctica (see also statistics in Table \ref{tab:samba_smb} a. and b. and Figure \ref{smb_obs_comp}). The highest mean bias, lowest RMSE and lowest r values in particular are given in the COSMO-CLM$^{2}$ and HIRHAM5 0.11\degree models at the lowest elevations."

P18L3: Consider introducing the concept of SSMB as well as the acronym to help the reader.
We have slightly restructured the paragraph to bring the explanation of SSMB forward.

P18L10: "...and therefore orographic precipitation." —> "...and, therefore, orographic precipitation."
Fixed

P22L8: "For example there is an..." —> "For example, there is an..."
Fixed

P25L5: Missing spaces after the degree symbols.
Fixed

P27L5: "...allows us to estimate not only the likely range of SMB over Antarctica, but also to identify..." —> "...allows us not only to estimate the likely range of SMB over Antarctica, but also to identify..."
Fixed

P28L1: "Nevertheless it is therefore also important..." —>"Nevertheless, it is also important..."
Fixed

P28L5: "...have quite strong trends, for example a steady..." —> "...have quite strong trends: for example, a steady..." or "...have quite strong trends (for example, a steady...)."

Fixed to: "Shorter periods within the time series appear to have quite strong trends. For example, a steady declining trend is apparent through the 1990s and 2000s but appears to reverse after 2014."

P28L7: I believe the comma after "very difficult" is meant to be a period.
Yes it was. Fixed

P28L12: Again, try to avoid "it is clear" as it may not be clear to all readers.
Fixed to "Figure 8 demonstrates.."

P28L26: "Basins in West Antarctica, and particularly on the Antarctic peninsula have very large..." —> "Basins in West Antarctica, and particularly on the Antarctic peninsula, have very large..."
Fixed

P29L1: "We found that although the variation..." —> "We find that, although the variation..."
Fixed

P29L21: "Nonetheless we are able..." —> "Nonetheless, we are able..."
Fixed - see below

P29L22-25: This is a long and confusing sentence that could be reworked.
The whole paragraph has been reworked:
"Evaluating the models against observations is very important for assessing where there are important biases, but evaluation of model performance is significantly hampered by the lack of observations in key regions. Nonetheless, Figure 1 shows that the models do have skill in simulating surface climate, particularly temperature and pressure. The skill in simulating surface climate does not however translate perfectly to simulating SMB, partly due to the difficulties of modelling and evaluating precipitation. Our analysis shows that for example, COSMO-CLM2 better simulated surface climate compared to observations than HIRHAM5 but it has a lower skill in SMB. Variables such as temperature and pressure are more easily measured and are assimilated into the reanalysis used to drive the models. RCMs have also been optimised to give good performance compared to these kinds of observations. However, Antarctic SMB is dominated by the precipitation term that is much harder to measure accurately and also has much higher uncertainty in models

P29L29: "It is therefore important..." —> "Therefore, it is important..."
Fixed

P29L32: "...perform, broadly speaking better than..." —> "...perform, broadly speaking, better than..."

Fixed

P30L6: "However table 2 shows..." —> "However, table 2 shows..."
Fixed

P30L7: Consider changing "clear" to "evident" (or something similar).
Fixed

P30L12: "Currently efforts are..." —> "Currently, efforts are..."P31L2: Missing space after the degree symbol.P21L18: Missing space after the period.
Fixed

P30L21: "Berg et al. (2013), argue..." —> "Berg et al. (2013) argue..."
Fixed

P31L31: "In this paper we have compared..." —> "In this paper, we have compared..."
Fixed

Figure 1 caption: Toward the end of the first sentence, there is an extra space in the (c) notation and a missing space after the period.The curved, centric lines are not explained (do they show CRMSE? What is CRMSE?).

We have remade the figure to enlarge it and remove the typo errors. As we explain in the caption: "Taylor diagrams showing model performance compared to daily observations of surface pressure (a), near-surface temperature (b), and observed wind speeds ( c).The horizontal and >vertical axes represent the standard deviation, the dashed line in bold shows the standard deviation of the observations. The Taylor plot also shows the correlation which is measured by the angle with the x-axis. Finally,the CRMSE is represented by the curved lines in light grey. The units of standard deviation, CRMSE, mean bias and mean of the observations are the same (hPa for surface pressure, K for near-surface temperature, and m/s >for wind speed)."
We also explain CRMSE in more detail just above Figure 1 and have added the equation to show more fully how it is calculated. 'CRMSE is equivalent to the Root Mean Square Error but systematic biases are removed by subtracting the mean observation and mean modelled values from each value.'

10  value as shown in equation 4

$$CRMSE = \sqrt{\frac{\sum_{i=0}^{n}(m_i - o_i)^2}{n} - (\overline{m} - \overline{o})^2}$$  (4)

Where n is the number of observations, $m_i$ is the modelled value, $o_i$ is the observed value and $\overline{m}$ and $\overline{o}$ are the average of the modelled and observed values respectively.

Figure 2: I realize this is in contrast from our previous review so at the risk of sounding contradictory and incredibly nitpick-y, would it be possible to make this figure a touch smaller such that it fits onto the previous page? The font size increase will help significantly if the figure size is reduced and (I think) the figure will still be readable. Also, the y-axis reads "Modeled" which is fine American English but the authors use "modelled" (which is fine British English) throughout the rest of the paper.

We have resized the image and the font to make it fit better on the page. The typo has been fixed back to British English to align with the rest of the manuscript.

Figure 4: This seems like a potentially really great figure but the images are a bit too small to glean a lot of high detail information. Perhaps consider rearranging to make the individual subplots larger.

We have reshaped this figure to make it clearer as below:

[Figure]

Table 3: Is the last column meant to have sigma spelled out?
Yes, a typo was introduced in the latex code. Now fixed.

Figure 6: The text on this figure (both the title and the numbers on the scale) are still quite small and hard to read.

We have substantially revised Figure 6 with a different colour scheme consistent with that in Figure 4 and larger text.

[Figure]

**Figure 6.** Sub-figure **a** shows the SMB ensemble mean for the common period, on the common mask. Sub-figure **b-g** show the difference between each model and the ensemble mean.

Figures 7, 8, & 9: Shouldn't the units for SMB (and precipitation) be Gt/yr and SMB trend Gt/yr2, respectively?

We recognise that the figure titles are perhaps a bit confusing here so we have relabelled these graphs to reduce the confusion around what they present. As they show the year on the x-axis the y-axis unit should be Gigatonnes. Figure 7 shows the annual variability in SMB for each model by subtracting the mean value so in fact the quantities are also Gigatonnes here.

Table 4 caption: "...standard deviations are also show." —> "...standard deviations are also shown

Fixed